# PROVABLE LEARNING-BASED ALGORITHM FOR SPARSE RECOVERY

**Xinshi Chen & Haoran Sun**
School of Mathematics
Georgia Institute of Technology
Atlanta, USA
{xinshi.chen,haoransun}@gatech.edu

**Le Song**
Machine Learning Department
MBZUAI & BioMap
UAE & China
songle@biomap.com

## ABSTRACT

Recovering sparse parameters from observational data is a fundamental problem in machine learning with wide applications. Many classic algorithms can solve this problem with theoretical guarantees, but their performances rely on choosing the correct hyperparameters. Besides, hand-designed algorithms do not fully exploit the particular problem distribution of interest. In this work, we propose a deep learning method for algorithm learning called `PLISA` (Provable Learning-based Iterative Sparse recovery Algorithm). `PLISA` is designed by unrolling a classic path-following algorithm for sparse recovery, with some components being more flexible and learnable. We theoretically show the improved recovery accuracy achievable by `PLISA`. Furthermore, we analyze the empirical Rademacher complexity of `PLISA` to characterize its generalization ability to solve new problems outside the training set. This paper contains novel theoretical contributions to the area of learning-based algorithms in the sense that (i) `PLISA` is generically applicable to a broad class of sparse estimation problems, (ii) generalization analysis has received less attention so far, and (iii) our analysis makes novel connections between the generalization ability and algorithmic properties such as stability and convergence of the unrolled algorithm, which leads to a tighter bound that can explain the empirical observations. The techniques could potentially be applied to analyze other learning-based algorithms in the literature.

## 1 INTRODUCTION

The problem of recovering a sparse vector $\boldsymbol{\beta}^*$ from finite observations $Z_{1:n} \sim (\mathbb{P}_{\boldsymbol{\beta}^*})^n$ is fundamental in machine learning, covering a broad family of problems including compressed sensing, sparse regression analysis, graphical model estimation, etc. It has also found applications in various domains. For example, in magnetic resonance imaging, sparse signals need to be reconstructed from measurements taken by a scanner. In computational biology, estimating a sparse graph structure from gene expression data is important for understanding gene regulatory networks.

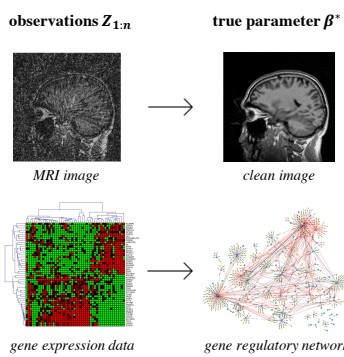

observations $Z_{1:n}$     true parameter $\boldsymbol{\beta}^*$

*MRI image*     *clean image*

*gene expression data*     *gene regulatory network*

Figure 1: Sparse recovery problems.

Various classic algorithms are available for solving sparse recovery problems. Many of them come with theoretical guarantees for the recovery accuracy. However, the theoretical performance often relies on choosing the correct hyperparameters, such as regularization parameters and the learning rate, which may depend on unknown constants. Furthermore, in practice, similar problems may need to be solved repeatedly, but it is hard for classic algorithms to fully utilize this information.

To alleviate these limitations, we consider the approach of learning-to-learn and propose a neural algorithm, called `PLISA` (Provable Learning-based Iterative Sparse recovery Algorithm). `PLISA` is a deep learning model that takes the observations $Z_{1:n}$ as the input and outputs an estimation for $\boldsymbol{\beta}^*$. To make use of classic techniques developed by domain experts, we design the architecture of `PLISA` by *unrolling and modifying a classic path-following algorithm* proposed by Wang et al.

(2014). To benefit from learning, some components in this classic algorithm are made more flexible with careful design and treated as learnable parameters in `PLISA`. These parameters can be learned by optimizing the performances on a set of training problems. The learned `PLISA` can then be used for solving other problems in the target distribution.

With the algorithm design problem converted to a deep learning problem, we ask the two fundamental questions in learning theory:

1. **Capacity:** What's the recovery accuracy achievable by `PLISA`? Can the flexible components in `PLISA` lead to an algorithm which effectively improves the recovery performance?
2. **Generalization:** How well can the learned `PLISA` solve new problems outside the training set? Is the generalization behavior related to the algorithmic properties of `PLISA`?

Aiming at supplying rigorous answers to these questions, we conduct theoretical analysis for `PLISA` to provide guarantees for its representation and generalization ability. The results and the techniques in our analysis can distinguish our work from existing studies on algorithm learning. We summarize our novel contributions into the following three aspects.

**1. Theoretical understanding.** In contrast to the plethora of empirical studies on algorithm learning, there have been relatively few studies devoted to the theoretical understanding. Existing theoretical efforts primarily focus on analyzing the convergence rate achievable by the neural algorithm (Chen et al., 2018; Liu et al., 2019a; Zhang & Ghanem, 2018; Wu et al., 2020), but the generalization error bound has received less attention so far. A substantial body of works only argue intuitively that algorithm unrolling architectures can generalize well because they contain a small number of parameters. In comparison, we provide theoretical guarantees for both the capacity and the generalization ability of `PLISA`, which are more solid arguments.

**2. General setting.** The problem setting in this paper is new and more challenging. Existing works mainly focus on a specific problem. For example, the compressed sensing problem with a fixed design matrix is the mostly investigated one. `PLISA`, however, is generic and is applicable to various sparse recovery problems as long as they satisfy certain conditions in Assumption C.1.

**3. Novel connection.** The algorithmic structure in `PLISA` can make it behaves differently from conventional neural networks. Therefore, we largely utilize the analysis techniques in classic algorithms to derive its generalization bound. By combining the analysis tools of deep learning theory and optimization algorithms, our result reveals a novel connection between the generalization ability of `PLISA` and the algorithmic properties including the convergence rate and stability of the unrolled algorithm. Benefit from this connection, our generalization bound is tight in the sense that it matches the interesting behavior of `PLISA` observed in experiments - the generalization gap could decrease in the number of layers, which is rarely observed in conventional neural networks.

## 2 `PLISA`: Learning To Solve Sparse Estimation Problems

A sparse estimation problem is to recover $\boldsymbol{\beta}^*$ from finite observations $Z_{1:n}$ sampled from $\mathbb{P}_{\boldsymbol{\beta}^*}$. As a concrete example, in a sparse linear regression problem, $n$ observations $\{Z_i = (\boldsymbol{x}_i, y_i)\}_{i=1}^n$ are sampled from a linear model $y = \boldsymbol{x}^\top \boldsymbol{\beta}^* + \epsilon$, and an algorithm needs to estimate $\boldsymbol{\beta}^*$ from $n$ observations. Classic algorithms recover $\boldsymbol{\beta}^*$ by minimizing a regularized empirical loss:

$$\widehat{\boldsymbol{\beta}}_\lambda \in \arg\min L_n(Z_{1:n}, \boldsymbol{\beta}) + P(\lambda, \boldsymbol{\beta}), \tag{1}$$

where $L_n$ is an empirical loss that measures the "fit" between the parameter $\boldsymbol{\beta}$ and observations $Z_{1:n}$, and $P(\lambda, \boldsymbol{\beta})$ is a sparsity regularization with coefficient $\lambda$. When $L_n$ is the least square loss and $P(\lambda, \boldsymbol{\beta})$ is $\lambda\|\boldsymbol{\beta}\|_1$, the optimization is known as LASSO and can be solved by the well-known algorithm ISTA (Daubechies et al., 2004). Based on the idea of **algorithm unrolling**, Gregor & LeCun (2010) proposed LISTA, a neural algorithm that interprets ISTA as layers of neural networks. It has been demonstrated that LISTA outperforms ISTA thanks to its learnable components. Since then, designing neural algorithms by unrolling ISTA has become an active research topic. However, existing works mostly focus on the compressed sensing problem with a fixed design matrix only.

To enable for more general applicability, we design the architecture of `PLISA` by unrolling a classic path-following algorithm called **APF** (Wang et al., 2014) instead of ISTA. APF is applicable to nonconvex losses and nonconvex penalty functions, covering a considerably larger range of objectives than LASSO. Designing the architecture based on APF allows `PLISA` to be applicable to

a broader class of problems such as nonlinear sparse regression, graphical model estimation, etc. Furthermore, employing nonconvexity can potentially lead to better statistical properties (Fan & Li, 2001; Fan et al., 2009; Loh & Wainwright, 2015), for which we will explain more in Section 3.

In the following, we will introduce APF and the architecture of our proposed PLISA. After that, we will describe how to optimize the parameters in PLISA under the learning-to-learn setting.

## 2.1 A BRIEF INTRODUCTION TO APF

We briefly introduce the classic algorithm APF (Wang et al., 2014), and its details are presented in Algorithm 3 in Appendix I. The key idea of path-following algorithms is creating **a sequence of** $T$ **many sub-objectives** to gradually approach the target objective that is supposed to be more difficult to solve. More specifically, APF approximates the local minimizers of a sequence of sub-objectives:

$$\boldsymbol{\beta}_t \approx \widehat{\boldsymbol{\beta}}_{\lambda_t} \in \arg\min_{\boldsymbol{\beta}} L_n(Z_{1:n}, \boldsymbol{\beta}) + P(\lambda_t, \boldsymbol{\beta}), \quad \text{for } t = 1, \cdots, T, \tag{2}$$

where $\lambda_1 > \lambda_2 > \cdots > \lambda_T$ is a decreasing sequence of regularization parameters. The last parameter $\lambda_T$ is the target regularization parameter. As a result, APF contains $T$ blocks, and each block contains an iterative algorithm that minimizes one sub-objective in Eq. 2. The output of the $(t-1)$-th block, denoted by $\boldsymbol{\beta}_{t-1}$, is used as the initialization of the $t$-th block, i.e., $\widetilde{\boldsymbol{\beta}}_t^0 = \boldsymbol{\beta}_{t-1}$. Then the $t$-th block minimizes the $t$-th sub-objective by the **modified proximal gradient** algorithm:

$$\text{for } k = 1, \cdots, K, \quad \widetilde{\boldsymbol{\beta}}_t^k \leftarrow \mathcal{T}_{\alpha \cdot \lambda_t} \left( \widetilde{\boldsymbol{\beta}}_t^{k-1} - \alpha \big( \nabla_{\boldsymbol{\beta}} L_n(Z_{1:n}, \widetilde{\boldsymbol{\beta}}_t^{k-1}) + \nabla_{\boldsymbol{\beta}} Q(\lambda_t, \widetilde{\boldsymbol{\beta}}_t^{k-1}) \big) \right). \tag{3}$$

$$\text{output of } t\text{-th block: } \quad \boldsymbol{\beta}_t = \widetilde{\boldsymbol{\beta}}_t^K. \tag{4}$$

The notation $\mathcal{T}_\delta(\boldsymbol{\beta}) := \text{sign}(\beta) \max\{|\boldsymbol{\beta}| - \delta, 0\}$ is the soft-thresholding function, and the function $Q$ is the concave component of $P$ defined as $Q(\lambda, \boldsymbol{\beta}) := P(\lambda, \boldsymbol{\beta}) - \lambda \|\boldsymbol{\beta}\|_1$. The number of steps $K$ in each block is determined by certain stopping criteria. It can be seem that the update steps in each block is similar to the ISTA algorithm, but it is modified in order to incorporate nonconvexity.

## 2.2 ARCHITECTURE OF PLISA

The architecture of PLISA is designed by unrolling the APF algorithm, and augmenting some learnable parameters $\theta$. Therefore, the architecture of $\text{PLISA}_\theta$ contains $T$ blocks and each block contains $K$ layers defined by the $K$-step algorithm in Eq. 3 (See Figure 2). Note that in $\text{PLISA}_\theta$, both $K$ and $T$ are pre-defined. The architecture of $\text{PLISA}_\theta$ is **different** from APF as summarized below:

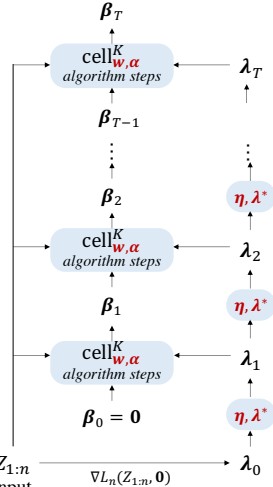

Figure 2: Architecture.

1. **Element-wise and learnable regularization parameters.** Most classic algorithms including APF employ a uniform regularization parameter $\lambda_t$ across all entries of $\boldsymbol{\beta}$, but $\text{PLISA}_\theta$ uses a $d$-dimensional vector $\boldsymbol{\lambda}_t = [\lambda_{t,1}, \cdots, \lambda_{t,d}]^\top$ to enforce different levels of sparsity to different entries in $\boldsymbol{\beta}$. Furthermore, the regularization parameters in $\text{PLISA}_\theta$ are learnable, which will be optimized during the training.

2. **Learnable penalty function:** Classic algorithms use a pre-defined sparse penalty function $P$, but $\text{PLISA}_\theta$ parameterizes it as a combination of $q$ different penalty functions and learns the weights of each penalty. In other words, $\text{PLISA}_\theta$ can learn to select a specific combination of the penalty functions from training data.

3. **Learnable step size:** The step sizes in APF are selected by line-search but they are learnable in $\text{PLISA}_\theta$. Experimentally, we find the learned step sizes lead to a much faster algorithm.

In later sections of this paper, we will show how such differences can make $\text{PLISA}_\theta$ perform better than APF both empirically and theoretically.

Algorithm 1 and 2 present the mathematical details of the architecture, follow which we explain some notations and definitions. Red-colored symbols indicate learnable parameters in $\text{PLISA}_\theta$.

**Algorithm 1:** `PLISA`$_\theta$ architecture

*#blocks: T, #layers per block: K*
*Parameters:* $\theta = \{\boldsymbol{\eta}, \boldsymbol{\lambda}^*, \boldsymbol{w}, \alpha\}$
*Input:* samples $Z_{1:n}$
$\boldsymbol{\beta}_0 \leftarrow \mathbf{0}, \quad \boldsymbol{\lambda}_0 \leftarrow \nabla_{\boldsymbol{\beta}} L_n(Z_{1:n}, \mathbf{0})$
1 **For** $t = 1, \ldots, T$ **do**
  $\boldsymbol{\lambda}_t \leftarrow \max\left\{\sigma(\boldsymbol{\eta}) \circ \boldsymbol{\lambda}_{t-1}, \boldsymbol{\lambda}^*\right\}$
  $\boldsymbol{\beta}_t \leftarrow \texttt{Block}_{\boldsymbol{w}, \alpha}^K (Z_{1:n}, \boldsymbol{\beta}_{t-1}, \boldsymbol{\lambda}_t)$
**return** $\boldsymbol{\beta}_T$

**Algorithm 2:** Layers in each block $\texttt{Block}_{\boldsymbol{w}, \alpha}^K$

*Input:* $Z_{1:n}, \boldsymbol{\beta}_{t-1}, \boldsymbol{\lambda}_t$
$\widetilde{\boldsymbol{\beta}}_t^0 \leftarrow \boldsymbol{\beta}_{t-1}$
**For** $k = 1, \ldots, K$ **do**
1   $\boldsymbol{g}_t^k \leftarrow \nabla_{\boldsymbol{\beta}} L_n(Z_{1:n}, \widetilde{\boldsymbol{\beta}}_t^{k-1}) + \nabla_{\boldsymbol{\beta}} Q_{\boldsymbol{w}}(\boldsymbol{\lambda}_t, \widetilde{\boldsymbol{\beta}}_t^{k-1})$
  $\widetilde{\boldsymbol{\beta}}_t^k \leftarrow \mathcal{T}_{\alpha \cdot \boldsymbol{\lambda}_t}\left(\widetilde{\boldsymbol{\beta}}_t^{k-1} - \alpha \cdot \boldsymbol{g}_t^k\right)$
**return** $\boldsymbol{\beta}_t = \widetilde{\boldsymbol{\beta}}_t^K$

**Regularzation parameters.** In `PLISA`$_\theta$, the element-wise regularization parameters are initialized by a vector $\boldsymbol{\lambda}_0 := \nabla_{\boldsymbol{\beta}} L_n(Z_{1:n}, \mathbf{0})$, and then updated sequentially by

$$\boldsymbol{\lambda}_t \leftarrow \max\left\{\sigma(\boldsymbol{\eta}) \circ \boldsymbol{\lambda}_{t-1}, \boldsymbol{\lambda}^*\right\}, \tag{5}$$

where $\sigma(\cdot)$, $\circ$, and $\max\{\cdot, \cdot\}$ are element-wise sigmoid function, multiplication, and maximization. $\{\boldsymbol{\eta}, \boldsymbol{\lambda}^*\}$ are both $d$-dimensional learnable parameters. Eq. 5 creates a sequence $\boldsymbol{\lambda}_1, \cdots, \boldsymbol{\lambda}_T$ through the decrease ratio $\sigma(\boldsymbol{\eta})$, until they reach the target regularization parameters $\boldsymbol{\lambda}^*$.

**Penalty function.** `PLISA`$_\theta$ parameterizes the penalty function as follows,

$$P_{\boldsymbol{w}}(\boldsymbol{\lambda}, \boldsymbol{\beta}) = \sum_{i=1}^q \widetilde{w_i} \cdot P^{(i)}(\boldsymbol{\lambda}, \boldsymbol{\beta}), \quad \text{where } \widetilde{w_i} = \frac{\exp(w_i)}{\sum_{i'=1}^q \exp(w_{i'})}. \tag{6}$$

In other words, $P_{\boldsymbol{w}}$ is a learnable convex combination of $q$ penalty functions $(P^{(1)}, \cdots, P^{(q)})$. The weights of these functions are determined by learnable parameters $\boldsymbol{w} = [w_1, \cdots, w_q]$. In this paper, we focus on learning the combination of three well-known penalty functions:

$$P^{(1)}(\boldsymbol{\lambda}, \boldsymbol{\beta}) = \|\boldsymbol{\lambda} \circ \boldsymbol{\beta}\|_1, \ P^{(2)}(\boldsymbol{\lambda}, \boldsymbol{\beta}) = \sum_{j=1}^d \text{MCP}(\lambda_j, \beta_j), \ P^{(3)}(\boldsymbol{\lambda}, \boldsymbol{\beta}) = \sum_{j=1}^d \text{SCAD}(\lambda_j, \beta_j),$$

where $P^{(1)}$ is convex, and MCP (Zhang, 2010a) and SCAD (Fan & Li, 2001) are *nonconvex penalties* whose analytical forms are given in Appendix B. One can include any other penalty functions as long as they satisfy a set of conditions specified in Appendix B. $Q_{\boldsymbol{w}}(\boldsymbol{\lambda}, \boldsymbol{\beta}) := P_{\boldsymbol{w}}(\boldsymbol{\lambda}, \boldsymbol{\beta}) - \|\boldsymbol{\lambda} \circ \boldsymbol{\beta}\|_1$ represents the concave component of $P_{\boldsymbol{w}}$. The analytical form of $\nabla_{\boldsymbol{\beta}} Q_{\boldsymbol{w}}(\boldsymbol{\lambda}_t, \boldsymbol{\beta})$ are in Appendix B.

## 2.3 LEARNING-TO-LEARN SETTING

Now we describe how to train the parameters $\theta$ in `PLISA`$_\theta$ under the learning-to-learn setting.

**Training set.** Similar to other works in this domain, we assume the access to $m$ problems from the target problem-space $\mathcal{P}$, and use them as the training set:

$$\mathcal{D}_m = \{(Z_{1:n_1}^{(1)}, \boldsymbol{\beta}^{*(1)}), \cdots, (Z_{1:n_m}^{(m)}, \boldsymbol{\beta}^{*(m)})\} \quad \text{with} \quad (Z_{1:n_i}^{(i)}, \boldsymbol{\beta}^{*(i)}) \in \mathcal{P}.$$

Here each estimation problem is represented by a pair of observations and the corresponding true parameter to be recovered. A different problem $i$ can contain a different number $n_i$ of observations.

**Training loss.** Since the intermediate outputs $\boldsymbol{\beta}_t(Z_{1:n}; \theta)$ of `PLISA`$_\theta$ are also estimates of $\boldsymbol{\beta}^*$, a common design of the training loss is the weighted sum of the intermediate estimation errors (Chen et al., 2021). More specifically, we employ the following training loss:

$$\mathcal{L}_{train}^\gamma(\mathcal{D}_m; \theta) := \frac{1}{m} \sum_{i=1}^m \sum_{t=1}^T \gamma^{T-t} \left\|\boldsymbol{\beta}_t(Z_{1:n_i}^{(i)}; \theta) - \boldsymbol{\beta}^{*(i)}\right\|_2^2, \tag{7}$$

where $\gamma < 1$ is a discounting factor If $\gamma = 0$ then the loss is only estimated at the last layer.

**Generalization error.** The ultimate goal of algorithm learning is to minimize the estimation error on expectation over all problems in the target problem distribution:

$$\mathcal{L}_{gen}(\mathbb{P}(\mathcal{P}); \theta) := \mathbb{E}_{(Z_{1:n}, \boldsymbol{\beta}^*) \sim \mathbb{P}(\mathcal{P})} \|\boldsymbol{\beta}_T(Z_{1:n}; \theta) - \boldsymbol{\beta}^*\|_2^2, \tag{8}$$

where $\mathbb{P}(\mathcal{P})$ is a distribution in the target problem-space $\mathcal{P}$. Let $\theta^* \in \arg\min \mathcal{L}_{train}^{\gamma=0}(\mathcal{D}_m; \theta)$ be a minimizer of the training loss. It is well-known that the generalization error can be bounded by:

$$\mathcal{L}_{gen}(\mathbb{P}(\mathcal{P}); \theta^*) \leq \underbrace{\mathcal{L}_{gen}(\mathbb{P}(\mathcal{P}); \theta^*) - \mathcal{L}_{train}^{\gamma=0}(\mathcal{D}_m; \theta^*)}_{\text{generalization gap: Theorem 4.1}} + \underbrace{\mathcal{L}_{train}^{\gamma=0}(\mathcal{D}_m; \theta^*)}_{\text{training error: Theorem 3.1}}. \tag{9}$$

We will theoretically characterize these two terms in Theorem 3.1 and Theorem 4.1.

## 3 CAPACITY OF PLISA

Can $\texttt{PLISA}_\theta$ achieve a small training error without using too many layers? How can designs of element-wise regularization and learnable penalty functions help $\texttt{PLISA}_\theta$ to achieve a smaller training error compared to classic algorithms? We answer this question theorectically in this section.

### 3.1 FIRST MAIN RESULT: CAPACITY

Let $\boldsymbol{\beta}_t(Z_{1:n}; \theta)$ be the output of the $t$-th block in $\texttt{PLISA}_\theta$. Let $\boldsymbol{x} \vee a$ denote entry-wise maximal value $\max\{\boldsymbol{x}, a\}$. Let $(\boldsymbol{x})_S$ denote the sub-vector of $\boldsymbol{x}$ with entries indexed by the set $S$.

**Theorem 3.1** (Capacity). *Assume the problem space $\mathcal{P}$ satisfies Assumption C.1 and $\mathcal{D}_m \subseteq \mathcal{P}$. Let $T$ be the number of blocks in $\texttt{PLISA}_\theta$ and let $K$ be the number of layers in each block. For any $\varepsilon > 0$, there exists a set of parameters $\theta = \{\boldsymbol{\eta}, \boldsymbol{\lambda}^*, \boldsymbol{w}, \alpha\}$ such that the estimation error of every problem $(Z_{1:n}, \boldsymbol{\beta}^*) \in \mathcal{D}_m$ is bounded as follows, $\forall T > t_0$,*

$$\|\boldsymbol{\beta}_T(Z_{1:n}; \theta) - \boldsymbol{\beta}^*\|_2 \leq \varepsilon^{-1} c_\theta s^* \exp(-C_\theta K(T - t_0)) \qquad \textit{optimization error} \quad (10)$$

$$+ c'_\theta \kappa_m \| (\nabla_{\boldsymbol{\beta}} L_n(Z_{1:n}, \boldsymbol{\beta}^*) \vee \varepsilon)_{S^*} \|_2, \qquad \textit{statistical error} \quad (11)$$

*where $S^* := \mathrm{supp}(\boldsymbol{\beta}^*)$ is the support indices of $\boldsymbol{\beta}^*$, $c_\theta, c'_\theta$, and $C_\theta$ are some positive values depending on the chosen $\theta$, and $\kappa_m$ is a condition number which reveals the similarity of the problems in $\mathcal{D}_m$. Note that $K$ and $t_0$ are required to be larger than certain values, but we will elaborate in Appendix E that the required lower bounds are small. See Appendix E for the proof of this theorem.*

This estimation error can be interpreted as a combination of the optimization error (in Eq. 10) and the statistical error (in Eq. 11). The optimization error decreases linearly in both $K$ and $T$. The statistical error occurs because of the randomness in $Z_{1:n}$. The gradient at the true parameter $\nabla_{\boldsymbol{\beta}} L_n(Z_{1:n}, \boldsymbol{\beta}^*)$ characterizes how well the finite samples $Z_{1:n}$ can represent the distribution $\mathbb{P}_{\boldsymbol{\beta}^*}$.

A direct consequence of Theorem 3.1 is that the training error can be small without using too many layers and blocks in $\texttt{PLISA}_\theta$. We will also elaborate on how the entry-wise regularization and learnable penalty function can effectively reduce the training error in the following.

*(i) Impact of entry-wise regularization.* Restricting the regularization to be uniform across entries will lead to an error bound that replaces the statistical error $\| (\nabla_{\boldsymbol{\beta}} L_n(Z_{1:n}, \boldsymbol{\beta}^*) \vee \varepsilon)_{S^*} \|_2$ in Eq. 11 by $\sqrt{s^*} (\|\nabla_{\boldsymbol{\beta}} L_n(Z_{1:n}, \boldsymbol{\beta}^*)\|_\infty \vee \varepsilon)$. To understand how the former has improved the latter, we can consider the sparse linear regression problem. If the design matrix is normalized such that $\max_{1 \leq j \leq d} \|([\boldsymbol{x}_1]_j, \cdots, [\boldsymbol{x}_n]_j)\|_2 \leq \sqrt{n}$, then $\| (\nabla_{\boldsymbol{\beta}} L_n(Z_{1:n}, \boldsymbol{\beta}^*) \vee \varepsilon)_{S^*} \|_2 \leq C\sqrt{s^*/n}$ with high probability. In comparison, $\sqrt{s^*}\|\nabla_{\boldsymbol{\beta}} L_n(Z_{1:n}, \boldsymbol{\beta}^*)\|_\infty \leq C\sqrt{s^* \log d/n}$ with high probability is a slower statistical rate due to the term $\log d$.

*(ii) Impact of learnable penalty function.* To explain the the benefit of using learnable penalty function, we give a more refined bound for the statistical error in Eq. 11 in the following lemma.

**Lemma 3.1** (Refined bound). *Assume the same conditions and parameters $\theta$ in Theorem 3.1. Assume $T \to \infty$ so that the optimization error can be ignored. For simplicity, assume $\widetilde{w_3} = 0$ and only consider the weights $\widetilde{w_1}$ and $\widetilde{w_2}$ for $\ell_1$ penalty and MCP. Then for every problem $(Z_{1:n}, \boldsymbol{\beta}^*) \in \mathcal{D}_m$:*

$$\|\boldsymbol{\beta}_\infty(Z_{1:n}; \theta) - \boldsymbol{\beta}^*\|_2 \leq \frac{1 + 8(1 + \widetilde{w_2})\kappa_m}{\rho_- - \widetilde{w_2}/b} \|(\nabla_{\boldsymbol{\beta}} L_n(Z_{1:n}, \boldsymbol{\beta}^*) \vee \varepsilon)_{S_1^*}\|_2 \quad \left(S_1^*: \textit{Small } |\beta_j^*|'s\right) \quad (12)$$

$$+ \frac{1 + 8(1 - \widetilde{w_2})\kappa_m}{\rho_- - \widetilde{w_2}/b} \|(\nabla_{\boldsymbol{\beta}} L_n(Z_{1:n}, \boldsymbol{\beta}^*) \vee \varepsilon)_{S_2^*}\|_2 \quad \left(S_2^*: \textit{Large } |\beta_j^*|'s\right), \quad (13)$$

*where $b > 1$ is a hyperparameter in MCP, and the index sets $S_1^*$ and $S_2^*$ are defined as $S_1^* := \{j \in S^* : |\beta_j^*| \leq b\lambda_j^*\}$ and $S_2^* := \{j \in S^* : |\beta_j^*| > b\lambda_j^*\}$. See Appendix E for the proof.*

This refined bound reveals the benefit of learning the penalty function because:

1. According to Lemma 3.1, the optimal penalty function is *problem-dependent*. For example, if $(8(b\rho_- + 1)\kappa_m + 1)\|(\nabla_{\boldsymbol{\beta}} L_n(Z_{1:n}, \boldsymbol{\beta}^*) \vee \varepsilon)_{S_1^*}\|_2 > (8(b\rho_- - 1)\kappa_m - 1)\|(\nabla_{\boldsymbol{\beta}} L_n(Z_{1:n}, \boldsymbol{\beta}^*) \vee \varepsilon)_{S_2^*}\|_2$, choosing $\widetilde{w_2} = 0$ can induce a smaller error bound. Otherwise, $\widetilde{w_2} = 1$ is better. Therefore, learning is a more suitable way of choosing the penalty function.
2. The convergence speed $C_\theta$ in Eq. 10 is also affected by the weights, monotonely decreasing in $\widetilde{w_2}$. Through gradient-based training, we can automatically find the optimal combination of penalty functions to strike a nice balance between the statistical error and convergence speed.

## 4 GENERALIZATION ANALYSIS

How well can the learned $\texttt{PLISA}_\theta$ solve new problems outside the training set? In this section, we conduct the generalization analysis in a novel way to focus on answering the questions:

> How is the *generalization bound* of $\texttt{PLISA}_\theta$ related to its *algorithmic properties*?
> And how is it *different from* conventional neural networks?

### 4.1 SECOND MAIN RESULT: GENERALIZATION BOUND

To analyze the generalization properties of neural networks, many works have adopted the analysis framework of Bartlett & Mendelson (2002) to bound the Rademacher complexity via Dudley's integral (Bartlett et al., 2017; Chen et al., 2019; Garg et al., 2020; Joukovsky et al., 2021). A key step in this analysis framework is deriving the robustness of the training loss to the small perturbation in model parameters $\theta$. Since we can view $\texttt{PLISA}_\theta$ as an iterative algorithm, we borrow the analysis tools of classic optimization algorithms to derive its robustness in $\theta$. The following lemma states this key intermediate result, which connects the Lipschitz constant to algorithmic properties of $\texttt{PLISA}_\theta$.

**Lemma 4.1** (Robustness to $\theta$). *Assume $\mathcal{P}$ satisfies Assumption C.1 and $\mathcal{D}_m \sim \mathbb{P}(\mathcal{P})^m$. Assume $\texttt{PLISA}_\theta$ contains $T > t_0$ blocks and $K$ layers. Consider a parameter space $\Theta$ in which the parameters satisfy (i) $\alpha \in [\alpha_{\min}, \frac{1}{\rho_+}]$, (ii) $\eta_j \in [\sigma^{-1}(0.9), \eta_{\max}]$, (iii) $\widetilde{w_2}\frac{1}{b} + \widetilde{w_3}\frac{1}{a-1} \leq \xi_{\max} < \rho_-$, and (iv) $\lambda_j^* \in [8\sup_{(Z_{1:n},\boldsymbol{\beta}^*)\in\mathcal{D}_m} |[\nabla_{\boldsymbol{\beta}} L_n(Z_{1:n}, \boldsymbol{\beta}^*)]_j| \vee \varepsilon, \lambda_{\max}]$ with some positive constants $\alpha_{\min}$, $\eta_{\max}$, $\xi_{\max}$, and $\lambda_{\max}$. Then for any $\theta = \{\boldsymbol{\eta}, \boldsymbol{\lambda}^*, \boldsymbol{w}, \alpha\}$ and $\theta' = \{\boldsymbol{\eta}', \boldsymbol{\lambda}^{*\prime}, \boldsymbol{w}', \alpha'\}$ in $\Theta$, and for any recovery problem $(Z_{1:n}, \boldsymbol{\beta}^*) \in \mathcal{D}_m$, the following inequality holds,*

$$\|\boldsymbol{\beta}_T(Z_{1:n};\theta) - \boldsymbol{\beta}_T(Z_{1:n};\theta')\|_2 \leq c_1 K(T-t_0)\sqrt{s^*} |\alpha - \alpha'| \underbrace{\exp(-C_\Theta K(T-t_0))}_{\text{convergence rate}} \quad (14)$$

$$+ \left(c_2\|\boldsymbol{\eta} - \boldsymbol{\eta}'\|_2 + c_3\|\boldsymbol{\lambda}^* - \boldsymbol{\lambda}^{*\prime}\|_2 + c_4\sqrt{d}\|\boldsymbol{w} - \boldsymbol{w}'\|_2\right) \underbrace{(1 - \exp(-C_\Theta KT))}_{\text{stability rate}}, \quad (15)$$

*where $c_1, c_2, c_3, c_4$ and $C_\Theta$ are some positive constants. Note that similar to Theorem 3.1, $K$ and $t_0$ are required to be larger than certain small values. See Appendix F.1 for the proof.*

**Convergence rate & step size perturbation.** In Eq. 14, the Lipschitz constant in the step size $\alpha$ scales at the same rate as the convergence rate of $\texttt{PLISA}_\theta$, decreasing exponentially in $T$ and $K$ (See Fig. 3 for a visualization). To understand this, consider when both step sizes $\alpha$ and $\alpha'$ are within the convergence region (i.e., $(0, \rho_+^{-1})$). After infinitely many steps, their induced outputs will both converge to the same optimal point. This intuitively explains why the output perturbation caused by $\alpha$-perturbation has the same decrease rate as the optimization error.

**Stability rate & regularization perturbation.** In the literature of optimization, stability of an algorithm expresses its robustness to small perturbation in the optimization objective. This is clearly related to the robustness of $\texttt{PLISA}_\theta$ to the perturbation in $\boldsymbol{\eta}, \boldsymbol{\lambda}^*, \boldsymbol{w}$, because these parameters jointly determine the regularization $P_{\boldsymbol{w}}(\boldsymbol{\lambda}_t, \boldsymbol{\beta})$, which is a part of the optimization objective. Therefore, we exploit the analysis techniques for algorithmic stability to derive the robustness in $(\boldsymbol{\eta}, \boldsymbol{\lambda}^*, \boldsymbol{w})$-perturbation and obtain the Lipschitz constant in Eq. 15, which is bounded but increasing in $T$ and $K$ (See Fig. 3 for a visualization).

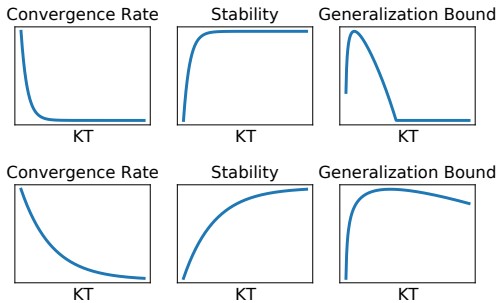

Figure 3: Visualization of convergence, stability, and generalization bound in Theorem 4.1. The two sets of visualizations are obtained by choosing different speeds $C_\Theta$ in the convergence rate and stability.

Based on the key result in Lemma 4.1, we can apply Dudley's integral to measure the empirical Rademachar complexity which immediately yields the following generalization bound.

**Theorem 4.1** (Generalization gap). *Assume the assumptions in Lemma 4.1. For any $\epsilon > 0$, with probability at least $1 - \epsilon$, the generalization gap is bounded by*

$$\mathcal{L}_{gen}(\mathbb{P}(\mathcal{P}); \theta) - \mathcal{L}_{train}^{\gamma=0}(\mathcal{D}_m; \theta) \leq c_1 \sqrt{m^{-1} \log(4\epsilon^{-1})} + \tag{16}$$

$$\sqrt{c_2 m^{-1} \log \big(\sqrt{m} KT \underbrace{\exp(-C_\Theta K(T - t_0))}_{\text{convergence rate}} \vee 1\big) + c_3 dm^{-1} \log \big(\sqrt{m}\underbrace{(1 - \exp(-C_\Theta KT))}_{\text{stability}}\big)},$$

*where $c_1, c_2, c_3, C_\Theta$ are constants independent of $d, m, K$ and $T$. See Appendix F for the proof.*

Fig. 3 visualizes how the generalization bound in Theorem 4.1 grows when $KT$ increases. The two sets of plots look slightly different by picking different constants $C_\Theta$. We have also tried varying the values of $c_2, c_3, d, m$ in Theorem 4.1. Overall, they lead to the two types of behaviors in Fig. 3.

An important observation in Theorem 4.1 and Figure 3 is that the generalization gap could *decrease in the number of layers*, and we will see in Section 7 that this matches the empirical observations. It also distinguishes algorithm-unrolling based architectures from conventional neural networks, whose generalization gaps rarely decrease in the number of layers.

*Remark.* The above generalization results are conducted on a constrained parameter space (as described in Lemma 4.1) so that we can utilize the algorithmic properties of PLISA$_\theta$. We focus on this space because the analysis contains more interesting and new ingredients. For parameters outside this space, the analysis procedure is similar to other conventional recurrent networks. Since the bound in Theorem 4.1 has matched the empirical observations, it is reasonable to believe that after training, the learned parameters are likely to be in this 'nice' constrained space.

## 5 EXTENSION TO UNSUPERVISED LEARNING-TO-LEARN SETTING

Real-world datasets may not contain the ground-truth parameters $\boldsymbol{\beta}^*$, but only contain the samples from each task, $\mathcal{D}_m^U = \{Z_{1:n}^{(1)}, \cdots, Z_{1:n}^{(m)}\}$. In this setting, we can construct an unsupervised training loss to minimize the empirical loss function $L_n$ (e.g., the likelihood function) on the samples.

**Unsupervised training loss:** $\quad \mathcal{L}_{train}^U(\mathcal{D}_m^U; \theta) := \frac{1}{m} \sum_{i=1}^{m} L_{n_2}\big(Z_{1:n_2}^{(i)}, \boldsymbol{\beta}_T(Z_{1:n_1}^{(i)}; \theta)\big) \tag{17}$

In this loss, both $Z_{1:n_1}^{(i)}$ and $Z_{1:n_2}^{(i)}$ are subsets of $Z_{1:n}^{(i)}$. The samples $Z_{1:n_1}^{(i)}$ are used as the input to PLISA$_\theta$ and the samples $Z_{1:n_2}^{(i)}$ are used for evaluating the output $\boldsymbol{\beta}_T$ from PLISA$_\theta$.

Let $\theta_U^* \in \arg\min \mathcal{L}_{train}^U(\mathcal{D}_m^U; \theta)$ be a minimizer to this unsupervised loss. **Theoretically**, to bound the generalization error of $\theta_U^*$, we can show that

$$\mathcal{L}_{gen}(\mathbb{P}(\mathcal{P}); \theta_U^*) \leq \underbrace{\frac{C}{m} \sum_{i=1}^{m} \Big(L_{n_2}\big(Z_{1:n_2}^{(i)}, \boldsymbol{\beta}_T(Z_{1:n_1}^{(i)}; \theta_U^*)\big) - L_{n_2}\big(Z_{1:n_2}^{(i)}, \boldsymbol{\beta}^{*(i)}\big)\Big)}_{\text{unsupervised training error}} \tag{18}$$

$$+ \underbrace{\mathcal{L}_{gen}(\mathbb{P}(\mathcal{P}); \theta_U^*) - \mathcal{L}_{train}^{\gamma=0}(\mathcal{D}_m; \theta_U^*)}_{\text{generalization gap: Theorem 4.1}} + \underbrace{\frac{C}{m} \sum_{i=1}^{m} \|\nabla_{\boldsymbol{\beta}} L_{n_2}(Z_{1:n_2}^{(i)}, \boldsymbol{\beta}^{*(i)})\|_2^2}_{\text{statistical error}}. \tag{19}$$

Compared to Eq. 9, this upper bound contains an additional statistical error, which appears because of the gap between the unsupervised loss and the true error that we aim to optimize. Clearly, in this upper bound, the generalization gap can be bounded by Theorem 4.1. Furthermore, the unsupervised training error in Eq. 18 can be bounded by combining Theorem 3.1 with the following in equality.

$$L_{n_2}\big(Z_{1:n_2}, \boldsymbol{\beta}_T(Z_{1:n_1}; \theta_U^*)\big) - L_{n_2}\big(Z_{1:n_2}, \boldsymbol{\beta}^*\big) \tag{20}$$

$$\leq L_{n_2}\big(Z_{1:n_2}, \boldsymbol{\beta}_T(Z_{1:n_1}; \theta^*)\big) - L_{n_2}\big(Z_{1:n_2}, \boldsymbol{\beta}^*\big)$$

$$\leq \|\nabla_{\boldsymbol{\beta}} L_{n_2}(Z_{1:n_2}, \boldsymbol{\beta}^*)\|_2 \underbrace{\|\boldsymbol{\beta}_T(Z_{1:n_1}; \theta^*) - \boldsymbol{\beta}^*\|_2}_{\text{bounded by Theorem 3.1}} + \frac{\rho_+}{2} \underbrace{\|\boldsymbol{\beta}_T(Z_{1:n_1}; \theta^*) - \boldsymbol{\beta}^*\|_2^2}_{\text{bounded by Theorem 3.1}}. \tag{21}$$

More details of the extension to unsupervised setting can be found in Appendix H.

## 6 RELATED WORK

Learning-to-learn has become an active research direction in recent years (Bora et al., 2017; Franceschi et al., 2017; Niculae et al., 2018; Denevi et al., 2018; Pogančić et al., 2019; Liu et al., 2019b; Berthet et al., 2020). Many works share the idea of unrolling or differentiating through algorithms to design the architecture (Yang et al., 2017; Borgerding et al., 2017; Corbineau et al., 2019; Xie et al., 2019; Shrivastava et al., 2020; Chen et al., 2020a; Wei et al., 2020; Indyk et al., 2019; Grover et al., 2019; Wu et al., 2019). A well-known example of learning-based algorithm is LISTA (Gregor & LeCun, 2010) which interprets ISTA (Daubechies et al., 2004) as layers of neural networks and has been an active research topic (Zhang & Ghanem, 2018; Kamilov & Mansour, 2016; Chen et al., 2018; Liu et al., 2019a; Wu et al., 2020; Kim & Park, 2020).

However, the generalization of algorithm learning has received less attention. The only exceptions are several works. However, Chen et al. (2020b) and Wang et al. (2021) only consider learning to optimize quadratic losses. Behboodi et al. (2020); Joukovsky et al. (2021) do not connect the generalization analysis with algorithmic properties to provide tighter bounds as in our work. Unlike our work that analyzes the Lipschitz continuity (Lemma 4.1), the work of Gupta & Roughgarden (2017); Balcan et al. (2021) studied the generalization of learning-based algorithms with a focus on scenarios when the Lipschitz continuity is unavailable. We will refer the audience to Shlezinger et al. (2020); Chen et al. (2021) for a more comprehensive summary of related works.

## 7 EXPERIMENTS

### 7.1 SYNTHETIC EXPERIMENTS

In synthetic datasets, we consider sparse linear regression problems and sparse precision matrix estimation problems for Gaussian graphical models. Specifically, we recover target vectors $\boldsymbol{\beta}^* \in \mathbb{R}^d$, where $d = \{256, 1024\}$ in SLR, and estimate precision matrices $\Theta^* \in \mathbb{R}^{d \times d}$, where $d = \{50, 100\}$ in SPE. See Appendix I.2 for descriptions of the dataset and data preparation.

#### 7.1.1 SPARSE LINEAR REGRESSION (SLR)

In this experiment, we compare PLISA with several baselines and also verify the theorems.

**Performance comparison.** We consider baselines including APF (Wang et al., 2014), ALISTA (Liu et al., 2019a), RNN (Andrychowicz et al., 2016), and RNN-$\ell_1$. APF is the path-following algorithm that is used as the basis of our architecture. ALISTA is a representative of algorithm unrolling based architectures, which is an advanced variant of LISTA. We have tried the vanilla LISTA, but it performs worse than ALISTA on our tasks so it is not reported. RNN

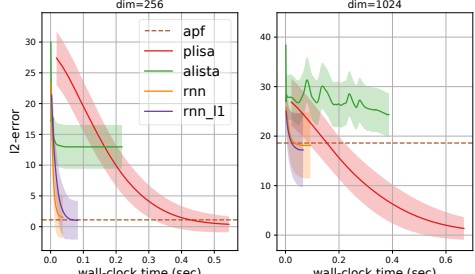

Figure 4: Convergence of recovery error. Since APF takes a long time to converge, its curve are outside the range of these plots. We use a dashline to represent the final $\ell_2$ error it achieves.

refers to the LSTM-based model in Andrychowicz et al. (2016). Besides, we add a soft-thresholding operator to this model to enforce sparsity, and include this variant as a baseline, called RNN-$\ell_1$. Except for APF, all methods are trained on the same set of training problems and selected by the validation problems. For APF, we perform grid-search to choose its hyperparameters, which is also selected by the validation set. The detailed specification of each model can be found in Appendix I.3.

Fig. 4 shows the convergence of $\|\boldsymbol{\beta}_t - \boldsymbol{\beta}^*\|_2$ for problems in the test set. The $x$-axis indicates the wall-clock time. In terms of the final recovery accuracy, PLISA outperforms all baseline methods. In the more difficult setting (i.e, $d = 1024$), its advantage is obvious. Although PLISA is slightly slower than other deep learning based models due to the computations of

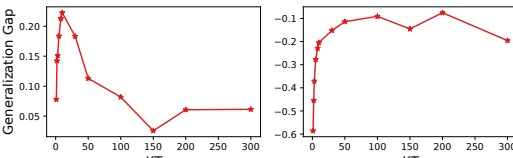

Figure 5: Generalization gap of PLISA with varying $KT$, for two different experimental settings.

MCP and SCAD, PLISA achieves a better accuracy and it has been converging much faster than the classic algorithm APF. APF is very slow mainly due to the use of line-search for selecting step sizes.

**Generalization gap.** We are interested in the generalization behavior of `PLISA`. As this experiment is conducted for theoretical interest, we do not use the validation set to select the model. We vary the number of layers ($K$) and blocks ($T$) in `PLISA` to create a set of models with different depths. For each depth, we train the model with 2000 training problems, and then test it on a separate set of 100 problems to approximate the generalization gap. In Fig. 5, we observe the interesting behavior of the generalization gap, where the left one increases in $KT$ at the beginning and then decrease to a constant, and the right one increases fast and then decrease very slowly. This surprisingly matches the two different visualizations in Fig. 3 of the predicted generalization gap given by Theorem 4.1.

### 7.1.2 SPARSE PRECISION MATRIX ESTIMATION (SPE)

We compare `PLISA` with APF, GLASSO (Friedman et al., 2008), GISTA (Guillot et al., 2012), and GGM (Belilovsky et al., 2017) on sparse precision estimation tasks in Gaussian graphical models. GLASSO estimates the precision matrix by block-coordinate decent methods. GISTA is a proximal gradient method for precision matrix estimation. GGM utilizes convolutional neural network to estimate the precision matrix. Details of each model and the training can be found in Appendix I.4.

Table 1 reports the Frobenius error $\|\Theta - \Theta^*\|_F^2$ between the estimation $\Theta$ and the true precision matrix $\Theta^*$, averaged over 100 test problems. `PLISA` achieves consistent improvements. Classic algorithm are slower because they perform linesearch. GLASSO is faster than other classic algorithm because we use the sklearn package (Pedregosa et al., 2011) in which the implementations are optimized.

Table 1: Recovery error in SPE. The reported time is the average wall-clock time for solving each instance in seconds.

| Sizes | $p = 50$ | | $p = 100$ | |
|---|---|---|---|---|
| Methods | $\|\Theta - \Theta^*\|_F^2$ | Time | $\|\Theta - \Theta^*\|_F^2$ | Time |
| PLISA | $119.47 \pm 12.23$ | 0.117 | $142.70 \pm 13.38$ | 0.132 |
| GLASSO | $169.63 \pm 17.99$ | 1.66 | $237.95 \pm 27.49$ | 3.12 |
| GISTA | $186.96 \pm 25.48$ | 53.47 | $373.66 \pm 41.72$ | 36.02 |
| APF | $269.51 \pm 32.28$ | 46.02 | $485.94 \pm 60.33$ | 86.82 |
| GGM | $194.26 \pm 10.73$ | 0.007 | $445.00 \pm 58.89$ | 0.008 |

### 7.1.3 DISCUSSION ON TRAINING-TESTING TIME

**Test time.** Fig. 4 and Table 1 shows the wall-clock time for solving test problems. Overall, classic algorithms are slower because they need to perform line-search. It is noteworthy that learning-based methods can solve a batch of problems parallelly but most classic algorithms cannot. To allow more advantages for classic algorithms, the test problems are solved without batching in all methods.

**Train time.** As metioned earlier, we perform grid-search to select the hyperparameters in classic algorithms using validation sets. The training time comparison is summarized in Table 3 and Table 4 in Appendix I.4. We can see that training time is not a bottleneck of this problem. Moreover, In SPE, classic algorithms even require a longer training time than learning-based methods.

### 7.2 UNSUPERVISED LEARNING ON REAL-WORLD DATASETS

We conduct experiments of unsupervised algorithm learning on 3 datasets: **Gene** expression dataset (Kouno et al., 2013), **Parkinsons** patient dataset (Tsanas et al., 2009), and **School** exam score dataset (Zhou et al., 2011). See Appendix I.6 for details of datasets and the configuration.

The goal of the algorithm is to estimation the sparse linear regression parameters $\beta^*$ for each problem. Recovery accuracy is estimated by the least-square error on a set of held-out samples for each problem. Table 2 shows the effectiveness of `PLISA`. Note that these real-world datasets may not satisfy the assumptions in this paper. This set of experiments are conducted only to demonstrate the robustness of the proposed method.

Table 2: Recovery accuracy on real-world datasets.

| | PLISA | APF | RNN | RNN-$\ell_1$ | ALISTA |
|---|---|---|---|---|---|
| Gene | 1.177 | 1.289 | 1.639 | 1.349 | 1.289 |
| Parkinsons | 11.63 | 11.86 | 11.91 | 13.05 | 34.843 |
| School | 296.6 | 367.9 | 561.5 | 310.3 | 884.2 |

## 8 DISCUSSION

We proposed `PLISA` for learning to solve sparse parameter recovery problems. We analyze its capacity and generalization ability. The techniques could be used to derive guarantees for other algorithm-unrolling based architectures. The model `PLISA` can be improved by using a more flexible penalty function (e.g., a conventional neural network) as long as it satisfies Assumption B.1.

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

## A  LIST OF DEFINITIONS AND NOTATIONS

For the convenience of the reader, we summarize a list of notations blow.

1. $\|\boldsymbol{\beta}\|_{\boldsymbol{\lambda},1} := \sum_{j=1}^{d} \lambda_j \, |\beta_j|$ denotes element-wise $\ell_1$ norm.

2. $(\boldsymbol{x})_S$ is the sub-vector of $\boldsymbol{x}$ with entries in the index set $S$.

3. $\boldsymbol{x} \vee \epsilon = \max\{\boldsymbol{x}, \epsilon\} = [\max\{x_1, \epsilon\}, \cdots, \max\{x_d, \epsilon\}]^\top$.

4. Element-wise soft-threshold:

$$\mathcal{T}_{s\boldsymbol{\lambda}}(\boldsymbol{\beta}) = (|\boldsymbol{\beta}| - s\boldsymbol{\lambda})_+ \circ \operatorname{sign}(\boldsymbol{\beta}) \tag{22}$$

$$= \begin{bmatrix} (|\beta_1| - s\lambda_1)_+ \operatorname{sign}(\beta_1) \\ \vdots \\ (|\beta_d| - s\lambda_d)_+ \operatorname{sign}(\beta_d) \end{bmatrix} \tag{23}$$

5. Denote a single iteration of the modified proximal gradient step as

$$\boldsymbol{\beta}^k = \mathtt{MPG}(\boldsymbol{\beta}^{k-1}; \boldsymbol{\lambda}, \boldsymbol{w}, \alpha)$$
$$:= \mathcal{T}_{\alpha \cdot \boldsymbol{\lambda}} \left( \boldsymbol{\beta}^{k-1} - \alpha \left( \nabla L_{\boldsymbol{\lambda}, \boldsymbol{w}}(\boldsymbol{\beta}^{k-1}) \right) \right)$$

## B  PENALTY FUNCTIONS

We provide detailed descriptions of the penalty functions used in `PLISA`. We focus on learning the combination of three well-known penalty functions:

$$P^{(1)}(\boldsymbol{\lambda}, \boldsymbol{\beta}) = \|\boldsymbol{\lambda} \circ \boldsymbol{\beta}\|_1, \ \ P^{(2)}(\boldsymbol{\lambda}, \boldsymbol{\beta}) = \sum_{j=1}^{p} \mathrm{MCP}(\lambda_j, \beta_j), \ \ P^{(3)}(\boldsymbol{\lambda}, \boldsymbol{\beta}) = \sum_{j=1}^{p} \mathrm{SCAD}(\lambda_j, \beta_j),$$

where MCP (Zhang, 2010a) and SCAD (Fan & Li, 2001) are nonconvex penalties whose analytical forms are given below.

### B.1  ANALYTICAL FORMS

The MCP penalty can be written as

$$\mathrm{MCP}(\lambda, \beta) = \left( \lambda \, |\beta| - \frac{\beta^2}{2b} \right) \cdot \mathbb{1}\left( |\beta| \leq b\lambda \right) + \frac{b\lambda^2}{2} \cdot \mathbb{1}\left( |\beta| > b\lambda \right),$$

where $b > 0$ is some hyperparameter.

The SCAD penalty can be written as

$$\mathrm{SCAD}(\lambda, \beta) = \lambda \, |\beta| \cdot \mathbb{1}\left( |\beta| \leq \lambda \right) - \frac{\beta^2 - 2a\lambda \, |\beta| + \lambda^2}{2(a-1)} \cdot \mathbb{1}\left( \lambda < |\beta| \leq a\lambda \right)$$
$$+ \frac{(a+1)\lambda^2}{2} \cdot \mathbb{1}\left( |\beta| > a\lambda \right),$$

where $a > 2$ is some hyperparameter.

`PLISA` uses the concave components of these penalty functions. Their analytical forms are as follows.

$$q^{(2)}(\lambda, \boldsymbol{\beta}) = \mathrm{MCP}(\lambda, \beta) - \lambda \, |\beta| = -\frac{\beta^2}{2b} \cdot \mathbb{1}\left( |\beta| \leq b\lambda \right) + \left( \frac{b\lambda^2}{2} - \lambda \, |\beta| \right) \cdot \mathbb{1}\left( |\beta| > b\lambda \right)$$

$$q^{(3)}(\lambda, \boldsymbol{\beta}) = \mathrm{SCAD}(\lambda, \beta) - \lambda \, |\beta| = \frac{2\lambda \, |\beta| - \beta^2 - \lambda^2}{2(a-1)} \cdot \mathbb{1}\left( \lambda < |\beta| \leq a\lambda \right)$$
$$+ \frac{(a+1)\lambda^2 - 2\lambda \, |\beta|}{2} \cdot \mathbb{1}\left( |\beta| > a\lambda \right)$$

### B.2 Implications

Based on the definitions of these penalty functions, it is easy to check that their convex combination satisfies a set of conditions as stated in the following Assumption B.1.

**Assumption B.1** (Regularization). *Let $P(\boldsymbol{\lambda}, \boldsymbol{\beta}) = Q(\boldsymbol{\lambda}, \boldsymbol{\beta}) + \|\boldsymbol{\beta}\|_{\boldsymbol{\lambda},1}$ where $Q$ is concave in $\boldsymbol{\beta}$, and that $Q(\boldsymbol{\lambda}, \boldsymbol{\beta}) = \sum_{j=1}^{d} q(\lambda_j, \beta_j)$. Denote the partial gradient of $q$ as $q'_\lambda(\beta) := \frac{\partial q(\lambda, \beta)}{\partial \beta}$. Assume the following regularization conditions.*

*(a) There exists constants $\xi \geq 0$ such that for any $\beta' > \beta$,*

$$-\xi \leq \frac{q'_\lambda(\beta') - q'_\lambda(\beta)}{\beta' - \beta} \leq 0.$$

*(b) $q(\lambda, -\beta) = q(\lambda, \beta)$.*

*(c) $q'_\lambda(0) = q(\lambda, 0) = 0$.*

*(d) $|q'_\lambda(\beta)| \leq \lambda$.*

*(e) $|q'_{\lambda_1}(\beta) - q'_{\lambda_2}(\beta)| \leq |\lambda_1 - \lambda_2|$.*

*Note that the penalty function $P_{\boldsymbol{w}}(\boldsymbol{\lambda}, \boldsymbol{\beta})$ in* `PLISA`$_\theta$ *satisfies all these conditions, with the constant $\xi$ in condition (a) being*

$$\xi_{\boldsymbol{w}} = \widetilde{w}_2 \frac{1}{b} + \widetilde{w}_3 \frac{1}{a-1},$$

*where $a$ and $b$ are the hyperparameters in* SCAD *and* MCP.

Next, we present an important lemma which states the restricted strongly convexity and smoothness of the modified loss

$$L_{\boldsymbol{\lambda}, \boldsymbol{w}}(\boldsymbol{\beta}) := L_n(Z_{1:n}, \boldsymbol{\beta}) + Q_{\boldsymbol{w}}(\boldsymbol{\lambda}, \boldsymbol{\beta}).$$

**Lemma B.1** (Restricted Strongly Convex and Restricted Strongly Smooth). *Assume $\mathcal{P}$ satisfies Assumption C.1 and $(Z_n, \boldsymbol{\beta}^*) \in \mathcal{P}$. If $Q(\boldsymbol{\lambda}, \boldsymbol{\beta})$ satisfies the regularization conditions in Assumption B.1 with a constant $\xi < \rho_-$. Then*

$$L_{\boldsymbol{\lambda}}(\boldsymbol{\beta}) := L_n(Z_{1:n}, \boldsymbol{\beta}) + Q(\boldsymbol{\lambda}, \boldsymbol{\beta})$$

*is strongly convex and smooth in the subspace $\{\|\boldsymbol{\beta}' - \boldsymbol{\beta}\|_0 \leq s^* + 2\tilde{s}\}$. That is, for any $\boldsymbol{\beta}, \boldsymbol{\beta}'$ such that $\|\boldsymbol{\beta}' - \boldsymbol{\beta}\|_0 \leq s^* + 2\tilde{s}$, the following inequalities hold*

$$L_{\boldsymbol{\lambda}}(\boldsymbol{\beta}') \geq L_{\boldsymbol{\lambda}}(\boldsymbol{\beta}) + \nabla L_{\boldsymbol{\lambda}}(\boldsymbol{\beta})^\top (\boldsymbol{\beta}' - \boldsymbol{\beta}) + \frac{\rho_- - \xi}{2} \|\boldsymbol{\beta}' - \boldsymbol{\beta}\|_2^2 \tag{24}$$

$$L_{\boldsymbol{\lambda}}(\boldsymbol{\beta}') \leq L_{\boldsymbol{\lambda}}(\boldsymbol{\beta}) + \nabla L_{\boldsymbol{\lambda}}(\boldsymbol{\beta})^\top (\boldsymbol{\beta}' - \boldsymbol{\beta}) + \frac{\rho_+}{2} \|\boldsymbol{\beta}' - \boldsymbol{\beta}\|_2^2 \tag{25}$$

*Proof.* The proof is straightforward by using the condition (a) in Assumption B.1 and the sparse-eigenvalue condition in Assumption C.1. $\square$

## C    PROBLEM SPACE ASSUMPTIONS

We follow the notations in Wang et al. (2014); Loh & Wainwright (2015) to describe some classic assumptions on the estimation problems.

**Assumption C.1** (Problem Space). *Let $s^*, \tilde{s}$ be positive integers and $\rho_-, \rho_+$ be positive constants such that $\tilde{s} > (121(\rho_+/\rho_-) + 144(\rho_+/\rho_-)^2)s^*$. Assume for every estimation problem $(Z_{1:n}, \boldsymbol{\beta}^*)$ in the space $\mathcal{P}$, the following conditions are satisfied.*

(a) *$\|\boldsymbol{\beta}^*\|_0 \leq s^*$ and $\|\boldsymbol{\beta}^*\|_\infty \leq B_1$;*

(b) *For any nonzero $\boldsymbol{v} \in \mathbb{R}^d$ with sparsity $\|\boldsymbol{v}\|_0 \leq s^* + 2\tilde{s}$, it holds $\frac{\boldsymbol{v}^\top \nabla_{\boldsymbol{\beta}}^2 L_n(Z_{1:n}, \boldsymbol{\beta})\boldsymbol{v}}{\|\boldsymbol{v}\|_2^2} \in [\rho_-, \rho_+]$;*

(c) *$8\,|[\nabla_{\boldsymbol{\beta}} L_n(Z_{1:n}, \boldsymbol{\beta}^*)]_j| \leq |[\nabla_{\boldsymbol{\beta}} L_n(Z_{1:n}, \mathbf{0})]_j| \leq B_2$, $\forall j = 1, \cdots, d$.*

Condition (a) assumes $\boldsymbol{\beta}^*$ is $s^*$-sparse and $B_1$-bounded. Condition (b) is commonly referred to as 'sparse eigenvalue condition' (Zhang, 2010b; Wang et al., 2014), which is weaker than the well-known restricted isometry property (RIP) in compressed sensing (Candes & Tao, 2005). Note that the class of functions satisfying conditions of this type is much larger than the class of convex losses. In the special case when $L_n(Z_{1:n}, \boldsymbol{\beta})$ is strongly convex in $\boldsymbol{\beta}$, condition (b) holds with $\tilde{s} \to \infty$. The last condition bounds the gradient of the empirical loss $L_n$ at the true parameter $\boldsymbol{\beta}^*$ and $\mathbf{0}$.

# D    RECOVERY GUARANTEE FOR PARAMETERS IN CONSTRAINED SPACE

Our generalization analysis is conducted on a constrained parameter space. In this section, we provide theoretical guarantees for the estimation error of $\texttt{PLISA}_\theta$ when $\theta$ is in the constrained space. We first formally define the parameter space as follows.

**Definition D.1** (Constrained parameter space). *Assume $\mathcal{P}$ satisfies Assumption C.1 and $\mathcal{D}_m \subseteq \mathcal{P}$. Given some positive values $\alpha_{\min}$, $\eta_{\max}$, $\xi_{\max}$, and $\lambda_{\max}$. We definite the constrained parameter space $\Theta(\mathcal{D}_m)$ as $\Theta(\mathcal{D}_m) := \{\theta = \{\boldsymbol{\eta}, \boldsymbol{\lambda}^*, \boldsymbol{w}, \alpha\} : \text{conditions (i) (ii) (iii) (iv) are satisfied}\}$, where the conditions are:*

*(i) $\alpha \in [\alpha_{\min}, \frac{1}{\rho_+}]$.*

*(ii) $\eta_j \in [\eta_{\min} := \sigma^{-1}(0.9), \eta_{\max}]$, for all $j = 1, \cdots, d$.*

*(iii) $\widetilde{w_2}\frac{1}{b} + \widetilde{w_3}\frac{1}{a-1} \leq \xi_{\max} < \rho_-$, where $b$ is the hyperparameter in MCP and $a$ is the hyperparameter in SCAD.*

*(iv) $\lambda_j^* \in [8 \sup_{(Z_{1:n}, \boldsymbol{\beta}^*) \in \mathcal{D}_m} |[\nabla_{\boldsymbol{\beta}} L_n(Z_{1:n}, \boldsymbol{\beta}^*)]_j| \vee \varepsilon, \lambda_{\max}]$, for all $j = 1, \cdots, d$.*

For parameters in this constrained space, we can show the performance guarantee for the outputs of $\texttt{PLISA}_\theta$, as stated in the following Theorem D.1.

**Theorem D.1** (Recovery guarantee). *Assume $\mathcal{P}$ satisfies Assumption C.1 and $\mathcal{D}_m \subseteq \mathcal{P}$. Let $\Theta(\mathcal{D}_m)$ be the constrained parameter space defined in Definition D.1 and assume $\theta = \{\boldsymbol{\eta}, \boldsymbol{\lambda}^*, \boldsymbol{w}, \alpha\} \in \Theta(\mathcal{D}_m)$. Assume*

$$K \geq \log\left(4\sqrt{\frac{29\rho_+}{\rho_- - \xi_{\max}}}(1 + \frac{1}{\alpha_{\min}\rho_+})\frac{\|(B_2 \vee \boldsymbol{\lambda}^*)_{S^*}\|_2}{\varepsilon}\right) / \log\left(\delta^{-1/2}\right) + 1,$$

*where $\delta := 1 - \alpha_{\min}(\rho_- - \xi_{\max})$. Then for any problem $(Z_{1:n}, \boldsymbol{\beta}^*)$ in the training set $\mathcal{D}_m$, all intermediate outputs of $\texttt{PLISA}_\theta$ have the following error bounds for all $t = 1, \cdots, T$.*

$$\|\widetilde{\boldsymbol{\beta}}_t^k(Z_{1:n}; \theta) - \boldsymbol{\beta}^*\|_2 \leq \frac{15/2}{\rho_- - \xi_{\max}}\|(\boldsymbol{\lambda}_t(Z_{1:n}; \theta))_{S^*}\|_2, \quad \forall k = 1, \cdots, K,$$

$$\|\boldsymbol{\beta}_t(Z_{1:n}; \theta) - \boldsymbol{\beta}^*\|_2 \leq \frac{19/8}{\rho_- - \xi_{\max}}\|(\boldsymbol{\lambda}_t(Z_{1:n}; \theta))_{S^*}\|_2.$$

*Proof.* The proof of this theorem is based on a series of key lemmas for the properties of the modified proximal gradient steps in each cell of $\texttt{PLISA}$. We state these key lemmas and their proofs in Section G. With these lemmas, the proof of this theorem is straightforward, and we state the proof below.

Given a fixed problem $(Z_{1:n}, \boldsymbol{\beta}^*) \in \mathcal{D}_m$ and a fixed set of parameters $\theta = \{\boldsymbol{\eta}, \boldsymbol{\lambda}^*, \boldsymbol{w}, \alpha\} \in \Theta(\mathcal{D}_m)$, we simplify some notations in this proof by removing the dependency on $Z_{1:n}$ and $\theta$. For example, we denote $\boldsymbol{\beta}_t = \boldsymbol{\beta}_t(Z_{1:n}; \theta), \widetilde{\boldsymbol{\beta}}_t^k = \widetilde{\boldsymbol{\beta}}_t^k(Z_{1:n}; \theta), \boldsymbol{\lambda}_t = \boldsymbol{\lambda}_t(Z_{1:n}; \theta)$, etc.

**Notations.** Furthermore, we will use the following simplified notations and definitions.

$$\begin{aligned}
\textit{empirical loss} \quad & L_n(\boldsymbol{\beta}) := L_n(Z_{1:n}, \boldsymbol{\beta}) \\
\textit{modified loss} \quad & L_{\boldsymbol{\lambda}}(\boldsymbol{\beta}) := L_n(Z_{1:n}, \boldsymbol{\beta}) + Q_{\boldsymbol{w}}(\boldsymbol{\lambda}, \boldsymbol{\beta}) \\
\textit{regularized loss} \quad & \phi_{\boldsymbol{\lambda}}(\boldsymbol{\beta}) := L_n(Z_{1:n}, \boldsymbol{\beta}) + P_{\boldsymbol{w}}(\boldsymbol{\lambda}, \boldsymbol{\beta}) = L_{\boldsymbol{\lambda}}(\boldsymbol{\beta}) + \|\boldsymbol{\beta}\|_{\boldsymbol{\lambda}, 1} \\
\textit{local optimal} \quad & \widehat{\boldsymbol{\beta}}_{\boldsymbol{\lambda}} \in \underset{\boldsymbol{\beta} \in \mathbb{R}^d : \|\boldsymbol{\beta}_{\overline{S^*}}\|_0 \leq \tilde{s}}{\arg\min} \phi_{\boldsymbol{\lambda}}(\boldsymbol{\beta}) \\
\textit{sub-optimality} \quad & \omega_{\boldsymbol{\lambda}}(\boldsymbol{\beta}) := \underset{\boldsymbol{\xi}' \in \partial\|\boldsymbol{\beta}\|_1}{\min} \underset{\boldsymbol{\beta}'}{\max}\left\{\frac{(\boldsymbol{\beta} - \boldsymbol{\beta}')^\top}{\|\boldsymbol{\beta} - \boldsymbol{\beta}'\|_{\boldsymbol{\lambda}, 1}}(\nabla L_{\boldsymbol{\lambda}}(\boldsymbol{\beta}) + \boldsymbol{\lambda} \circ \boldsymbol{\xi}')\right\}.
\end{aligned}$$

Now we are ready to state the proof. We first show the following statement holds true for all $t \leq T$ by mathematical induction.

$$\textbf{Statement}(t): \quad \omega_{\boldsymbol{\lambda}_t}(\widetilde{\boldsymbol{\beta}}_t^0) \leq \frac{1}{2}, \quad \text{and} \quad \|(\widetilde{\boldsymbol{\beta}}_t^k)_{\overline{S^*}}\|_0 \leq \tilde{s}, \quad \forall k = 1, \cdots, K.$$

**Statement(1).** First we verify that Statement(1) holds true. Recall that $\boldsymbol{\lambda}_0 = |\nabla L_n(\mathbf{0})|$. In the following, we prove that $\widetilde{\boldsymbol{\beta}}_1^0$ is a local solution of $\phi_{\boldsymbol{\lambda}_0}(\boldsymbol{\beta}) := L_{\boldsymbol{\lambda}_0}(\boldsymbol{\beta}) + \|\boldsymbol{\beta}\|_{\boldsymbol{\lambda}_0, 1}$.

$$
\begin{aligned}
& \nabla L_{\boldsymbol{\lambda}_0}(\widetilde{\boldsymbol{\beta}}_1^0) + \boldsymbol{\lambda}_0 \circ \partial \|\widetilde{\boldsymbol{\beta}}_1^0\|_1 \\
&= \nabla L_n(\mathbf{0}) + \nabla_{\boldsymbol{\beta}} Q_{\boldsymbol{w}}(\boldsymbol{\lambda}_0, \mathbf{0}) + \boldsymbol{\lambda}_0 \circ \partial \|\mathbf{0}\|_1 \\
&= \operatorname{sign}(\nabla L_n(\mathbf{0})) \circ |\nabla L_n(\mathbf{0})| + \mathbf{0} + \boldsymbol{\lambda}_0 \circ \partial \|\mathbf{0}\|_1 \\
&= \operatorname{sign}(\nabla L_n(\mathbf{0})) \circ \boldsymbol{\lambda}_0 + \boldsymbol{\lambda}_0 \circ \partial \|\mathbf{0}\|_1.
\end{aligned}
$$

Since $-\operatorname{sign}(\nabla L_n(\mathbf{0})) \in \partial\|\mathbf{0}\|_1$, then we have $\mathbf{0} \in \nabla L_{\boldsymbol{\lambda}_0}(\widetilde{\boldsymbol{\beta}}_1^0) + \boldsymbol{\lambda}_0 \circ \partial\|\widetilde{\boldsymbol{\beta}}_1^0\|_1$. Therefore, $\omega_{\boldsymbol{\lambda}_0}(\widetilde{\boldsymbol{\beta}}_1^0) = 0$. Since $\boldsymbol{\lambda}_1 = \boldsymbol{\lambda}_0 \circ \boldsymbol{\eta}_1$, by Lemma G.6, we have

$$
\omega_{\boldsymbol{\lambda}_1}(\widetilde{\boldsymbol{\beta}}_1^0) \leq \frac{\omega_{\boldsymbol{\lambda}_0}(\widetilde{\boldsymbol{\beta}}_1^0) + 0.2}{0.9} = \frac{2}{9} \leq 1/2.
$$

By Lemma G.5, we have $\phi_{\boldsymbol{\lambda}_1}(\widetilde{\boldsymbol{\beta}}_1^0) - \phi_{\boldsymbol{\lambda}_1}(\boldsymbol{\beta}^*) \leq \frac{21/2\|(\boldsymbol{\lambda}_1)_{S^*}\|_2^2}{\rho_- - \xi_{\max}}$, which implies that the conditions in Lemma G.4 are satisfied. Therefore, we have proved that

$$
\|(\boldsymbol{\beta}_1^k)_{\overline{S^*}}\|_0 \leq \tilde{s} \quad \forall k = 1, \ldots, K.
$$

This finishes the proof of Statement(1).

**Statement(t).** Now we assume Statement($t-1$) is true and prove Statement($t$). First, we prove that $\omega_{\boldsymbol{\lambda}_{t-1}}(\widetilde{\boldsymbol{\beta}}_{t-1}^K) \leq 1/4$. By Lemma G.2 and Lemma G.5,

$$
\omega_{\boldsymbol{\lambda}_{t-1}}(\widetilde{\boldsymbol{\beta}}_{t-1}^K) \leq \frac{\sqrt{2}(1 + \frac{1}{\alpha_{\min}\rho_+})\sqrt{\rho_+}}{\min(\boldsymbol{\lambda}_{t-1})} \sqrt{(1 - \alpha_{\min}(\rho_- - \xi_{\max}))^{K-1} \frac{29/2\|(\boldsymbol{\lambda}_{t-1})_{S^*}\|_2^2}{\rho_- - \xi_{\max}}}
$$

$$
\leq \sqrt{\frac{29\rho_+}{\rho_- - \xi_{\max}}} (1 + \frac{1}{\alpha_{\min}\rho_+}) \frac{\|(\boldsymbol{\lambda}_0 \vee \boldsymbol{\lambda}^*)_{S^*}\|_2}{\varepsilon} \sqrt{(1 - \alpha_{\min}(\rho_- - \xi_{\max}))}^{K-1}.
$$

Assume $K \geq \log\left(4\sqrt{\frac{29\rho_+}{\rho_- - \xi_{\max}}}(1 + \frac{1}{\alpha_{\min}\rho_+}) \frac{\|(\boldsymbol{\lambda}_0 \vee \boldsymbol{\lambda}^*)_{S^*}\|_2}{\varepsilon}\right) / \log\left(\delta^{-1/2}\right) + 1$, where

$$
\delta := 1 - \alpha_{\min}(\rho_- - \xi_{\max}).
$$

Then we have

$$
\omega_{\boldsymbol{\lambda}_{t-1}}(\widetilde{\boldsymbol{\beta}}_{t-1}^K) \leq 1/4. \tag{26}
$$

Since $\widetilde{\boldsymbol{\beta}}_t^0 = \widetilde{\boldsymbol{\beta}}_{t-1}^K$, Lemma G.6 implies that

$$
\omega_{\boldsymbol{\lambda}_t}(\widetilde{\boldsymbol{\beta}}_t^0) \leq 1/2.
$$

Furthermore, by Lemma G.5, $\omega_{\boldsymbol{\lambda}_t}(\widetilde{\boldsymbol{\beta}}_t^0) \leq 1/2$ implies

$$
\phi_{\boldsymbol{\lambda}_t}(\widetilde{\boldsymbol{\beta}}_t^0) - \phi_{\boldsymbol{\lambda}_t}(\boldsymbol{\beta}^*) \leq \frac{21/2\|(\boldsymbol{\lambda}_t)_{S^*}\|_2^2}{\rho_- - \xi_{\max}}, \tag{27}
$$

which in turns implies that the conditions in Lemma G.4 are satisfied. Therefore, we have

$$
\|(\widetilde{\boldsymbol{\beta}}_t^k)_{\overline{S^*}}\|_0 \leq \tilde{s} \quad \forall k = 1, \ldots, K,
$$

which completes the proof of Statement($t$).

Now we derive the error bounds. Similar to how we derive Eq. 26 and Eq. 27, it is easy to show that

$$
\omega_{\boldsymbol{\lambda}_t}(\widetilde{\boldsymbol{\beta}}_t^K) \leq 1/4, \quad \forall t = 1, \cdots, T,
$$

$$
\text{and } \phi_{\boldsymbol{\lambda}_t}(\widetilde{\boldsymbol{\beta}}_t^0) - \phi_{\boldsymbol{\lambda}_t}(\boldsymbol{\beta}^*) \leq \frac{21/2\|(\boldsymbol{\lambda}_t)_{S^*}\|_2^2}{\rho_- - \xi_{\max}} \quad \forall t = 1, \cdots, T.
$$

Combining $\phi_{\boldsymbol{\lambda}_t}(\widetilde{\boldsymbol{\beta}}_t^0) - \phi_{\boldsymbol{\lambda}_t}(\boldsymbol{\beta}^*) \leq \frac{21/2\|(\boldsymbol{\lambda}_t)_{S^*}\|_2^2}{\rho_- - \xi_{\max}}$ with Lemma G.3, we have

$$
\|\widetilde{\boldsymbol{\beta}}_t^k - \boldsymbol{\beta}^*\|_2 \leq \frac{15/2\|(\boldsymbol{\lambda}_t)_{S^*}\|_2^2}{\rho_- - \xi_{\max}}, \quad \forall k = 1, \cdots, K. \tag{28}
$$

Combining $\omega_{\boldsymbol{\lambda}_t}(\widetilde{\boldsymbol{\beta}}_t^K) \leq 1/4$ with Lemma G.5, we have

$$
\|\widetilde{\boldsymbol{\beta}}_t^K - \boldsymbol{\beta}^*\|_2 \leq \left(\frac{1}{4} + \frac{17}{8}\right) \frac{\|(\boldsymbol{\lambda}_t)_{S^*}\|_2^2}{\rho_- - \xi_{\max}} = \frac{19/8\|(\boldsymbol{\lambda}_t)_{S^*}\|_2^2}{\rho_- - \xi_{\max}}.
$$

$\square$

# E  CAPACITY ANALYSIS: PROOF OF THEOREM 3.1

*Proof.* This theorem is a direct result of Theorem D.1, by taking a suitable set of parameters $\theta$. Similar to the proof of Theorem D.1, some notations are simplified. Please refer to the proof of Theorem D.1 for detailed illustrations.

Denote the union support set of the true parameters $\boldsymbol{\beta}^*$ by $S_m^* := \{j : j \in \text{supp}(\boldsymbol{\beta}^*), (Z_{1:n}, \boldsymbol{\beta}^*) \in \mathcal{D}_m\}$. Then we specify a set of parameters $\theta = \{\boldsymbol{\eta}, \boldsymbol{\lambda}^*, \boldsymbol{w}, \alpha\}$ for $\texttt{PLISA}_\theta$ as follows.

- $\alpha = \frac{1}{\rho_+}$.
- $\boldsymbol{w}$: The weights $\boldsymbol{w}$ can take any values as long as they satisfy $\widetilde{w}_2 \frac{1}{b} + \widetilde{w}_3 \frac{1}{a-1} < \rho_-$. Since $\widetilde{w}_2$ and $\widetilde{w}_3$ and be arbitrary close to 0, there must exists a set of weights $\boldsymbol{w}$ that satisfy this constraint. In particular, we denote this value by $\xi_{\boldsymbol{w}} := \widetilde{w}_2 \frac{1}{b} + \widetilde{w}_3 \frac{1}{a-1}$.
- $\boldsymbol{\lambda}^*$: For each $j \in S_m^*$, we take $\lambda_j^* := 8 \max_{(Z_n, \boldsymbol{\beta}^*) \in \mathcal{D}_m} |\nabla L_n(Z_n, \boldsymbol{\beta}^*)_j| \vee \varepsilon$ as the target regularization parameter. For each $j \notin S_m^*$, we take $\lambda_j^* = B_2$, which is the upper bound of $\|\nabla_{\boldsymbol{\beta}} L_n(Z_{1:n}, \mathbf{0})\|_\infty$ by Assumption C.1.
- $\boldsymbol{\eta}$: For all $j$, take $\eta_j = \log 9$ so that the decrease ratio is $\sigma(\eta_j) = 0.9$.

Clearly, this set of parameters satisfy the conditions in Theorem D.1, so we can apply its result in this proof.

In the following, we will show that, with this specification of the parameters, $\texttt{PLISA}_\theta$ can achieve the recovery accuracy stated in Theorem 3.1.

Let $t_0 = \min \left\{ t : t \geq 0, \boldsymbol{\lambda}^* = \max\{0.9^t |\nabla_{\boldsymbol{\beta}} L_n(Z_{1:n}, \mathbf{0})|, \boldsymbol{\lambda}^*\}, Z_{1:n} \in \mathcal{D}_m, \boldsymbol{\lambda}^* \in \theta \in \Theta(\mathcal{D}_M) \right\}$ be the number of blocks after which $\boldsymbol{\lambda}_t(Z_{1:n}; \theta) = \boldsymbol{\lambda}^*$. We know $t_0$ is a small number because the decrease rate is linear. To be more clear, it is easy to show that $t_0 \leq \log\left(\frac{B_2}{\varepsilon}\right) / \log(10/9)$. Therefore, after $t_0$ cells, the regularization parameters do not change anymore. That is,

$$\boldsymbol{\lambda}_t(Z_{1:n}; \theta) = \boldsymbol{\lambda}^*, \quad \forall t \geq t_0.$$

Since the specified parameters satisfy the conditions in Theorem D.1, we can follow the same way as how we derive Eq. 26 to obtain that

$$\omega_{\boldsymbol{\lambda}_{t-1}}(\boldsymbol{\beta}_{t-1}) \leq 1/4, \quad \forall t = 1, \cdots, T.$$

Since we have $\boldsymbol{\lambda}_t = \boldsymbol{\lambda}^*$ for $t \geq t_0$. The last $(T - t_0)$ cells can be viewed as a single cell with $K(T - t_0)$ many steps. Therefore, we can apply Lemma G.2 and Lemma G.5 to obtain

$$\omega_{\boldsymbol{\lambda}_t}(\widetilde{\boldsymbol{\beta}}_t^K) \leq \frac{2\sqrt{2\rho_+}}{\epsilon} \sqrt{(1 - \alpha(\rho_- - \xi_{\boldsymbol{w}}))^{K(t-t_0)-1} \frac{29/2\|(\boldsymbol{\lambda}_{t-1})_{S^*}\|_2^2}{\rho_- - \xi_{\boldsymbol{w}}}}$$

$$\leq \sqrt{\frac{29\rho_+}{\rho_- - \xi_{\boldsymbol{w}}}} (1 + \frac{1}{\alpha\rho_+}) \frac{\|(\boldsymbol{\lambda}_0 \vee \boldsymbol{\lambda}^*)_{S^*}\|_2}{\varepsilon} \sqrt{(1 - \alpha(\rho_- - \xi_{\boldsymbol{w}}))}^{K(t-t_0)-1}$$

$$\leq \sqrt{\frac{29\rho_+}{\rho_- - \xi_{\boldsymbol{w}}}} (1 + \frac{1}{\alpha\rho_+}) \frac{\sqrt{s^*} B_2}{\varepsilon} \sqrt{(1 - \alpha(\rho_- - \xi_{\boldsymbol{w}}))}^{K(t-t_0)-1}$$

$$= c_\theta \sqrt{s^*} \varepsilon^{-1} \exp(-C_\theta K(t - t_0)),$$

where $C_\theta = -\frac{1}{2} \log(1 - \alpha(\rho_- - \xi_{\boldsymbol{w}}))$, $c_\theta = \sqrt{\frac{29\rho_+}{\rho_- - \xi_{\boldsymbol{w}}}} (1 + \frac{1}{\alpha\rho_+}) \frac{B_2}{1 - \alpha(\rho_- - \xi_{\boldsymbol{w}})}$. By Lemma G.5,

$$\|\boldsymbol{\beta}_t - \boldsymbol{\beta}^*\|_2 \leq \frac{\left(\omega_{\boldsymbol{\lambda}_t}(\widetilde{\boldsymbol{\beta}}_t^K) + \frac{17}{8}\right) \|\boldsymbol{\lambda}_{S^*}^*\|_2}{\rho_- - \xi_{\boldsymbol{w}}}$$

$$\leq \frac{\|\boldsymbol{\lambda}_{S^*}^*\|_2}{\rho_- - \xi_{\boldsymbol{w}}} c_\theta \varepsilon^{-1} \sqrt{s^*} \exp(-C_\theta K(t - t_0)) + \frac{\frac{17}{8}\|\boldsymbol{\lambda}_{S^*}^*\|_2}{\rho_- - \xi_{\boldsymbol{w}}}$$

$$\leq \tilde{c}_\theta \varepsilon^{-1} s^* \exp(-C_\theta K(t - t_0)) + c_\theta' \|\boldsymbol{\lambda}_{S^*}^*\|_2 \tag{29}$$

where $\tilde{c}_\theta = \frac{\sqrt{s^*} B_2}{\rho_- - \xi_{\boldsymbol{w}}} c_\theta$ and $c_\theta' = \frac{17}{8(\rho_- - \xi_{\boldsymbol{w}})}$. What remains is bounding $\|\boldsymbol{\lambda}_{S^*}^*\|_2$. First, we define a condition number as

$$\kappa_m = \max_{j \in S_m^*} \left\{ \frac{8 \max_{(Z_n, \boldsymbol{\beta}^*) \in \mathcal{D}_m} |\nabla_{\boldsymbol{\beta}} L_n(Z_n, \boldsymbol{\beta}^*)_j|}{8 \min_{(Z_n, \boldsymbol{\beta}^*) \in \mathcal{D}_m} |\nabla_{\boldsymbol{\beta}} L_n(Z_n, \boldsymbol{\beta}^*)_j| \vee \varepsilon} \right\}.$$

In fact, $\kappa_m \leq \frac{B_2}{\varepsilon}$. Then recall the specification of $\boldsymbol{\lambda}^*$ at the beginning of this proof. We have

$$
\begin{aligned}
\|\boldsymbol{\lambda}_{S^*}^*\|_2 &\leq \sum_{j \in S^*} 8\kappa_m \left|\nabla_{\boldsymbol{\beta}} L_n(Z_n, \boldsymbol{\beta}^*)_j\right| \vee \varepsilon \\
&\leq 8\kappa_m \|(\nabla_{\boldsymbol{\beta}} L_n(Z_n, \boldsymbol{\beta}^*) \vee \varepsilon)_{S^*}\|_2.
\end{aligned}
$$

Plugging this back to Eq. 29, we have

$$
\|\boldsymbol{\beta}_t - \boldsymbol{\beta}^*\|_2 \leq \tilde{c}_\theta \varepsilon^{-1} s^* \exp(-C_\theta K(t - t_0)) + c_\theta' 8\kappa_m \|(\nabla_{\boldsymbol{\beta}} L_n(Z_n, \boldsymbol{\beta}^*) \vee \varepsilon)_{S^*}\|_2.
$$

This is equivalent to the statement to be proved. $\square$

# F   GENERALIZATION ANALYSIS: PROOF OF THEOREM 4.1

We first state a well-known inequality that bounds the generalization gap by *empirical Rademacher complexity*.

**Theorem F.1** (Adapted from Theorem 26.5 (Shalev-Shwartz & Ben-David, 2014)). *Assume that for all $(Z_{1:n}, \boldsymbol{\beta}^*) \in \mathcal{P}$ and all $\theta \in \Theta$ we have that $\ell(\boldsymbol{\beta}_T(Z_{1:n}; \theta), \boldsymbol{\beta}^*) \leq c$. Then with probability at least $1 - \epsilon$, for all $\theta \in \Theta$,*

$$\mathcal{L}_{gen}(\mathbb{P}(\mathcal{P}); \theta) - \mathcal{L}_{train}^{\gamma=0}(\mathcal{D}_m; \theta) \leq 2R_m(\ell_\Theta) + 4c\sqrt{\frac{2\log(4\epsilon^{-1})}{m}},$$

*where $\ell_\Theta := \{(Z_{1:n}, \boldsymbol{\beta}^*) \mapsto \|\boldsymbol{\beta}_T(Z_{1:n}; \theta) - \boldsymbol{\beta}^*\|_2^2 : \theta \in \Theta\}$ is the space of loss functions of our model $\texttt{PLISA}_\theta$, and $R_m(\ell_\Theta)$ is its empirical Rademacher complexity. It is defined as*

$$R_m(\ell_\Theta) := \mathbb{E}_{\boldsymbol{\sigma}} \sup_{\theta \in \Theta} \frac{1}{m} \sum_{i=1}^m \sigma_i \left\| \boldsymbol{\beta}_T(Z_{1:n_i}^{(i)}; \theta) - \boldsymbol{\beta}^{*(i)} \right\|_2^2,$$

*where $\{\sigma_i\}_{i=1}^m$ are $m$ independent Rademacher random variables.*

Therefore, it resorts to bound the empirical Rademacher complexity, which can be bounded via Dudley's integral. The following theorem states the bound for the empirical Rademacher complexity and its proof follows.

**Theorem F.2** (Empirical Rademacher complexity bound). *Assume the assumptions in Theorem 4.1. Let $\Theta = \Theta(\mathcal{D}_m)$ be the constrained space defined in Definition D.1. Let $\ell_\Theta := \{(Z_{1:n}, \boldsymbol{\beta}^*) \mapsto \|\boldsymbol{\beta}_T(Z_{1:n}; \theta) - \boldsymbol{\beta}^*\|_2^2 : \theta \in \Theta\}$ be the space of loss functions of our model $\texttt{PLISA}_\theta$. Then the empirical Rademacher complexity is bounded by $R_m \ell_\Theta \leq$*

$$\frac{c_1}{\sqrt{m}}\sqrt{\log\left(c_2\sqrt{m}K(T-t_0)\exp(-C_\Theta K(T-t_0)) \vee 1\right) + c_3 d\log\left(c_4\sqrt{m}\left(1 - \exp(-C_\Theta KT)\right)\right)}$$

*where $c_1$, $c_2$, $c_3$, $c_4$ are some constants independent of $d, m, K$ and $T$.*

*Proof.* The classical Dudley's entropy integral bound gives us an upper bound for the empirical Rademacher complexity in terms of the covering number.

$$
\begin{aligned}
R_m \ell_\Theta &\leq \inf_{\alpha > 0} \left( 4\alpha + \frac{12}{\sqrt{m}} \int_\alpha^{\|\ell_\Theta\|_{P_m,\infty}} \sqrt{\log \mathcal{N}\left(\epsilon, \ell_\Theta, L_2(P_m)\right)}\, d\epsilon \right) \\
&\leq \frac{4}{\sqrt{m}} + \frac{12}{\sqrt{m}} \int_{\frac{1}{\sqrt{m}}}^{\|\ell_\Theta\|_{P_m,\infty}} \sqrt{\log \mathcal{N}\left(\epsilon, \ell_\Theta, L_2(P_m)\right)}\, d\epsilon \\
&\leq \frac{4}{\sqrt{m}} + \frac{12\|\ell_\Theta\|_{P_m,\infty}}{\sqrt{m}} \sqrt{\log \mathcal{N}\left(\frac{1}{\sqrt{m}}, \ell_\Theta, L_2(P_m)\right)}.
\end{aligned}
$$

The notation $P_m$ means the empirical measure defined based on the samples in $\mathcal{D}_m$, and

$$
\begin{aligned}
\|\ell_\Theta\|_{P_m,\infty} &:= \sup_{\theta \in \Theta} P_m(\ell_\theta) \\
&= \sup_{\theta \in \Theta} \frac{1}{m} \sum_{(Z_{1:n}, \boldsymbol{\beta}^*) \in \mathcal{D}_m} \|\boldsymbol{\beta}_T(Z_{1:n}; \theta) - \boldsymbol{\beta}^*\|_2^2 \\
\text{by Theorem D.1} \quad &\leq \sup_{\theta \in \Theta} \frac{1}{m} \sum_{(Z_{1:n}, \boldsymbol{\beta}^*) \in \mathcal{D}_m} \frac{19/8}{\rho_- - \xi_{\max}} \|(\boldsymbol{\lambda}_t(Z_{1:n}; \theta))_{S^*}\|_2 \\
&\leq \sup_{\theta \in \Theta} \frac{1}{m} \sum_{(Z_{1:n}, \boldsymbol{\beta}^*) \in \mathcal{D}_m} \frac{19/8}{\rho_- - \xi_{\max}} \sqrt{s^*} \max\{B_2, \lambda_{\max}\} \leq C.
\end{aligned}
$$

Here we use a constant $C$ as the upper bound because we only care about the scales in $d, m, K$ and $T$. Next, we bound the covering number $\mathcal{N}\left(\frac{1}{\sqrt{m}}, \ell_\Theta, L_2(P_m)\right)$. By the key result in Lemma 4.1, it

is easy to derive that

$$\|\ell_\theta - \ell_{\theta'}\|_{L_2(P_m)} \leq 2C \Big( c_1 K(T - t_0)\sqrt{s^*} |\alpha - \alpha'| \exp(-C_\Theta K(T - t_0))$$

$$+ \Big( c_2 \|\boldsymbol{\eta} - \boldsymbol{\eta}'\|_2 + c_3 \|\boldsymbol{\lambda}^* - \boldsymbol{\lambda}^{*\prime}\|_2 + c_4 \sqrt{d} \|\boldsymbol{w} - \boldsymbol{w}'\|_2 \Big) (1 - \exp(-C_\Theta KT)) \Big)$$

$$\leq cK(T - t_0) |\alpha - \alpha'| \exp(-C_\Theta K(T - t_0))$$

$$+ c \Big( \|\boldsymbol{\eta} - \boldsymbol{\eta}'\|_2 + \|\boldsymbol{\lambda}^* - \boldsymbol{\lambda}^{*\prime}\|_2 + \sqrt{d} \|\boldsymbol{w} - \boldsymbol{w}'\|_2 \Big) (1 - \exp(-C_\Theta KT)) /$$

From now on, we abuse the symbol $c$ to generally represent some constant that does not depend on $d, m, K, T$.

$$\mathcal{N} (\epsilon, \ell_\Theta, L_2(P_m)) \leq \mathcal{N} \left( \frac{\epsilon}{cK(T - t_0) \exp(-C_\Theta K(T - t_0))}, \left[ \alpha_{\min}, \frac{1}{\rho_+} \right], |\cdot| \right)$$

$$\times \mathcal{N} \left( \frac{\epsilon}{c (1 - \exp(-C_\Theta KT))}, [\eta_{\min}, \eta_{\max}]^d, \ell_2 \right)$$

$$\times \mathcal{N} \left( \frac{\epsilon}{c (1 - \exp(-C_\Theta KT))}, \prod_{j=1}^d \times [\lambda_{j,\min}, \lambda_{j,\max}], \ell_2 \right)$$

$$\times \mathcal{N} \left( \frac{\epsilon}{c\sqrt{d} (1 - \exp(-C_\Theta KT))}, [w_{\min}, w_{\max}]^3, \ell_2 \right).$$

For the covering number of a $d$-dimensional compact vector space $T \subseteq \mathbb{R}^d$, we can bound it by the $\epsilon$-packing number. That is, $\mathcal{N}(T, \epsilon, \ell_2) \leq \mathcal{M}(T, \epsilon, \ell_2) \leq c \prod_{j=1}^d \left( \frac{T_{j,\max} - T_{j,\min}}{\epsilon} + 1 \right)$ for some constant $c$. Applying this fact, we have

$$\mathcal{N} (\epsilon, \ell_\Theta, L_2(P_m)) \leq \left( \frac{cK(T - t_0) \exp(-C_\Theta K(T - t_0))}{\epsilon} \vee 1 \right)$$

$$\times \prod_{j=1}^d \left( \frac{c (1 - \exp(-C_\Theta KT))}{\epsilon} + 1 \right)$$

$$\times \prod_{j=1}^d \left( \frac{c (1 - \exp(-C_\Theta KT))}{\epsilon} + 1 \right)$$

$$\times \left( \frac{3c\sqrt{d} (1 - \exp(-C_\Theta KT))}{\epsilon} \right)^3$$

which implies

$$\log \mathcal{N} \left( \frac{1}{\sqrt{m}}, \ell_\Theta, L_2(P_m) \right)$$

$$\leq \log \left( c\sqrt{m} K(T - t_0) \exp(-C_\Theta K(T - t_0)) \vee 1 \right)$$

$$+ (2d) \log \left( c\sqrt{m} (1 - \exp(-C_\Theta KT)) + 1 \right) + 3 \log \left( c\sqrt{dm} (1 - \exp(-C_\Theta KT)) \right)$$

$$\leq \log \left( c\sqrt{m} K(T - t_0) \exp(-C_\Theta K(T - t_0)) \vee 1 \right) + cd \log \left( c\sqrt{m} (1 - \exp(-C_\Theta KT)) + 1 \right).$$

Therefore, $R_m \ell_\Theta \leq$

$$\frac{c}{\sqrt{m}} \sqrt{\log \left( c\sqrt{m} K(T - t_0) \exp(-C_\Theta K(T - t_0)) \vee 1 \right) + cd \log \left( c\sqrt{m} (1 - \exp(-C_\Theta KT)) \right)}.$$

$\square$

## F.1 ROBUSTNESS TO PARAMETER PERTURBATION: PROOF OF LEMMA 4.1

We split the parameters $\theta = \{\boldsymbol{\lambda}^*, \boldsymbol{w}, \boldsymbol{\eta}, \alpha\}$ into two sets $\theta = \theta_1 \cup \theta_2$, where $\theta_1 = \{\alpha\}$ and $\theta_2 = \{\boldsymbol{\lambda}^*, \boldsymbol{w}, \boldsymbol{\eta}\}$. Clearly, the robustness can be bounded by

$$\|\boldsymbol{\beta}_T(Z_{1:n}; \theta_1 \cup \theta_2) - \boldsymbol{\beta}_T(Z_{1:n}; \theta_1' \cup \theta_2)\|_2 + \|\boldsymbol{\beta}_T(Z_{1:n}; \theta_1' \cup \theta_2) - \boldsymbol{\beta}_T(Z_{1:n}; \theta_1' \cup \theta_2')\|_2.$$

We derive the upper bounds of these two terms in Lemma F.1 and Lemma F.2, which imply

$$\|\boldsymbol{\beta}_T(Z_{1:n}; \theta) - \boldsymbol{\beta}_T(Z_{1:n}; \theta')\|_2 \leq C\sqrt{s^*}K(T - t_0)\, \delta^{K(T-t_0)}|\alpha - \alpha'|$$
$$+ \left(C_1\|\boldsymbol{\eta} - \boldsymbol{\eta}'\|_2 + C_2\|\boldsymbol{\lambda}^* - \boldsymbol{\lambda}^{*'}\|_2 + C_4\sqrt{d}\|\boldsymbol{w} - \boldsymbol{w}'\|_2\right)\frac{1 - \delta^{KT}}{1 - \delta},$$

where $\delta := \sqrt{1 - \alpha_{\min}(\rho_- - \xi_{\max})} < 1$. Lemma 4.1 is equivalent to this inequality by taking $C_\Theta = -\log(\delta) > 0$. Therefore, the key derivation steps are in the proof of Lemma F.1 and Lemma F.2, as stated below.

**Lemma F.1.** *Assume the assumptions in Lemma 4.1 are satisfied. Let*

$$\theta = \{\boldsymbol{\lambda}^*, \boldsymbol{w}, \boldsymbol{\eta}, \alpha\} \quad and \quad \theta' = \{\boldsymbol{\lambda}^*, \boldsymbol{w}, \boldsymbol{\eta}, \alpha'\}$$

*be parameters in the constrained space $\Theta$. Let $\{\boldsymbol{\beta}_t\}_{t=1}^T = \mathtt{PLISA}_\theta(Z_{1:n})$ and $\{\boldsymbol{\beta}_t'\}_{t=1}^T = \mathtt{PLISA}_{\theta'}(Z_{1:n})$ be the intermediate outputs of $\mathtt{PLISA}$ with different parameters $\theta$ and $\theta'$. Assume*

$$K \geq \log\left(4\sqrt{\frac{29\rho_+}{\rho_- - \xi_{\max}}}(1 + \frac{1}{\alpha_{\min}\rho_+})\frac{\sqrt{s^*}(B_2 + \lambda_{\max})}{\varepsilon}\right)/\log\left(\delta^{-1}\right) + 1,$$

*where $\delta := \sqrt{1 - \alpha_{\min}(\rho_- - \xi_{\max})} < 1$. Then*

$$\|\boldsymbol{\beta}_T - \boldsymbol{\beta}_T'\|_2 \leq C\sqrt{s^*}K(T - t_0)\,\delta^{K(T-t_0)}|\alpha - \alpha'|.$$

*Proof.* By triangular inequality,

$$\left\|\widetilde{\boldsymbol{\beta}}_t^k - \widetilde{\boldsymbol{\beta}}_t^{k'}\right\|_2 = \left\|\mathtt{MPG}(\widetilde{\boldsymbol{\beta}}_t^{k-1}; \boldsymbol{\lambda}_t, \boldsymbol{w}, \alpha) - \mathtt{MPG}(\widetilde{\boldsymbol{\beta}}_t^{k-1'}; \boldsymbol{\lambda}_t, \boldsymbol{w}, \alpha')\right\|_2$$
$$\leq \left\|\mathtt{MPG}(\widetilde{\boldsymbol{\beta}}_t^{k-1}; \boldsymbol{\lambda}_t, \boldsymbol{w}, \alpha) - \mathtt{MPG}(\widetilde{\boldsymbol{\beta}}_t^{k-1}; \boldsymbol{\lambda}_t, \boldsymbol{w}, \alpha')\right\|_2$$
$$+ \left\|\mathtt{MPG}(\widetilde{\boldsymbol{\beta}}_t^{k-1}; \boldsymbol{\lambda}_t, \boldsymbol{w}, \alpha') - \mathtt{MPG}(\widetilde{\boldsymbol{\beta}}_t^{k-1'}; \boldsymbol{\lambda}_t, \boldsymbol{w}, \alpha')\right\|_2$$

WLOG, assume $\alpha > \alpha'$. Applying Lemma F.3 and Lemma F.4 to the above two terms on the right hand side, we obtain

$$\left\|\widetilde{\boldsymbol{\beta}}_t^k - \widetilde{\boldsymbol{\beta}}_t^{k'}\right\|_2 \leq \delta\|\widetilde{\boldsymbol{\beta}}_t^{k-1} - \widetilde{\boldsymbol{\beta}}_t^{k-1'}\|_2 + \sqrt{\frac{|\alpha - \alpha'|}{\alpha + \alpha'}}\left\|\widetilde{\boldsymbol{\beta}}_t^k - \widetilde{\boldsymbol{\beta}}_t^{k-1}\right\|_2$$
$$\leq \delta\|\widetilde{\boldsymbol{\beta}}_t^{k-1} - \widetilde{\boldsymbol{\beta}}_t^{k-1'}\|_2 + c\,|\alpha - \alpha'|\left\|\widetilde{\boldsymbol{\beta}}_t^k - \widetilde{\boldsymbol{\beta}}_t^{k-1}\right\|_2 \tag{30}$$

where $c$ is the Lipschtiz constant. $c$ is a finite number because $\alpha$ is bounded and positive. Applying it recursively, we have

$$\left\|\widetilde{\boldsymbol{\beta}}_t^K - \widetilde{\boldsymbol{\beta}}_t^{K'}\right\|_2 \leq \delta^K\|\widetilde{\boldsymbol{\beta}}_t^0 - \widetilde{\boldsymbol{\beta}}_t^{0'}\|_2 + c\,|\alpha - \alpha'|\sum_{k=1}^K \delta^{K-k}\left\|\widetilde{\boldsymbol{\beta}}_t^k - \widetilde{\boldsymbol{\beta}}_t^{k-1}\right\|_2. \tag{31}$$

Now we bound the term $\left\|\widetilde{\boldsymbol{\beta}}_t^k - \widetilde{\boldsymbol{\beta}}_t^{k-1}\right\|_2$. By Lemma G.1 and Lemma G.2,

$$\left\|\widetilde{\boldsymbol{\beta}}_t^k - \widetilde{\boldsymbol{\beta}}_t^{k-1}\right\|_2 \leq \sqrt{\frac{2\alpha_{\min}}{2 - \alpha_{\min}\rho_+}\left(\phi_{\boldsymbol{\lambda}_t,\boldsymbol{w}}(\widetilde{\boldsymbol{\beta}}_t^{k-1}) - \phi_{\boldsymbol{\lambda}_t,\boldsymbol{w}}(\widehat{\boldsymbol{\beta}}_{\boldsymbol{\lambda}_t})\right)}$$

$$\leq \sqrt{\frac{2}{\rho_+}\left(\phi_{\boldsymbol{\lambda}_t,\boldsymbol{w}}(\widetilde{\boldsymbol{\beta}}_t^{0}) - \phi_{\boldsymbol{\lambda}_t,\boldsymbol{w}}(\widehat{\boldsymbol{\beta}}_{\boldsymbol{\lambda}_t})\right)}\delta^{k-1}$$

$$\text{(By Lemma G.6)} \quad \leq \sqrt{\frac{2}{\rho_+}}\sqrt{\frac{10}{\rho_- - \xi_{\max}}}\|(\boldsymbol{\lambda}_t)_{S^*}\|_2\delta^{k-1}$$

$$\leq \sqrt{\frac{2}{\rho_+}}\sqrt{\frac{10}{\rho_- - \xi_{\max}}}B_2\sqrt{s^*}\delta^{k-1}$$

$$= c'\sqrt{s^*}\delta^k,$$

where $c' := \sqrt{\frac{2}{\rho_+}}\sqrt{\frac{10}{\rho_- - \xi_{\max}}}B_2\delta^{-1}$. However, we can have a tighter bound for this term if $t$ is larger than a certain integer. More precisely, recall the definition that $\boldsymbol{\lambda}_t = \max\{\boldsymbol{\lambda}_0 \circ \sigma(\boldsymbol{\eta})^t, \boldsymbol{\lambda}^*\}$. Since $\sigma(\eta_{\max}) < 1$, the entries of $\boldsymbol{\lambda}_0 \circ \sigma(\boldsymbol{\eta})^t$ will decrease exponentially to reach the entry values of $\boldsymbol{\lambda}^*$. Let $t_0 = \min\{t : t \geq 0, \boldsymbol{\lambda}^* = \max\{\boldsymbol{\lambda}_0 \circ \sigma(\eta_{\max})^t, \boldsymbol{\lambda}^*\}, \forall \boldsymbol{\lambda}^* \in \theta \in \Theta(\mathcal{D}_M)\}$ be the number of blocks after which $\boldsymbol{\lambda}_t = \boldsymbol{\lambda}^*$. We know $t_0$ is a small number because the decrease rate is linear.

Now, the regularization parameters satisfy $\boldsymbol{\lambda}_t = \boldsymbol{\lambda}^*, \forall t \geq t_0$. Therefore, we can apply Lemma G.1 and Lemma G.2 again to obtain the following bound for all $t \geq t_0$:

$$\left\|\widetilde{\boldsymbol{\beta}}_t^k - \widetilde{\boldsymbol{\beta}}_t^{k-1}\right\|_2 \leq \sqrt{\frac{2\alpha_{\min}}{2 - \alpha_{\min}\rho_+}\left(\phi_{\boldsymbol{\lambda}^*,\boldsymbol{w}}(\widetilde{\boldsymbol{\beta}}_t^{k-1}) - \phi_{\boldsymbol{\lambda}^*,\boldsymbol{w}}(\widehat{\boldsymbol{\beta}}_{\boldsymbol{\lambda}^*})\right)}$$

$$\leq \sqrt{\frac{2}{\rho_+}\left(\phi_{\boldsymbol{\lambda}^*,\boldsymbol{w}}(\widetilde{\boldsymbol{\beta}}_{t_0}^{k-1}) - \phi_{\boldsymbol{\lambda}^*,\boldsymbol{w}}(\widehat{\boldsymbol{\beta}}_{\boldsymbol{\lambda}^*})\right)}\delta^{K(t-t_0)+k-1}$$

$$\text{(By Lemma G.6)} \quad \leq \sqrt{\frac{2}{\rho_+}}\sqrt{\frac{10}{\rho_- - \xi_{\max}}}\|(\boldsymbol{\lambda}^*)_{S^*}\|_2\delta^{K(t-t_0)+k-1}$$

$$= c'\sqrt{s^*}\delta^{K(t-t_0)+k}.$$

Combining the two different bounds for $\left\|\widetilde{\boldsymbol{\beta}}_t^k - \widetilde{\boldsymbol{\beta}}_t^{k-1}\right\|_2$ with Eq. 31, we have

$$\|\boldsymbol{\beta}_t - \boldsymbol{\beta}_t'\|_2 \leq \delta^K\|\boldsymbol{\beta}_{t-1} - \boldsymbol{\beta}_{t-1}'\|_2 + \begin{cases} C|\alpha - \alpha'|\sqrt{s^*}\sum_{k=1}^K \delta^{K-k}\delta^k & \text{if } t < t_0 \\ C|\alpha - \alpha'|\sqrt{s^*}\sum_{k=1}^K \delta^{K-k}\delta^{K(t-t_0)+k} & \text{if } t \geq t_0 \end{cases}$$

$$= \delta^K\|\boldsymbol{\beta}_{t-1} - \boldsymbol{\beta}_{t-1}'\|_2 + \begin{cases} C|\alpha - \alpha'|\sqrt{s^*}K\delta^K & \text{if } t < t_0 \\ C|\alpha - \alpha'|\sqrt{s^*}K\delta^{K(t-t_0+1)} & \text{if } t \geq t_0 \end{cases}$$

where $C = cc'$. We apply the above inequality recursively and obtain that

$$\|\boldsymbol{\beta}_T - \boldsymbol{\beta}_T'\|_2 \leq C\sqrt{s^*}K\delta^K\left(\sum_{t=1}^{t_0-1}\delta^{K(T-t)}|\alpha - \alpha'| + \sum_{t=t_0}^{T}\delta^{K(T-t)}\delta^{K(t-t_0)}|\alpha - \alpha'|\right)$$

$$= C\sqrt{s^*}K\delta^K\left(\delta^{K(T-t_0+1)}\frac{1 - \delta^{K(t_0-1)}}{1 - \delta^K} + (T - t_0)\delta^{K(T-t_0)}\right)|\alpha - \alpha'|$$

$$= C\sqrt{s^*}K\delta^K\left(\left(\frac{\delta^K(1 - \delta^{K(t_0-1)})}{1 - \delta^K} + (T - t_0)\right)\delta^{K(T-t_0)}\right)|\alpha - \alpha'|$$

$$\leq C\sqrt{s^*}K(T - t_0 + 1)\delta^{K(T-t_0+1)}|\alpha - \alpha'|$$

$$\leq C\sqrt{s^*}K(T - t_0)\delta^{K(T-t_0)}|\alpha - \alpha'|$$

$\square$

**Lemma F.2.** *Assume the assumptions in Lemma 4.1 are satisfied. Let*

$$\theta = \{\boldsymbol{\lambda}^*, \boldsymbol{w}, \boldsymbol{\eta}, \alpha\} \quad and \quad \theta' = \{\boldsymbol{\lambda}^{*\prime}, \boldsymbol{w}', \boldsymbol{\eta}', \alpha\}$$

*be parameters in the constrained space* $\Theta$. *Let* $\{\boldsymbol{\beta}_t\}_{t=1}^T = \texttt{PLISA}_\theta(Z_{1:n})$ *and* $\{\boldsymbol{\beta}_t{}'\}_{t=1}^T = \texttt{PLISA}_{\theta'}(Z_{1:n})$ *be the intermediate outputs of* PLISA *with different parameters* $\theta$ *and* $\theta'$. *Then* $\|\boldsymbol{\beta}_T(Z_{1:n}; \theta) - \boldsymbol{\beta}_T(Z_{1:n}, \theta')\|_2 \leq$

$$\left(C_1\|\boldsymbol{\eta} - \boldsymbol{\eta}'\|_2 + C_2\|\boldsymbol{\lambda}^* - \boldsymbol{\lambda}^{*\prime}\|_2 + C_4\sqrt{d}\|\boldsymbol{w} - \boldsymbol{w}'\|_2\right) \frac{1 - \delta^{KT}}{1 - \delta},$$

*where* $\delta := \sqrt{1 - \alpha_{\min}(\rho_- - \xi_{\max})} < 1$ *and* $C_{1,2,4}$ *are some constants.*

*Proof.* Let $\{\boldsymbol{\beta}_t\}_{t=1}^T \cup \{\widetilde{\boldsymbol{\beta}}_t^k\}_{t=1}^T{}_{k=1}^K$ and $\{\boldsymbol{\beta}_t{}'\}_{t=1}^T \cup \{\widetilde{\boldsymbol{\beta}}_t^{k\prime}\}_{t=1}^T{}_{k=1}^K$ be the intermediate outputs of $\texttt{PLISA}_\theta(Z_{1:n})$ and $\texttt{PLISA}_{\theta'}(Z_{1:n})$ respectively. By triangle inequality,

$$\begin{aligned}
\left\|\widetilde{\boldsymbol{\beta}}_t^k - \widetilde{\boldsymbol{\beta}}_t^{k\prime}\right\|_2 &= \left\|\texttt{MPG}(\widetilde{\boldsymbol{\beta}}_t^{k-1}; \boldsymbol{\lambda}_t, \boldsymbol{w}, \alpha) - \texttt{MPG}(\widetilde{\boldsymbol{\beta}}_t^{k-1\prime}; \boldsymbol{\lambda}_t{}', \boldsymbol{w}', \alpha')\right\|_2 \\
&\leq \left\|\texttt{MPG}(\widetilde{\boldsymbol{\beta}}_t^{k-1}; \boldsymbol{\lambda}_t, \boldsymbol{w}, \alpha) - \texttt{MPG}(\widetilde{\boldsymbol{\beta}}_t^{k-1\prime}; \boldsymbol{\lambda}_t, \boldsymbol{w}, \alpha)\right\|_2 \\
&\quad + \left\|\texttt{MPG}(\widetilde{\boldsymbol{\beta}}_t^{k-1\prime}; \boldsymbol{\lambda}_t, \boldsymbol{w}, \alpha) - \texttt{MPG}(\widetilde{\boldsymbol{\beta}}_t^{k-1\prime}; \boldsymbol{\lambda}_t{}', \boldsymbol{w}, \alpha)\right\|_2 \\
&\quad + \left\|\texttt{MPG}(\widetilde{\boldsymbol{\beta}}_t^{k-1\prime}; \boldsymbol{\lambda}_t{}', \boldsymbol{w}, \alpha) - \texttt{MPG}(\widetilde{\boldsymbol{\beta}}_t^{k-1\prime}; \boldsymbol{\lambda}_t{}', \boldsymbol{w}', \alpha)\right\|_2.
\end{aligned}$$

Applying Lemma F.3, Lemma F.5, and Lemma F.6 to the above 3 terms on the right hand side, we obtain

$$\left\|\widetilde{\boldsymbol{\beta}}_t^k - \widetilde{\boldsymbol{\beta}}_t^{k\prime}\right\|_2 \leq \delta\|\widetilde{\boldsymbol{\beta}}_t^{k-1} - \widetilde{\boldsymbol{\beta}}_t^{k-1\prime}\|_2 + \underbrace{\frac{2}{\rho_+}\|\boldsymbol{\lambda}_t - \boldsymbol{\lambda}_t'\|_2}_{(i)} + \underbrace{C\|\boldsymbol{w} - \boldsymbol{w}'\|_2\|\boldsymbol{\lambda}_t\|_2}_{(ii)}, \qquad (32)$$

where $C$ is some absolute constant.

Term $(i)$. By definition and triangular inequality,

$$\begin{aligned}
\|\boldsymbol{\lambda}_t - \boldsymbol{\lambda}_t{}'\|_2 &\leq \|\max\{\boldsymbol{\lambda}_0 \circ \sigma(\boldsymbol{\eta})^t, \boldsymbol{\lambda}^*\} - \max\{\boldsymbol{\lambda}_0 \circ \sigma(\boldsymbol{\eta}')^t, \boldsymbol{\lambda}^*\}\|_2 \\
&\quad + \|\max\{\boldsymbol{\lambda}_0 \circ \sigma(\boldsymbol{\eta}')^t, \boldsymbol{\lambda}^*\} - \max\{\boldsymbol{\lambda}_0 \circ \sigma(\boldsymbol{\eta}')^t, \boldsymbol{\lambda}^{*\prime}\}\|_2 \\
&\leq \|\boldsymbol{\lambda}_0\|_\infty t\sigma(\eta_{\max})^{t-1}t\|\sigma(\boldsymbol{\eta}) - \sigma(\boldsymbol{\eta}')\|_2 + \|\boldsymbol{\lambda}^* - \boldsymbol{\lambda}^{*\prime}\|_2 \\
&\leq \frac{B_2}{4}\sigma(\eta_{\max})^t t\|\boldsymbol{\eta} - \boldsymbol{\eta}'\|_2 + \|\boldsymbol{\lambda}^* - \boldsymbol{\lambda}^{*\prime}\|_2
\end{aligned}$$

The last inequality holds by Assumption C.1 and the bounded first derivative of $\sigma(\cdot)$. Therefore, for some constants $C_2 = \frac{2}{\rho_+}$ and $\tilde{C}_1 = C_2 \frac{B_2}{4}$,

$$\begin{aligned}
(i) = \frac{2}{\rho_+}\|\boldsymbol{\lambda}_t - \boldsymbol{\lambda}_t'\|_2 &\leq \tilde{C}_1 \sigma(\eta_{\max})^t t\|\boldsymbol{\eta} - \boldsymbol{\eta}'\|_2 + C_2\|\boldsymbol{\lambda}^* - \boldsymbol{\lambda}^{*\prime}\|_2 \\
&\leq C_1\|\boldsymbol{\eta} - \boldsymbol{\eta}'\|_2 + C_2\|\boldsymbol{\lambda}^* - \boldsymbol{\lambda}^{*\prime}\|_2.
\end{aligned}$$

The last inequality holds because the value $\sigma(\eta_{\max})^t t$ is bounded by some constant.

Term $(ii)$. Since $\|\boldsymbol{\lambda}_t\|_2 \leq \|\max\{\sigma(\eta_{\max})^t\boldsymbol{\lambda}_0, \boldsymbol{\lambda}^{*\prime}\}\|_2 \leq \|\max\{\sigma(\eta_{\max})^t\boldsymbol{\lambda}_0, \boldsymbol{\lambda}_0\}\|_2 = \|\boldsymbol{\lambda}_0\|_2 \leq B\sqrt{d}$,

$$(ii) = C\|\boldsymbol{w} - \boldsymbol{w}'\|_2\|\boldsymbol{\lambda}_t\|_2 \leq C_4\sqrt{d}\|\boldsymbol{w} - \boldsymbol{w}'\|_2.$$

Plugging the bounds for (i)-(iii) into Eq. 32, we have

$$\left\|\widetilde{\boldsymbol{\beta}}_t^k - \widetilde{\boldsymbol{\beta}}_t^{k\prime}\right\|_2 \leq \delta\|\widetilde{\boldsymbol{\beta}}_t^{k-1} - \widetilde{\boldsymbol{\beta}}_t^{k-1\prime}\|_2 + C_1\|\boldsymbol{\eta} - \boldsymbol{\eta}'\|_2 + C_2\|\boldsymbol{\lambda}^* - \boldsymbol{\lambda}^{*\prime}\|_2 + C_4\sqrt{d}\|\boldsymbol{w} - \boldsymbol{w}'\|_2.$$

By direct computation,

$$\left\|\widetilde{\boldsymbol{\beta}}_t^K - \widetilde{\boldsymbol{\beta}}_t^{K'}\right\|_2$$

$$\leq \delta^K \left\|\widetilde{\boldsymbol{\beta}}_t^0 - \widetilde{\boldsymbol{\beta}}_t^{0'}\right\|_2 + \left(C_1\|\boldsymbol{\eta} - \boldsymbol{\eta}'\|_2 + C_2\|\boldsymbol{\lambda}^* - \boldsymbol{\lambda}^{*'}\|_2 + C_4\sqrt{d}\|\boldsymbol{w} - \boldsymbol{w}'\|_2\right)\sum_{k=0}^{K-1}\delta^k$$

$$= \delta^K \left\|\widetilde{\boldsymbol{\beta}}_t^0 - \widetilde{\boldsymbol{\beta}}_t^{0'}\right\|_2 + \left(C_1\|\boldsymbol{\eta} - \boldsymbol{\eta}'\|_2 + C_2\|\boldsymbol{\lambda}^* - \boldsymbol{\lambda}^{*'}\|_2 + C_4\sqrt{d}\|\boldsymbol{w} - \boldsymbol{w}'\|_2\right)\frac{1-\delta^K}{1-\delta}.$$

Recall that $\widetilde{\boldsymbol{\beta}}_t^K = \boldsymbol{\beta}_t$ and $\widetilde{\boldsymbol{\beta}}_t^0 = \boldsymbol{\beta}_{t-1}$. We can apply the above inequality recursively and obtain that

$$\|\boldsymbol{\beta}_T - \boldsymbol{\beta}_T{'}\|_2$$

$$\leq \delta^K \|\boldsymbol{\beta}_{T-1} - \boldsymbol{\beta}_{T-1}{'}\|_2 + \left(C_1\|\boldsymbol{\eta} - \boldsymbol{\eta}'\|_2 + C_2\|\boldsymbol{\lambda}^* - \boldsymbol{\lambda}^{*'}\|_2 + C_4\sqrt{d}\|\boldsymbol{w} - \boldsymbol{w}'\|_2\right)\frac{1-\delta^K}{1-\delta}$$

$$\leq \left(C_1\|\boldsymbol{\eta} - \boldsymbol{\eta}'\|_2 + C_2\|\boldsymbol{\lambda}^* - \boldsymbol{\lambda}^{*'}\|_2 + C_4\sqrt{d}\|\boldsymbol{w} - \boldsymbol{w}'\|_2\right)\frac{1-\delta^K}{1-\delta}\sum_{t=0}^{T-1}(\delta^K)^t$$

$$= \left(C_1\|\boldsymbol{\eta} - \boldsymbol{\eta}'\|_2 + C_2\|\boldsymbol{\lambda}^* - \boldsymbol{\lambda}^{*'}\|_2 + C_4\sqrt{d}\|\boldsymbol{w} - \boldsymbol{w}'\|_2\right)\frac{1-\delta^{KT}}{1-\delta}.$$

$\square$

**Lemma F.3** (Robustness to $\boldsymbol{\beta}$). *Assume* $(Z_{1:n}, \boldsymbol{\beta}^*) \in \mathcal{P}$ *and* $\mathcal{P}$ *satisfies Assumption C.1. Assume* $\boldsymbol{\beta}, \boldsymbol{\beta}'$ *satisfy* $\|\boldsymbol{\beta} - \boldsymbol{\beta}'\|_0 \leq s = s^* + 2\tilde{s}$ *and* $\alpha \leq \frac{1}{\rho_+}$. *Then*

$$\|\text{MPG}(\boldsymbol{\beta}; \boldsymbol{\lambda}, \boldsymbol{w}, \alpha) - \text{MPG}(\boldsymbol{\beta}'; \boldsymbol{\lambda}, \boldsymbol{w}, \alpha)\|_2 \leq \sqrt{1 - 2\alpha\frac{(\rho_- - \xi_{\boldsymbol{w}})\rho_+}{\rho_- - \xi_{\boldsymbol{w}} + \rho_+}}\|\boldsymbol{\beta} - \boldsymbol{\beta}'\|_2 \qquad (33)$$

$$\leq \sqrt{1 - \alpha(\rho_- - \xi_{\boldsymbol{w}})}\|\boldsymbol{\beta} - \boldsymbol{\beta}'\|_2 \qquad (34)$$

*Proof.*

$$\|\text{MPG}(\boldsymbol{\beta}; \boldsymbol{\lambda}, \boldsymbol{w}, \alpha) - \text{MPG}(\boldsymbol{\beta}'; \boldsymbol{\lambda}, \boldsymbol{w}, \alpha)\|_2^2$$

$$= \|\mathcal{T}_{\alpha\cdot\boldsymbol{\lambda}}\left(\boldsymbol{\beta} - \alpha\left(\nabla L_{\boldsymbol{\lambda},\boldsymbol{w}}(\boldsymbol{\beta})\right)\right) - \mathcal{T}_{\alpha\cdot\boldsymbol{\lambda}}\left(\boldsymbol{\beta}' - \alpha\left(\nabla L_{\boldsymbol{\lambda},\boldsymbol{w}}(\boldsymbol{\beta}')\right)\right)\|_2^2$$

$$\leq \|\boldsymbol{\beta} - \alpha\left(\nabla L_{\boldsymbol{\lambda},\boldsymbol{w}}(\boldsymbol{\beta})\right) - \boldsymbol{\beta}' + \alpha\left(\nabla L_{\boldsymbol{\lambda},\boldsymbol{w}}(\boldsymbol{\beta}')\right)\|_2^2$$

$$= \|\boldsymbol{\beta} - \boldsymbol{\beta}'\|_2^2 + \alpha^2\|\nabla L_{\boldsymbol{\lambda},\boldsymbol{w}}(\boldsymbol{\beta}) - \nabla L_{\boldsymbol{\lambda},\boldsymbol{w}}(\boldsymbol{\beta}')\|_2^2 - 2\alpha\langle\boldsymbol{\beta} - \boldsymbol{\beta}', \nabla L_{\boldsymbol{\lambda},\boldsymbol{w}}(\boldsymbol{\beta}) - \nabla L_{\boldsymbol{\lambda},\boldsymbol{w}}(\boldsymbol{\beta}')\rangle.$$

$$(35)$$

Since on the restricted subspace $\{\boldsymbol{\beta} : \|\boldsymbol{\beta} - \boldsymbol{\beta}^*\|_0 \leq s^* + 2\tilde{s}\}$, $L_{\boldsymbol{\lambda}}$ is $(\rho_- - \xi_{\boldsymbol{w}})$-strongly convex and $\rho_+$-smooth, by Lemma G.7,

$$\langle\boldsymbol{\beta} - \boldsymbol{\beta}', \nabla L_{\boldsymbol{\lambda},\boldsymbol{w}}(\boldsymbol{\beta}) - \nabla L_{\boldsymbol{\lambda},\boldsymbol{w}}(\boldsymbol{\beta}')\rangle$$

$$\geq \frac{(\rho_- - \xi_{\boldsymbol{w}})\rho_+}{\rho_- - \xi_{\boldsymbol{w}} + \rho_+}\|\boldsymbol{\beta} - \boldsymbol{\beta}'\|_2^2 + \frac{\|\nabla L_{\boldsymbol{\lambda},\boldsymbol{w}}(\boldsymbol{\beta}) - \nabla L_{\boldsymbol{\lambda},\boldsymbol{w}}(\boldsymbol{\beta}')\|_2^2}{\rho_- - \xi_{\boldsymbol{w}} + \rho_+}.$$

Combining this inequality with Eq. 35, we have

$$\|\text{MPG}(\boldsymbol{\beta}; \boldsymbol{\lambda}, \boldsymbol{w}, \alpha) - \text{MPG}(\boldsymbol{\beta}'; \boldsymbol{\lambda}, \boldsymbol{w}, \alpha)\|_2^2$$

$$\leq \left(1 - 2\alpha\frac{(\rho_- - \xi_{\boldsymbol{w}})\rho_+}{\rho_- - \xi_{\boldsymbol{w}} + \rho_+}\right)\|\boldsymbol{\beta} - \boldsymbol{\beta}'\|_2^2$$

$$+ \alpha\left(\alpha - \frac{2}{\rho_- - \xi_{\boldsymbol{w}} + \rho_+}\right)\|\nabla L_{\boldsymbol{\lambda},\boldsymbol{w}}(\boldsymbol{\beta}) - \nabla L_{\boldsymbol{\lambda},\boldsymbol{w}}(\boldsymbol{\beta}')\|_2^2$$

By the assumption that $\alpha \leq \frac{1}{\rho_+}$, the second term in the above inequality is non-positive. Furthermore, this assumption also implies that $1 - 2\alpha\frac{(\rho_- - \xi_{\boldsymbol{w}})\rho_+}{\rho_- - \xi_{\boldsymbol{w}} + \rho_+} \geq 0$. Therefore,

$$\|\text{MPG}(\boldsymbol{\beta}; \boldsymbol{\lambda}, \boldsymbol{w}, \alpha) - \text{MPG}(\boldsymbol{\beta}'; \boldsymbol{\lambda}, \boldsymbol{w}, \alpha)\|_2 \leq \sqrt{1 - 2\alpha\frac{(\rho_- - \xi_{\boldsymbol{w}})\rho_+}{\rho_- - \xi_{\boldsymbol{w}} + \rho_+}}\|\boldsymbol{\beta} - \boldsymbol{\beta}'\|_2$$

$\square$

**Lemma F.4** (Robustness to $\alpha$). *With out loss of generalization, assume $\alpha \geq \alpha'$. Denote*

$$\boldsymbol{\beta}^+ := \mathrm{MPG}(\boldsymbol{\beta}; \boldsymbol{\lambda}, \boldsymbol{w}, \alpha) \quad and \quad \boldsymbol{\beta}^{+\prime} := \mathrm{MPG}(\boldsymbol{\beta}; \boldsymbol{\lambda}, \boldsymbol{w}, \alpha').$$

*Then*

$$\|\boldsymbol{\beta}^{+\prime} - \boldsymbol{\beta}^+\|_2 \leq \sqrt{\frac{\alpha - \alpha'}{\alpha + \alpha'}} \|\boldsymbol{\beta}^+ - \boldsymbol{\beta}\|_2$$

*Proof.* Define the quadratic approximation function as

$$\psi_{\alpha,\boldsymbol{\lambda},\boldsymbol{w}}(\boldsymbol{z}, \boldsymbol{\beta}) := L_{\boldsymbol{\lambda},\boldsymbol{w}}(\boldsymbol{\beta}) + \langle \nabla L_{\boldsymbol{\lambda},\boldsymbol{w}}(\boldsymbol{\beta}), \boldsymbol{z} - \boldsymbol{\beta} \rangle + \frac{1}{2\alpha} \|\boldsymbol{z} - \boldsymbol{\beta}\|_2^2 + \|\boldsymbol{z}\|_{\boldsymbol{\lambda},1}. \tag{36}$$

Clearly,

$$\mathrm{MPG}(\boldsymbol{\beta}; \boldsymbol{\lambda}, \boldsymbol{w}, \alpha) = \arg\min_{\boldsymbol{z}} \psi_{\alpha,\boldsymbol{\lambda},\boldsymbol{w}}(\boldsymbol{z}, \boldsymbol{\beta}).$$

Since $\psi_{\alpha,\boldsymbol{\lambda},\boldsymbol{w}}(\boldsymbol{z}, \boldsymbol{\beta})$ is $\frac{1}{\alpha}$-strongly convex in $\boldsymbol{z}$, and that $\boldsymbol{\beta}^+ := \mathrm{MPG}(\boldsymbol{\beta}; \boldsymbol{\lambda}, \boldsymbol{w}, \alpha)$ is its optimal point,

$$\psi_{\alpha,\boldsymbol{\lambda},\boldsymbol{w}}(\boldsymbol{z}, \boldsymbol{\beta}) \geq \psi_{\alpha,\boldsymbol{\lambda},\boldsymbol{w}}(\boldsymbol{\beta}^+, \boldsymbol{\beta}) + \frac{1}{2\alpha} \|\boldsymbol{z} - \boldsymbol{\beta}^+\|_2^2 \quad \forall \boldsymbol{z}. \tag{37}$$

Similarly,

$$\psi_{\alpha',\boldsymbol{\lambda},\boldsymbol{w}}(\boldsymbol{z}, \boldsymbol{\beta}) \geq \psi_{\alpha',\boldsymbol{\lambda},\boldsymbol{w}}(\boldsymbol{\beta}^{+\prime}, \boldsymbol{\beta}) + \frac{1}{2\alpha'} \|\boldsymbol{z} - \boldsymbol{\beta}^{+\prime}\|_2^2 \quad \forall \boldsymbol{z}. \tag{38}$$

Taking $\boldsymbol{z} = \boldsymbol{\beta}^{+\prime}$ in Eq. 37 and $\boldsymbol{z} = \boldsymbol{\beta}^+$ in Eq. 38 yields

$$\frac{1}{2\alpha} \|\boldsymbol{\beta}^+ - \boldsymbol{\beta}^{+\prime}\|_2^2 \leq \psi_{\alpha,\boldsymbol{\lambda},\boldsymbol{w}}(\boldsymbol{\beta}^{+\prime}, \boldsymbol{\beta}) - \psi_{\alpha,\boldsymbol{\lambda},\boldsymbol{w}}(\boldsymbol{\beta}^+, \boldsymbol{\beta}),$$

$$\frac{1}{2\alpha'} \|\boldsymbol{\beta}^+ - \boldsymbol{\beta}^{+\prime}\|_2^2 \leq \psi_{\alpha',\boldsymbol{\lambda},\boldsymbol{w}}(\boldsymbol{\beta}^+, \boldsymbol{\beta}) - \psi_{\alpha',\boldsymbol{\lambda},\boldsymbol{w}}(\boldsymbol{\beta}^{+\prime}, \boldsymbol{\beta}).$$

Summing the above two inequalities, we have

$$\frac{1}{2}\left(\frac{1}{\alpha} + \frac{1}{\alpha'}\right) \|\boldsymbol{\beta}^+ - \boldsymbol{\beta}^{+\prime}\|_2^2 \leq \left(\psi_{\alpha,\boldsymbol{\lambda},\boldsymbol{w}}(\boldsymbol{\beta}^{+\prime}, \boldsymbol{\beta}) - \psi_{\alpha',\boldsymbol{\lambda},\boldsymbol{w}}(\boldsymbol{\beta}^{+\prime}, \boldsymbol{\beta})\right)$$

$$+ \left(\psi_{\alpha',\boldsymbol{\lambda},\boldsymbol{w}}(\boldsymbol{\beta}^+, \boldsymbol{\beta}) - \psi_{\alpha,\boldsymbol{\lambda},\boldsymbol{w}}(\boldsymbol{\beta}^+, \boldsymbol{\beta})\right)$$

$$(\text{by } \alpha \geq \alpha') \quad \leq \psi_{\alpha',\boldsymbol{\lambda},\boldsymbol{w}}(\boldsymbol{\beta}^+, \boldsymbol{\beta}) - \psi_{\alpha,\boldsymbol{\lambda},\boldsymbol{w}}(\boldsymbol{\beta}^+, \boldsymbol{\beta})$$

$$= \frac{1}{2}\left(\frac{1}{\alpha'} - \frac{1}{\alpha}\right) \|\boldsymbol{\beta}^+ - \boldsymbol{\beta}\|_2^2.$$

$$\square$$

**Lemma F.5** (Robustness to $\boldsymbol{\lambda}$). *Let $\boldsymbol{\beta}^+ = \mathrm{MPG}(\boldsymbol{\beta}; \boldsymbol{\lambda}, \boldsymbol{w}, \alpha)$ and $\boldsymbol{\beta}^{+\prime} = \mathrm{MPG}(\boldsymbol{\beta}; \boldsymbol{\lambda}', \boldsymbol{w}, \alpha)$. Then*

$$\|\boldsymbol{\beta}^+ - \boldsymbol{\beta}^{+\prime}\|_2 \leq 2\alpha \|(\boldsymbol{\lambda} - \boldsymbol{\lambda}')_S\|_2, \tag{39}$$

*where $S := \mathrm{supp}(\boldsymbol{\beta}^+) \cup \mathrm{supp}(\boldsymbol{\beta}^{+\prime})$.*

*Proof.* Recall the quadratic approximation in Eq. 51 and the definition that $\mathrm{MPG}(\boldsymbol{\beta}; \boldsymbol{\lambda}, \boldsymbol{w}, \alpha) = \arg\min_{\boldsymbol{z}} \psi_{\alpha,\boldsymbol{\lambda},\boldsymbol{w}}(\boldsymbol{z}, \boldsymbol{\beta})$. By the $\frac{1}{\alpha}$-strongly convexity of $\psi_{\alpha,\boldsymbol{\lambda},\boldsymbol{w}}$ in $\boldsymbol{z}$, it holds true for all $\boldsymbol{z}$ that

$$\frac{1}{2\alpha} \|\boldsymbol{z} - \boldsymbol{\beta}^+\|_2^2 \leq \psi_{\alpha,\boldsymbol{\lambda},\boldsymbol{w}}(\boldsymbol{z}, \boldsymbol{\beta}) - \psi_{\alpha,\boldsymbol{\lambda},\boldsymbol{w}}(\boldsymbol{\beta}^+, \boldsymbol{\beta})$$

$$= \langle \nabla L_{\boldsymbol{\lambda},\boldsymbol{w}}(\boldsymbol{\beta}), \boldsymbol{z} - \boldsymbol{\beta}^+ \rangle + \frac{1}{2\alpha}\left(\|\boldsymbol{z} - \boldsymbol{\beta}\|_2^2 - \|\boldsymbol{\beta}^+ - \boldsymbol{\beta}\|_2^2\right) + \left(\|\boldsymbol{z}\|_{\boldsymbol{\lambda},1} - \|\boldsymbol{\beta}^+\|_{\boldsymbol{\lambda},1}\right) \tag{40}$$

Similarly,

$$\frac{1}{2\alpha} \|\boldsymbol{z} - \boldsymbol{\beta}^{+\prime}\|_2^2 \tag{41}$$

$$\leq \langle \nabla L_{\boldsymbol{\lambda}',\boldsymbol{w}}(\boldsymbol{\beta}), \boldsymbol{z} - \boldsymbol{\beta}^{+\prime} \rangle + \frac{1}{2\alpha}\left(\|\boldsymbol{z} - \boldsymbol{\beta}\|_2^2 - \|\boldsymbol{\beta}^{+\prime} - \boldsymbol{\beta}\|_2^2\right) + \left(\|\boldsymbol{z}\|_{\boldsymbol{\lambda}',1} - \|\boldsymbol{\beta}^{+\prime}\|_{\boldsymbol{\lambda}',1}\right). \tag{42}$$

Taking $z = \beta^{+\prime}$ in Eq. 40, taking $z = \beta^+$ in Eq. 42, and summing up the two inequalities, we have

$$
\begin{aligned}
\frac{1}{\alpha} \left\| \beta^+ - \beta^{+\prime} \right\|_2^2 &\leq \langle \nabla L_{\boldsymbol{\lambda}, \boldsymbol{w}}(\boldsymbol{\beta}) - \nabla L_{\boldsymbol{\lambda}', \boldsymbol{w}}(\boldsymbol{\beta}), \beta^{+\prime} - \beta^+ \rangle \\
&\quad + \left\| \beta^+ \circ (\boldsymbol{\lambda}' - \boldsymbol{\lambda}) \right\|_1 - \left\| \beta^{+\prime} \circ (\boldsymbol{\lambda}' - \boldsymbol{\lambda}) \right\|_1 \\
&= \langle \nabla \mathcal{Q}_{\boldsymbol{\lambda}, \boldsymbol{w}}(\boldsymbol{\beta}) - \nabla \mathcal{Q}_{\boldsymbol{\lambda}', \boldsymbol{w}}(\boldsymbol{\beta}), \beta^{+\prime} - \beta^+ \rangle + \left\| \beta^+ - \beta^{+\prime} \right\|_{|\boldsymbol{\lambda} - \boldsymbol{\lambda}'|, 1} \\
&\leq \sum_{j=1}^{d} |q'_{\boldsymbol{w}}(\lambda_j, \beta_j) - q'_{\boldsymbol{w}}(\lambda_j', \beta_j)| |\beta_j^{+\prime} - \beta_j^+| + \left\| \beta^+ - \beta^{+\prime} \right\|_{|\boldsymbol{\lambda} - \boldsymbol{\lambda}'|, 1} \\
\text{(by Assumption B.1)} \quad &\leq \sum_{j=1}^{d} |\lambda_j - \lambda_j'| |\beta_j^{+\prime} - \beta_j^+| + \left\| \beta^+ - \beta^{+\prime} \right\|_{|\boldsymbol{\lambda} - \boldsymbol{\lambda}'|, 1} \\
&= 2 \left\| \beta^+ - \beta^{+\prime} \right\|_{|\boldsymbol{\lambda} - \boldsymbol{\lambda}'|, 1} \\
&\leq 2 \left\| (\boldsymbol{\lambda} - \boldsymbol{\lambda}')_S \right\|_2 \left\| \beta^{+\prime} - \beta^+ \right\|_2
\end{aligned}
$$

where $S := \mathrm{supp}(\beta^+) \cup \mathrm{supp}(\beta^{+\prime})$. Dividing both side by $\left\| \beta^{+\prime} - \beta^+ \right\|_2$ draws the conclusion. $\qquad \square$

**Lemma F.6** (Robustness to $\boldsymbol{w}$). *Let* $\beta^+ = \mathrm{MPG}(\boldsymbol{\beta}; \boldsymbol{\lambda}, \boldsymbol{w}, \alpha)$ *and* $\beta^{+\prime} = \mathrm{MPG}(\boldsymbol{\beta}; \boldsymbol{\lambda}, \boldsymbol{w}', \alpha)$. *Then*

$$
\| \beta^+ - \beta^{+\prime} \|_2 \leq C \| \boldsymbol{w} - \boldsymbol{w}' \|_2 \| (\boldsymbol{\lambda})_S \|_2, \tag{43}
$$

*where* $S := \mathrm{supp}(\beta^+) \cup \mathrm{supp}(\beta^{+\prime})$ *and* $C$ *is some absolute constant.*

*Proof.* Recall the quadratic approximation in Eq. 51 and the definition that $\mathrm{MPG}(\boldsymbol{\beta}; \boldsymbol{\lambda}, \boldsymbol{w}, \alpha) = \arg\min_{\boldsymbol{z}} \psi_{\alpha, \boldsymbol{\lambda}, \boldsymbol{w}}(\boldsymbol{z}, \boldsymbol{\beta})$. By the $\frac{1}{\alpha}$-strongly convexity of $\psi_{\alpha, \boldsymbol{\lambda}, \boldsymbol{w}}$ in $\boldsymbol{\beta}$, it holds true for all $\boldsymbol{z}$ in the restricted subspace that

$$
\begin{aligned}
\frac{1}{2\alpha} \left\| \boldsymbol{z} - \beta^+ \right\|_2^2 &\leq \psi_{\alpha, \boldsymbol{\lambda}, \boldsymbol{w}}(\boldsymbol{z}, \boldsymbol{\beta}) - \psi_{\alpha, \boldsymbol{\lambda}, \boldsymbol{w}}(\beta^+, \boldsymbol{\beta}) \\
&= \langle \nabla L_{\boldsymbol{\lambda}, \boldsymbol{w}}(\boldsymbol{\beta}), \boldsymbol{z} - \beta^+ \rangle + \frac{1}{2\alpha} \left( \| \boldsymbol{z} - \boldsymbol{\beta} \|_2^2 - \| \beta^+ - \boldsymbol{\beta} \|_2^2 \right) + (\| \boldsymbol{z} \|_{\boldsymbol{\lambda}, 1} - \| \beta^+ \|_{\boldsymbol{\lambda}, 1}) \tag{44}
\end{aligned}
$$

Similarly,

$$
\begin{aligned}
\frac{1}{2\alpha} &\left\| \boldsymbol{z} - \beta^{+\prime} \right\|_2^2 \\
&\leq \langle \nabla L_{\boldsymbol{\lambda}, \boldsymbol{w}'}(\boldsymbol{\beta}), \boldsymbol{z} - \beta^{+\prime} \rangle + \frac{1}{2\alpha} \left( \| \boldsymbol{z} - \boldsymbol{\beta} \|_2^2 - \| \beta^{+\prime} - \boldsymbol{\beta} \|_2^2 \right) + (\| \boldsymbol{z} \|_{\boldsymbol{\lambda}, 1} - \| \beta^{+\prime} \|_{\boldsymbol{\lambda}, 1}). \tag{45}
\end{aligned}
$$

Taking $\boldsymbol{z} = \beta^{+\prime}$ in Eq. 44, taking $\boldsymbol{z} = \beta^+$ in Eq. 45, and summing up the two inequalities, we have

$$
\begin{aligned}
\frac{1}{\alpha} \left\| \beta^+ - \beta^{+\prime} \right\|_2^2 &\leq \langle \nabla L_{\boldsymbol{\lambda}, \boldsymbol{w}}(\boldsymbol{\beta}) - \nabla L_{\boldsymbol{\lambda}, \boldsymbol{w}'}(\boldsymbol{\beta}), \beta^{+\prime} - \beta^+ \rangle \\
&= \langle \nabla \mathcal{Q}_{\boldsymbol{\lambda}, \boldsymbol{w}}(\boldsymbol{\beta}) - \nabla \mathcal{Q}_{\boldsymbol{\lambda}, \boldsymbol{w}'}(\boldsymbol{\beta}), \beta^{+\prime} - \beta^+ \rangle \\
&\leq \sum_{j=1}^{d} |q'_{\boldsymbol{w}}(\lambda_j, \beta_j) - q'_{\boldsymbol{w}'}(\lambda_j, \beta_j)| |\beta_j^{+\prime} - \beta_j^+|.
\end{aligned}
$$

Note that $q'_{\boldsymbol{w}}(\lambda_j, \beta_j) = \sum_{i=1}^{3} \frac{\exp(w_i)}{Z(\boldsymbol{w})} q^{(i)\prime}(\lambda_j, \boldsymbol{\beta}_j)$ where $Z(\boldsymbol{w}) = \sum_{i=1}^{3} \exp(w_i)$. Then

$$
|q'_{\boldsymbol{w}}(\lambda_j, \beta_j) - q'_{\boldsymbol{w}'}(\lambda_j, \beta_j)| \leq \sum_{i=1}^{3} \left| \frac{\exp(w_i)}{Z(\boldsymbol{w})} - \frac{\exp(w_i')}{Z(\boldsymbol{w}')} \right| \left| q^{(i)\prime}(\lambda_j, \boldsymbol{\beta}_j) \right|
$$

$$
\text{(by Assumption B.1)} \quad \leq \sum_{i=1}^{3} \left| \frac{\exp(w_i)}{Z(\boldsymbol{w})} - \frac{\exp(w_i')}{Z(\boldsymbol{w}')} \right| \lambda_j.
$$

Therefore,

$$\frac{1}{\alpha}\left\|\boldsymbol{\beta}^+ - \boldsymbol{\beta}^{+\prime}\right\|_2^2 \leq \sum_{i=1}^{3}\left|\frac{\exp(w_i)}{Z(\boldsymbol{w})} - \frac{\exp(w_i')}{Z(\boldsymbol{w}')}\right|\sum_{j=1}^{d}\lambda_j|\beta_j^{+\prime} - \beta_j^+|$$

$$\leq \sum_{i=1}^{3}\left|\frac{\exp(w_i)}{Z(\boldsymbol{w})} - \frac{\exp(w_i')}{Z(\boldsymbol{w}')}\right|\|(\boldsymbol{\lambda})_S\|_2\left\|\boldsymbol{\beta}^{+\prime} - \boldsymbol{\beta}^+\right\|_2$$

(By Lipschitz continuity) $\quad \leq C\|\boldsymbol{w} - \boldsymbol{w}'\|_2\|(\boldsymbol{\lambda})_S\|_2\left\|\boldsymbol{\beta}^{+\prime} - \boldsymbol{\beta}^+\right\|_2,$

for some constant $C$. $\hfill\square$

## G KEY LEMMAS

This section supplies some fundamental results for the modified proximal gradient algorithm. All the results in this section are based on the following assumption and notations.

**Assumption G.1.** *Assume $\mathcal{P}$ satisfies Assumption C.1 and consider a problem $(Z_n, \boldsymbol{\beta}^*) \in \mathcal{P}$. Assume we have a penalty function $P(\boldsymbol{\lambda}, \boldsymbol{\beta})$ whose concave component $Q(\boldsymbol{\lambda}, \boldsymbol{\beta}) = P(\boldsymbol{\lambda}, \boldsymbol{\beta}) - \|\boldsymbol{\lambda} \circ \boldsymbol{\beta}\|_1$ satisfies the regularization conditions in Assumption B.1 with a Lipschitz constant $\xi$ for the condition (a). We adopt the following notations for the statements in this sections.*

$$\begin{aligned}
\text{empirical loss} \quad & L_n(\boldsymbol{\beta}) := L_n(Z_{1:n}, \boldsymbol{\beta}) \\
\text{modified loss} \quad & L_{\boldsymbol{\lambda}}(\boldsymbol{\beta}) := L_n(Z_{1:n}, \boldsymbol{\beta}) + Q(\boldsymbol{\lambda}, \boldsymbol{\beta}) \\
\text{regularized loss} \quad & \phi_{\boldsymbol{\lambda}}(\boldsymbol{\beta}) := L_n(Z_{1:n}, \boldsymbol{\beta}) + P(\boldsymbol{\lambda}, \boldsymbol{\beta}) = L_{\boldsymbol{\lambda}}(\boldsymbol{\beta}) + \|\boldsymbol{\beta}\|_{\boldsymbol{\lambda},1} \\
\text{local optimal} \quad & \widehat{\boldsymbol{\beta}}_{\boldsymbol{\lambda}} \in \operatorname*{arg\,min}_{\boldsymbol{\beta} \in \mathbb{R}^d : \|\boldsymbol{\beta}_{\overline{S^*}}\|_0 \le \tilde{s}} \phi_{\boldsymbol{\lambda}}(\boldsymbol{\beta})
\end{aligned}$$

$$\text{quadratic approximation} \quad \psi_{\alpha, \boldsymbol{\lambda}}(\boldsymbol{z}, \boldsymbol{\beta}) := L_{\boldsymbol{\lambda}}(\boldsymbol{\beta}) + \langle \nabla L_{\boldsymbol{\lambda}}(\boldsymbol{\beta}), \boldsymbol{z} - \boldsymbol{\beta} \rangle + \frac{1}{2\alpha} \|\boldsymbol{z} - \boldsymbol{\beta}\|_2^2 + \|\boldsymbol{z}\|_{\boldsymbol{\lambda},1}$$

$$\text{sub-optimality} \quad \omega_{\boldsymbol{\lambda}}(\boldsymbol{\beta}) := \min_{\boldsymbol{\xi}' \in \partial \|\boldsymbol{\beta}\|_1} \max_{\boldsymbol{\beta}'} \left\{ \frac{(\boldsymbol{\beta} - \boldsymbol{\beta}')^\top}{\|\boldsymbol{\beta} - \boldsymbol{\beta}'\|_{\boldsymbol{\lambda},1}} (\nabla L_{\boldsymbol{\lambda}}(\boldsymbol{\beta}) + \boldsymbol{\lambda} \circ \boldsymbol{\xi}') \right\}.$$

**Lemma G.1** (Contraction of MPG). *Assume Assumption G.1. Let $\boldsymbol{\beta}^0, \cdots, \boldsymbol{\beta}^k$ be a sequence of vectors obtained by the modified proximal gradient updates:*

$$\begin{aligned}
\boldsymbol{\beta}^k &= \texttt{MPG}(\boldsymbol{\beta}^{k-1}; \boldsymbol{\lambda}, \boldsymbol{w}, \alpha) \\
&:= \mathcal{T}_{\alpha \cdot \boldsymbol{\lambda}} \left( \boldsymbol{\beta}^{k-1} - \alpha \left( \nabla L_n(\boldsymbol{\beta}^{k-1}) + \nabla_{\boldsymbol{\beta}} Q(\boldsymbol{\lambda}, \boldsymbol{\beta}^{k-1}) \right) \right).
\end{aligned}$$

*Assume $\|(\boldsymbol{\beta}^k)_{\overline{S^*}}\|_0 \le \tilde{s}$. Then*

$$\left( \frac{1}{\alpha} - \frac{\rho_+}{2} \right) \|\boldsymbol{\beta}^k - \boldsymbol{\beta}^{k-1}\|_2^2 \le \phi_{\boldsymbol{\lambda}}(\boldsymbol{\beta}^{k-1}) - \phi_{\boldsymbol{\lambda}}(\boldsymbol{\beta}^k). \tag{46}$$

*Proof.* Note that

$$\boldsymbol{\beta}^k = \boldsymbol{\beta}^{k-1} - \alpha \boldsymbol{\delta}_\alpha(\boldsymbol{\beta}^{k-1}), \quad \text{where } \boldsymbol{\delta}_\alpha(\boldsymbol{\beta}) := \alpha^{-1} \left( \boldsymbol{\beta} - \mathcal{T}_{\alpha \boldsymbol{\lambda}} \left( \boldsymbol{\beta} - \alpha \nabla L_{\boldsymbol{\lambda}}(\boldsymbol{\beta}) \right) \right).$$

It is easy to observe that $\boldsymbol{\delta}_\alpha(\boldsymbol{\beta}) \in \nabla L_{\boldsymbol{\lambda}}(\boldsymbol{\beta}) + \boldsymbol{\lambda} \circ \partial \|\boldsymbol{\beta} - \alpha \boldsymbol{\delta}_\alpha(\boldsymbol{\beta})\|_1$. By RSS in Lemma B.1, it holds that

$$L_{\boldsymbol{\lambda}}\left(\boldsymbol{\beta}^k\right) \le L_{\boldsymbol{\lambda}}(\boldsymbol{\beta}^{k-1}) - \alpha \nabla L_{\boldsymbol{\lambda}}(\boldsymbol{\beta}^{k-1})^\top \boldsymbol{\delta}_\alpha(\boldsymbol{\beta}^{k-1}) + \frac{\rho_+}{2} \alpha^2 \|\boldsymbol{\delta}_\alpha(\boldsymbol{\beta}^{k-1})\|_2^2.$$

Furthermore, by RSC in Lemma B.1, for any $\boldsymbol{z}$ in the space $\{\|\boldsymbol{z} - \boldsymbol{\beta}^{k-1}\|_0 \le s^* + 2\tilde{s}\}$,

$$L_{\boldsymbol{\lambda}}\left(\boldsymbol{\beta}^k\right) \le L_{\boldsymbol{\lambda}}(\boldsymbol{z}) + \nabla L_{\boldsymbol{\lambda}}(\boldsymbol{\beta}^{k-1})^\top(\boldsymbol{\beta} - \boldsymbol{z}) - \alpha \nabla L_{\boldsymbol{\lambda}}(\boldsymbol{\beta}^{k-1})^\top \boldsymbol{\delta}_\alpha(\boldsymbol{\beta}^{k-1}) \tag{47}$$

$$+ \frac{\rho_+}{2} \alpha^2 \|\boldsymbol{\delta}_\alpha(\boldsymbol{\beta}^{k-1})\|_2^2 - \frac{\rho_- - \xi}{2} \|\boldsymbol{\beta}^{k-1} - \boldsymbol{z}\|_2^2. \tag{48}$$

Denote $\boldsymbol{v} = \boldsymbol{\delta}_\alpha(\boldsymbol{\beta}^{k-1}) - \nabla L_{\boldsymbol{\lambda}}(\boldsymbol{\beta}^{k-1})$. We have $\boldsymbol{v} \in \boldsymbol{\lambda} \circ \partial \|\boldsymbol{\beta}^{k-1} - \alpha \boldsymbol{\delta}_\alpha(\boldsymbol{\beta}^{k-1})\|_1$ since $\boldsymbol{\delta}_\alpha(\boldsymbol{\beta}^{k-1}) \in \nabla L_{\boldsymbol{\lambda}}(\boldsymbol{\beta}^{k-1}) + \boldsymbol{\lambda} \circ \partial \|\boldsymbol{\beta}^{k-1} - \alpha \boldsymbol{\delta}_\alpha(\boldsymbol{\beta}^{k-1})\|_1$. By the convexity of $\|\cdot\|_{\boldsymbol{\lambda},1}$,

$$\|\boldsymbol{\beta}^{k-1} - \alpha \boldsymbol{\delta}_\alpha(\boldsymbol{\beta}^{k-1})\|_{\boldsymbol{\lambda},1} \le \|\boldsymbol{z}\|_{\boldsymbol{\lambda},1} + \boldsymbol{v}^\top \left( \boldsymbol{\beta}^{k-1} - \alpha \boldsymbol{\delta}_\alpha(\boldsymbol{\beta}^{k-1}) - \boldsymbol{z} \right).$$

Combining this inequality with Eq. 48, we have

$$L_{\boldsymbol{\lambda}}(\boldsymbol{\beta}^k) + \|\boldsymbol{\beta}^k\|_{\boldsymbol{\lambda},1} \le L_{\boldsymbol{\lambda}}(\boldsymbol{z}) + \lambda \|\boldsymbol{z}\|_{\boldsymbol{\lambda},1} + \boldsymbol{\delta}_\alpha(\boldsymbol{\beta}^{k-1})^\top(\boldsymbol{\beta}^{k-1} - \boldsymbol{z}) \tag{49}$$

$$- \alpha \left( 1 - \frac{\alpha(\rho_+)}{2} \right) \|\boldsymbol{\delta}_\alpha(\boldsymbol{\beta}^{k-1})\|_2^2 - \frac{\rho_- - \xi}{2} \|\boldsymbol{\beta}^{k-1} - \boldsymbol{z}\|_2^2. \tag{50}$$

Taking $\boldsymbol{z} = \boldsymbol{\beta}^{k-1}$ in Eq. 50 implies that

$$L_{\boldsymbol{\lambda}}(\boldsymbol{\beta}^k) + \|\boldsymbol{\beta}^k\|_{\boldsymbol{\lambda},1} \le L_{\boldsymbol{\lambda}}(\boldsymbol{\beta}^{k-1}) + \|\boldsymbol{\beta}^{k-1}\|_{\boldsymbol{\lambda},1} - \frac{1}{\alpha} \left( 1 - \frac{\alpha(\rho_+)}{2} \right) \|\boldsymbol{\beta}^k - \boldsymbol{\beta}^{k-1}\|_2^2.$$

Therefore,

$$\left(\frac{1}{\alpha} - \frac{\rho_+}{2}\right) \left\|\boldsymbol{\beta}^k - \boldsymbol{\beta}^{k-1}\right\|_2^2 \leq \phi_{\boldsymbol{\lambda}}(\boldsymbol{\beta}^{k-1}) - \phi_{\boldsymbol{\lambda}}(\boldsymbol{\beta}^k).$$

$\square$

**Lemma G.2** (Convergence of modified proximal gradient)**.** *Assume Assumption G.1. Let* $\boldsymbol{\beta}^0, \cdots, \boldsymbol{\beta}^k$ *be a sequence of vectors obtained by the modified proximal gradient updates:*

$$\boldsymbol{\beta}^k = \mathrm{MPG}(\boldsymbol{\beta}^{k-1}; \boldsymbol{\lambda}, \boldsymbol{w}, \alpha)$$
$$:= \mathcal{T}_{\alpha \cdot \boldsymbol{\lambda}} \left(\boldsymbol{\beta}^{k-1} - \alpha \left(\nabla L_n(\boldsymbol{\beta}^{k-1}) + \nabla_{\boldsymbol{\beta}} Q(\boldsymbol{\lambda}, \boldsymbol{\beta}^{k-1})\right)\right)$$

*with a step size* $\alpha$ *in* $(0, \frac{1}{\rho_+}]$*. Assume* $\|(\boldsymbol{\beta}^k)_{\overline{S^*}}\|_0 \leq \tilde{s}$*. Then*

$$\phi_{\boldsymbol{\lambda}}(\boldsymbol{\beta}^k) - \phi_{\boldsymbol{\lambda}}(\widehat{\boldsymbol{\beta}}_{\boldsymbol{\lambda}}) \leq (1 - \alpha(\rho_- - \xi_{\boldsymbol{w}}))^k \left(\phi_{\boldsymbol{\lambda}}(\boldsymbol{\beta}^0) - \phi_{\boldsymbol{\lambda}}(\widehat{\boldsymbol{\beta}}_{\boldsymbol{\lambda}})\right),$$

$$\|\boldsymbol{\beta}^k - \widehat{\boldsymbol{\beta}}_{\boldsymbol{\lambda}}\|_2^2 \leq (1 - \alpha(\rho_- - \xi_{\boldsymbol{w}}))^k \|\boldsymbol{\beta}^0 - \widehat{\boldsymbol{\beta}}_{\boldsymbol{\lambda}}\|_2^2,$$

$$\omega_{\boldsymbol{\lambda}}(\boldsymbol{\beta}^k) \leq \frac{\sqrt{2}(1 + \frac{1}{\alpha \rho_+})\sqrt{\rho_+}}{\min(\boldsymbol{\lambda})} \sqrt{(1 - \alpha(\rho_- - \xi_{\boldsymbol{w}}))^{k-1} \left(\phi_{\boldsymbol{\lambda}}(\boldsymbol{\beta}^0) - \phi_{\boldsymbol{\lambda}}(\widehat{\boldsymbol{\beta}}_{\boldsymbol{\lambda}})\right)}.$$

*Proof.* Consider the quadratic approximation function

$$\psi_{\alpha, \boldsymbol{\lambda}}(\boldsymbol{z}, \boldsymbol{\beta}) := L_{\boldsymbol{\lambda}}(\boldsymbol{\beta}) + \langle \nabla L_{\boldsymbol{\lambda}}(\boldsymbol{\beta}), \boldsymbol{z} - \boldsymbol{\beta} \rangle + \frac{1}{2\alpha} \|\boldsymbol{z} - \boldsymbol{\beta}\|_2^2 + \|\boldsymbol{z}\|_{\boldsymbol{\lambda},1}. \tag{51}$$

By definition,

$$\mathrm{MPG}(\boldsymbol{\beta}^{k-1}; \boldsymbol{\lambda}, \boldsymbol{w}, \alpha) = \arg\min_{\boldsymbol{z}} \psi_{\alpha, \boldsymbol{\lambda}}(\boldsymbol{z}, \boldsymbol{\beta}^{k-1}).$$

Since $\psi_{\alpha, \boldsymbol{\lambda}}(\boldsymbol{z}, \boldsymbol{\beta}^{k-1})$ is $\frac{1}{\alpha}$-strongly convex with respect to $\boldsymbol{z}$, and that $\boldsymbol{\beta}^k$ is its optimal point, we can obtain

$$\psi_{\alpha, \boldsymbol{\lambda}}(\boldsymbol{z}, \boldsymbol{\beta}^{k-1}) \geq \psi_{\alpha, \boldsymbol{\lambda}}(\boldsymbol{\beta}^k, \boldsymbol{\beta}^{k-1}) + \frac{1}{2\alpha} \|\boldsymbol{z} - \boldsymbol{\beta}^k\|_2^2, \quad \forall \boldsymbol{z}. \tag{52}$$

Note that by Lemma B.1,

$$\psi_{\alpha, \boldsymbol{\lambda}}(\boldsymbol{\beta}^k, \boldsymbol{\beta}^{k-1}) = L_{\boldsymbol{\lambda}}(\boldsymbol{\beta}^{k-1}) + \nabla L_{\boldsymbol{\lambda}}(\boldsymbol{\beta}^{k-1})^\top (\boldsymbol{\beta}^k - \boldsymbol{\beta}^{k-1}) + \frac{1}{2\alpha} \|\boldsymbol{\beta}^k - \boldsymbol{\beta}^{k-1}\|_2^2$$
$$+ \|\boldsymbol{\beta}^k\|_{\boldsymbol{\lambda},1}$$
$$\geq L_{\boldsymbol{\lambda}}(\boldsymbol{\beta}^k) + \frac{\frac{1}{\alpha} - \rho_+}{2} \|\boldsymbol{\beta}^k - \boldsymbol{\beta}^{k-1}\|_2^2 + \|\boldsymbol{\beta}^k\|_{\boldsymbol{\lambda},1}$$
$$= \phi_{\boldsymbol{\lambda}}(\boldsymbol{\beta}^k) + \frac{\frac{1}{\alpha} - \rho_+}{2} \|\boldsymbol{\beta}^k - \boldsymbol{\beta}^{k-1}\|_2^2.$$

Similarly, by Lemma B.1,

$$\psi_{\alpha, \boldsymbol{\lambda}}(\boldsymbol{z}, \boldsymbol{\beta}^{k-1}) = L_{\boldsymbol{\lambda}}(\boldsymbol{\beta}^{k-1}) + \nabla L_{\boldsymbol{\lambda}}(\boldsymbol{\beta}^{k-1})^\top (\boldsymbol{z} - \boldsymbol{\beta}^{k-1}) + \frac{1}{2\alpha} \|\boldsymbol{z} - \boldsymbol{\beta}^{k-1}\|_2^2$$
$$+ \|\boldsymbol{z}\|_{\boldsymbol{\lambda},1}$$
$$\leq \phi_{\boldsymbol{\lambda}}(\boldsymbol{z}) + \frac{\frac{1}{\alpha} - (\rho_- - \xi)}{2} \|\boldsymbol{z} - \boldsymbol{\beta}^{k-1}\|_2^2.$$

Combining the above two results with Eq. 52, we have

$$\phi_{\boldsymbol{\lambda}}(\boldsymbol{\beta}^k) \leq \phi_{\boldsymbol{\lambda}}(\boldsymbol{z}) + \frac{\frac{1}{\alpha} - (\rho_- - \xi)}{2} \|\boldsymbol{z} - \boldsymbol{\beta}^{k-1}\|_2^2 - \frac{\frac{1}{\alpha} - \rho_+}{2} \|\boldsymbol{\beta}^k - \boldsymbol{\beta}^{k-1}\|_2^2$$
$$- \frac{1}{2\alpha} \|\boldsymbol{z} - \boldsymbol{\beta}^k\|_2^2 \tag{53}$$
$$\leq \phi_{\boldsymbol{\lambda}}(\boldsymbol{z}) + \frac{\frac{1}{\alpha} - (\rho_- - \xi)}{2} \|\boldsymbol{z} - \boldsymbol{\beta}^{k-1}\|_2^2 - \frac{\frac{1}{\alpha} - \rho_+}{2} \|\boldsymbol{\beta}^k - \boldsymbol{\beta}^{k-1}\|_2^2.$$

Taking $z = c\widehat{\boldsymbol{\beta}}_{\boldsymbol{\lambda}} + (1-c)\boldsymbol{\beta}^{k-1}$ for some $c \in [0,1]$, then

$$\phi_{\boldsymbol{\lambda}}(\boldsymbol{\beta}^k)$$
$$\leq c\phi_{\boldsymbol{\lambda}}(\widehat{\boldsymbol{\beta}}_{\boldsymbol{\lambda}}) + (1-c)\phi_{\boldsymbol{\lambda}}(\boldsymbol{\beta}^{k-1}) + \frac{\left(\frac{1}{\alpha} - (\rho_- - \xi)\right)c^2}{2}\|\boldsymbol{\beta}^{k-1} - \widehat{\boldsymbol{\beta}}_{\boldsymbol{\lambda}}\|_2^2 - \frac{\frac{1}{\alpha} - \rho_+}{2}\|\boldsymbol{\beta}^k - \boldsymbol{\beta}^{k-1}\|_2^2$$
$$\leq c\phi_{\boldsymbol{\lambda}}(\widehat{\boldsymbol{\beta}}_{\boldsymbol{\lambda}}) + (1-c)\phi_{\boldsymbol{\lambda}}(\boldsymbol{\beta}^{k-1}) + \frac{c\left(c\frac{1}{\alpha} - (\rho_- - \xi)\right)}{2}\|\boldsymbol{\beta}^{k-1} - \widehat{\boldsymbol{\beta}}_{\boldsymbol{\lambda}}\|_2^2 - \frac{\frac{1}{\alpha} - \rho_+}{2}\|\boldsymbol{\beta}^k - \boldsymbol{\beta}^{k-1}\|_2^2$$

Taking $c = \alpha(\rho_- - \xi)$, it implies

$$\phi_{\boldsymbol{\lambda}}(\boldsymbol{\beta}^k) - \phi_{\boldsymbol{\lambda}}(\widehat{\boldsymbol{\beta}}_{\boldsymbol{\lambda}}) \leq (1 - \alpha(\rho_- - \xi))\left(\phi_{\boldsymbol{\lambda}}(\boldsymbol{\beta}^{k-1}) - \phi_{\boldsymbol{\lambda}}(\widehat{\boldsymbol{\beta}}_{\boldsymbol{\lambda}})\right) - \frac{\frac{1}{\alpha} - \rho_+}{2}\|\boldsymbol{\beta}^k - \boldsymbol{\beta}^{k-1}\|_2^2.$$

Assume $\alpha \leq \frac{1}{\rho_+}$. Then

$$\phi_{\boldsymbol{\lambda}}(\boldsymbol{\beta}^k) - \phi_{\boldsymbol{\lambda}}(\widehat{\boldsymbol{\beta}}_{\boldsymbol{\lambda}}) \leq (1 - \alpha(\rho_- - \xi))\left(\phi_{\boldsymbol{\lambda}}(\boldsymbol{\beta}^{k-1}) - \phi_{\boldsymbol{\lambda}}(\widehat{\boldsymbol{\beta}}_{\boldsymbol{\lambda}})\right).$$

Taking $z = \widehat{\boldsymbol{\beta}}_{\boldsymbol{\lambda}}$ in Eq. 53 implies

$$\phi_{\boldsymbol{\lambda}}(\boldsymbol{\beta}^k) \leq \phi_{\boldsymbol{\lambda}}(\widehat{\boldsymbol{\beta}}_{\boldsymbol{\lambda}}) + \frac{\frac{1}{\alpha} - (\rho_- - \xi)}{2}\|\widehat{\boldsymbol{\beta}}_{\boldsymbol{\lambda}} - \boldsymbol{\beta}^{k-1}\|_2^2 - \frac{1}{2\alpha}\|\widehat{\boldsymbol{\beta}}_{\boldsymbol{\lambda}} - \boldsymbol{\beta}^k\|_2^2.$$

By the optimality of $\widehat{\boldsymbol{\beta}}_{\boldsymbol{\lambda}}$, we have

$$\frac{\frac{1}{\alpha} - (\rho_- - \xi)}{2}\|\widehat{\boldsymbol{\beta}}_{\boldsymbol{\lambda}} - \boldsymbol{\beta}^{k-1}\|_2^2 - \frac{1}{2\alpha}\|\widehat{\boldsymbol{\beta}}_{\boldsymbol{\lambda}} - \boldsymbol{\beta}^k\|_2^2 \geq 0,$$

which implies

$$\|\widehat{\boldsymbol{\beta}}_{\boldsymbol{\lambda}} - \boldsymbol{\beta}^k\|_2^2 \leq (1 - \alpha(\rho_- - \xi))\|\widehat{\boldsymbol{\beta}}_{\boldsymbol{\lambda}} - \boldsymbol{\beta}^{k-1}\|_2^2.$$

Finally, we derive the upper bound for $\omega_{\boldsymbol{\lambda}}(\boldsymbol{\beta}^k)$. Since $\boldsymbol{\beta}^k = \arg\min_z \psi_{\alpha,\boldsymbol{\lambda}}(z, \boldsymbol{\beta}^{k-1})$, by the optimality condition, for each $j = 1, \cdots, d$, there exists a subgradient $\xi_j \in \partial|\beta_j^k|$ such that

$$\nabla L_{\boldsymbol{\lambda}}(\boldsymbol{\beta}^{k-1}) + \frac{1}{\alpha}(\boldsymbol{\beta}^k - \boldsymbol{\beta}^{k-1}) + \boldsymbol{\lambda} \circ \boldsymbol{\xi} = 0.$$

By the definition of $\omega_{\boldsymbol{\lambda}}$ and combining it with the above equality,

$$\omega_{\boldsymbol{\lambda}}(\boldsymbol{\beta}^k) = \min_{\boldsymbol{\xi}' \in \partial\|\boldsymbol{\beta}^k\|_1} \max_{\boldsymbol{\beta}' \in \Omega} \left\{ \frac{(\boldsymbol{\beta}^k - \boldsymbol{\beta}')^\top}{\|\boldsymbol{\beta}^k - \boldsymbol{\beta}'\|_{\boldsymbol{\lambda},1}}(\nabla L_{\boldsymbol{\lambda}}(\boldsymbol{\beta}^k) + \boldsymbol{\lambda} \circ \boldsymbol{\xi}') \right\}$$
$$\leq \max_{\boldsymbol{\beta}' \in \Omega} \left\{ \frac{(\boldsymbol{\beta}^k - \boldsymbol{\beta}')^\top}{\|\boldsymbol{\beta}^k - \boldsymbol{\beta}'\|_{\boldsymbol{\lambda},1}}(\nabla L_{\boldsymbol{\lambda}}(\boldsymbol{\beta}^k) + \boldsymbol{\lambda} \circ \boldsymbol{\xi}) \right\}$$
$$= \max_{\boldsymbol{\beta}' \in \Omega} \left\{ \frac{(\boldsymbol{\beta}^k - \boldsymbol{\beta}')^\top}{\|\boldsymbol{\beta}^k - \boldsymbol{\beta}'\|_{\boldsymbol{\lambda},1}}\left(\nabla L_{\boldsymbol{\lambda}}(\boldsymbol{\beta}^k) - \nabla L_{\boldsymbol{\lambda}}(\boldsymbol{\beta}^{k-1}) + \frac{1}{\alpha}(\boldsymbol{\beta}^{k-1} - \boldsymbol{\beta}^k)\right) \right\}$$
$$= \max_{\boldsymbol{\beta}' \in \Omega} \left\{ \sum_{j=1}^d \frac{(\beta_j^k - \beta_j')\lambda_j}{\|\boldsymbol{\beta}^k - \boldsymbol{\beta}'\|_{\boldsymbol{\lambda},1}}\frac{\left(\nabla L_{\boldsymbol{\lambda}}(\boldsymbol{\beta}^k) - \nabla L_{\boldsymbol{\lambda}}(\boldsymbol{\beta}^{k-1}) + \frac{1}{\alpha}(\boldsymbol{\beta}^{k-1} - \boldsymbol{\beta}^k)\right)_j}{\lambda_j} \right\}$$
$$\leq \frac{1}{\min(\boldsymbol{\lambda})}\|\nabla L_{\boldsymbol{\lambda}}(\boldsymbol{\beta}^k) - \nabla L_{\boldsymbol{\lambda}}(\boldsymbol{\beta}^{k-1}) + \frac{1}{\alpha}(\boldsymbol{\beta}^{k-1} - \boldsymbol{\beta}^k)\|_\infty$$
$$\leq \frac{1}{\min(\boldsymbol{\lambda})}\|\nabla L_{\boldsymbol{\lambda}}(\boldsymbol{\beta}^k) - \nabla L_{\boldsymbol{\lambda}}(\boldsymbol{\beta}^{k-1}) + \frac{1}{\alpha}(\boldsymbol{\beta}^{k-1} - \boldsymbol{\beta}^k)\|_2$$
$$\leq \frac{1}{\min(\boldsymbol{\lambda})}\|\nabla L_{\boldsymbol{\lambda}}(\boldsymbol{\beta}^k) - \nabla L_{\boldsymbol{\lambda}}(\boldsymbol{\beta}^{k-1})\|_2 + \frac{1}{\alpha}\|(\boldsymbol{\beta}^{k-1} - \boldsymbol{\beta}^k)\|_2$$
$$\text{(by RSS)} \leq \frac{1}{\min(\boldsymbol{\lambda})}(\rho_+ + \frac{1}{\alpha})\|(\boldsymbol{\beta}^{k-1} - \boldsymbol{\beta}^k)\|_2.$$

By Lemma G.1, $\|(\boldsymbol{\beta}^{k-1} - \boldsymbol{\beta}^k)\|_2 \leq \sqrt{\frac{2}{2\frac{1}{\alpha}-(\rho_+)} (\phi_{\boldsymbol{\lambda}}(\boldsymbol{\beta}^{k-1}) - \phi_{\boldsymbol{\lambda}}(\boldsymbol{\beta}^k))}$. Therefore,

$$
\begin{aligned}
\omega_{\boldsymbol{\lambda}}(\boldsymbol{\beta}^k) &\leq \frac{1}{\min(\boldsymbol{\lambda})}(\rho_+ + \frac{1}{\alpha})\sqrt{\frac{2}{2\frac{1}{\alpha}-(\rho_+)} (\phi_{\boldsymbol{\lambda}}(\boldsymbol{\beta}^{k-1}) - \phi_{\boldsymbol{\lambda}}(\boldsymbol{\beta}^k))} \\
&\leq \frac{1}{\min(\boldsymbol{\lambda})}(\rho_+ + \frac{1}{\alpha})\sqrt{\frac{2}{\rho_+} (\phi_{\boldsymbol{\lambda}}(\boldsymbol{\beta}^{k-1}) - \phi_{\boldsymbol{\lambda}}(\boldsymbol{\beta}^k))} \\
&\leq \frac{1}{\min(\boldsymbol{\lambda})}(\rho_+ + \frac{1}{\alpha})\sqrt{\frac{2}{\rho_+} \left(\phi_{\boldsymbol{\lambda}}(\boldsymbol{\beta}^{k-1}) - \phi_{\boldsymbol{\lambda}}(\widehat{\boldsymbol{\beta}}_{\boldsymbol{\lambda}})\right)} \\
&\leq \frac{1}{\min(\boldsymbol{\lambda})}(\rho_+ + \frac{1}{\alpha})\sqrt{\frac{2}{\rho_+} (1 - \alpha(\rho_- - \xi))^{k-1} \left(\phi_{\boldsymbol{\lambda}}(\boldsymbol{\beta}^0) - \phi_{\boldsymbol{\lambda}}(\widehat{\boldsymbol{\beta}}_{\boldsymbol{\lambda}})\right)}.
\end{aligned}
$$

$\square$

**Lemma G.3** (Statistical $\ell_2$ error). *Assume Assumption G.1. If the following conditions are satisfied*

$$\|\boldsymbol{\beta} - \boldsymbol{\beta}^*\|_0 \leq s^* + 2\tilde{s}, \tag{54}$$

$$\phi_{\boldsymbol{\lambda}}(\boldsymbol{\beta}) - \phi_{\boldsymbol{\lambda}}(\boldsymbol{\beta}^*) \leq \frac{21/2}{\rho_- - \xi}\|\boldsymbol{\lambda}_{S^*}\|_2^2, \tag{55}$$

$$\lambda_j \geq 8\left|[\nabla L_n(\boldsymbol{\beta}^*)]_j\right|, \tag{56}$$

*then* $\|\boldsymbol{\beta} - \boldsymbol{\beta}^*\|_2 \leq \min_{a \geq 0} \max\left\{\sqrt{21 + \frac{17}{4}a}, \frac{21}{a} + \frac{17}{4}\right\}\frac{\|\boldsymbol{\lambda}_{S^*}\|_2}{\rho_- - \xi} \leq \frac{15/2\|\boldsymbol{\lambda}_{S^*}\|_2}{\rho_- - \xi}.$

*Proof.* By the restricted strong convexity of $L_{\boldsymbol{\lambda}}$ in Lemma B.1,

$$
\begin{aligned}
\frac{\rho_- - \xi}{2}&\|\boldsymbol{\beta} - \boldsymbol{\beta}^*\|_2^2 \\
&\leq L_{\boldsymbol{\lambda}}(\boldsymbol{\beta}) - L_{\boldsymbol{\lambda}}(\boldsymbol{\beta}^*) - \nabla L_{\boldsymbol{\lambda}}(\boldsymbol{\beta}^*)^\top(\boldsymbol{\beta} - \boldsymbol{\beta}^*) \\
&= \phi_{\boldsymbol{\lambda}}(\boldsymbol{\beta}) - \phi_{\boldsymbol{\lambda}}(\boldsymbol{\beta}^*) + (\|\boldsymbol{\beta}^*\|_{\boldsymbol{\lambda},1} - \|\boldsymbol{\beta}\|_{\boldsymbol{\lambda},1}) - \nabla L_{\boldsymbol{\lambda}}(\boldsymbol{\beta}^*)^\top(\boldsymbol{\beta} - \boldsymbol{\beta}^*) \\
&\leq \frac{21/2}{\rho_- - \xi}\|\boldsymbol{\lambda}_{S^*}\|_2^2 + \underbrace{(\|\boldsymbol{\beta}^*\|_{\boldsymbol{\lambda},1} - \|\boldsymbol{\beta}\|_{\boldsymbol{\lambda},1})}_{(i)} \underbrace{-\nabla L_{\boldsymbol{\lambda}}(\boldsymbol{\beta}^*)^\top(\boldsymbol{\beta} - \boldsymbol{\beta}^*)}_{(ii)}. \tag{57}
\end{aligned}
$$

For the term (i),

$$
\begin{aligned}
\|\boldsymbol{\beta}^*\|_{\boldsymbol{\lambda},1} - \|\boldsymbol{\beta}\|_{\boldsymbol{\lambda},1} &= \|\boldsymbol{\beta}_{S^*}^*\|_{\boldsymbol{\lambda}_{S^*},1} - \|\boldsymbol{\beta}_{S^*}\|_{\boldsymbol{\lambda}_{S^*},1} + (\|\boldsymbol{\beta}_{\overline{S^*}}^*\|_{\boldsymbol{\lambda}_{\overline{S^*}},1} - \|\boldsymbol{\beta}_{\overline{S^*}}\|_{\boldsymbol{\lambda}_{\overline{S^*}},1}) \\
&= \|\boldsymbol{\beta}_{S^*}^*\|_{\boldsymbol{\lambda}_{S^*},1} - \|\boldsymbol{\beta}_{S^*}\|_{\boldsymbol{\lambda}_{S^*},1} - \|\boldsymbol{\beta}_{\overline{S^*}}\|_{\boldsymbol{\lambda}_{\overline{S^*}},1} \\
&\leq \|(\boldsymbol{\beta} - \boldsymbol{\beta}^*)_{S^*}\|_{\boldsymbol{\lambda}_{S^*},1} - \|\boldsymbol{\beta}_{\overline{S^*}}\|_{\boldsymbol{\lambda}_{\overline{S^*}},1} \\
&= \|(\boldsymbol{\beta} - \boldsymbol{\beta}^*)_{S^*}\|_{\boldsymbol{\lambda}_{S^*},1} - \|(\boldsymbol{\beta} - \boldsymbol{\beta}^*)_{\overline{S^*}}\|_{\boldsymbol{\lambda}_{\overline{S^*}},1}.
\end{aligned}
$$

For the term (ii),

$$
\begin{aligned}
-\nabla L_{\boldsymbol{\lambda}}(\boldsymbol{\beta}^*)^\top(\boldsymbol{\beta} - \boldsymbol{\beta}^*) &= -\nabla L_n(\boldsymbol{\beta}^*)^\top(\boldsymbol{\beta} - \boldsymbol{\beta}^*) - \nabla Q_{\boldsymbol{\lambda}}(\boldsymbol{\beta}^*)^\top(\boldsymbol{\beta} - \boldsymbol{\beta}^*) \\
\text{(By Eq. 56)} &\leq \frac{1}{8}\|\boldsymbol{\beta} - \boldsymbol{\beta}^*\|_{\boldsymbol{\lambda},1} - \nabla Q_{\boldsymbol{\lambda}}(\boldsymbol{\beta}^*)^\top(\boldsymbol{\beta} - \boldsymbol{\beta}^*) \\
\text{(since } q_{\lambda_j}'(0) = 0) &= \frac{1}{8}\|\boldsymbol{\beta} - \boldsymbol{\beta}^*\|_{\boldsymbol{\lambda},1} - \nabla Q_{\boldsymbol{\lambda}}(\boldsymbol{\beta}_{S^*}^*)^\top(\boldsymbol{\beta} - \boldsymbol{\beta}^*)_{S^*} \\
\text{(since } \left|q_{\lambda_j}'(\beta)\right| \leq \lambda_j) &\leq \frac{1}{8}\|\boldsymbol{\beta} - \boldsymbol{\beta}^*\|_{\boldsymbol{\lambda},1} + \|(\boldsymbol{\beta} - \boldsymbol{\beta}^*)_{S^*}\|_{\boldsymbol{\lambda}_{S^*},1} \\
&= \frac{9}{8}\|(\boldsymbol{\beta} - \boldsymbol{\beta}^*)_{S^*}\|_{\boldsymbol{\lambda}_{S^*},1} + \frac{1}{8}\|(\boldsymbol{\beta} - \boldsymbol{\beta}^*)_{\overline{S^*}}\|_{\boldsymbol{\lambda}_{\overline{S^*}},1}.
\end{aligned}
$$

Combining the upper bounds of term (i) and term (ii) with Eq. 57, we have

$$\|\boldsymbol{\beta} - \boldsymbol{\beta}^*\|_2^2$$

$$\leq \frac{21}{(\rho_- - \xi)^2}\|\boldsymbol{\lambda}_{S^*}\|_2^2 + \frac{17}{4(\rho_- - \xi)}\|(\boldsymbol{\beta} - \boldsymbol{\beta}^*)_{S^*}\|_{\boldsymbol{\lambda}_{S^*},1} - \frac{7}{4(\rho_- - \xi)}\|(\boldsymbol{\beta} - \boldsymbol{\beta}^*)_{\overline{S}^*}\|_{\boldsymbol{\lambda}_{\overline{S}^*},1}, \quad (58)$$

$$\leq \frac{1}{\rho_- - \xi}\left(\frac{21}{\rho_- - \xi}\|\boldsymbol{\lambda}_{S^*}\|_2^2 + \frac{17}{4}\|(\boldsymbol{\beta} - \boldsymbol{\beta}^*)_{S^*}\|_{\boldsymbol{\lambda}_{S^*},1}\right).$$

Let $a \geq 0$ be a constant. If $\|(\boldsymbol{\beta} - \boldsymbol{\beta}^*)_{S^*}\|_{\boldsymbol{\lambda}_{S^*},1} \leq \frac{a}{\rho_- - \xi}\|\boldsymbol{\lambda}_{S^*}\|_2^2$, then

$$\|\boldsymbol{\beta} - \boldsymbol{\beta}^*\|_2 \leq \sqrt{21 + \frac{17}{4}a}\frac{\|\boldsymbol{\lambda}_{S^*}\|_2}{\rho_- - \xi}.$$

If $\|(\boldsymbol{\beta} - \boldsymbol{\beta}^*)_{S^*}\|_{\boldsymbol{\lambda}_{S^*},1} > \frac{a}{\rho_- - \xi}\|\boldsymbol{\lambda}_{S^*}\|_2^2$, then

$$\|\boldsymbol{\beta} - \boldsymbol{\beta}^*\|_2^2 \leq \frac{1}{\rho_- - \xi}\left(\frac{21}{a} + \frac{17}{4}\right)\|(\boldsymbol{\beta} - \boldsymbol{\beta}^*)_{S^*}\|_{\boldsymbol{\lambda}_{S^*},1} \leq \left(\frac{21}{a} + \frac{17}{4}\right)\frac{\|\boldsymbol{\lambda}_{S^*}\|_2}{\rho_- - \xi}\|(\boldsymbol{\beta} - \boldsymbol{\beta}^*)\|_2,$$

which implies $\|\boldsymbol{\beta} - \boldsymbol{\beta}^*\|_2 \leq \left(\frac{21}{a} + \frac{17}{4}\right)\frac{\|\boldsymbol{\lambda}_{S^*}\|_2}{\rho_- - \xi}$. Combining the two cases, we know that

$$\|\boldsymbol{\beta} - \boldsymbol{\beta}^*\|_2 \leq \min_{a \geq 0}\max\left\{\sqrt{21 + \frac{17}{4}a}, \frac{21}{a} + \frac{17}{4}\right\}\frac{\|\boldsymbol{\lambda}_{S^*}\|_2}{\rho_- - \xi}.$$

Taking $a = 7$ it is easy to derive that

$$\|\boldsymbol{\beta} - \boldsymbol{\beta}^*\|_2 \leq \frac{15/2\|\boldsymbol{\lambda}_{S^*}\|_2}{\rho_- - \xi}.$$

$\square$

**Lemma G.4** (Retain in the restricted space)**.** *Assume Assumption G.1. Let $\boldsymbol{\beta}^0, \cdots, \boldsymbol{\beta}^k$ be a sequence of vectors obtained by the following update:*

$$\boldsymbol{\beta}^k = \mathrm{MPG}(\boldsymbol{\beta}^{k-1}; \boldsymbol{\lambda}, \boldsymbol{w}, \alpha)$$

*with a step sizes $\alpha$ in $[\alpha_{\min}, \frac{1}{\rho_+}]$. Assume the following conditions:*

$$\|(\boldsymbol{\beta}^0)_{\overline{S}^*}\|_0 \leq \tilde{s},$$

$$\phi_{\boldsymbol{\lambda}}(\boldsymbol{\beta}^0) - \phi_{\boldsymbol{\lambda}}(\boldsymbol{\beta}^*) \leq \frac{21/2}{\rho_- - \xi}\|\boldsymbol{\lambda}_{S^*}\|_2^2,$$

$$\lambda_j \geq 8\left|\left[\nabla L_n(\boldsymbol{\beta}^*)\right]_j\right|,$$

$$\left(\frac{121}{\alpha_{\min}(\rho_- - \xi)} + 144\left(\frac{\rho_+}{\rho_- - \xi}\right)^2\right)\left(\frac{\|\boldsymbol{\lambda}_{S^*}\|_2}{\min(\boldsymbol{\lambda}_{\overline{S}^*})}\right)^2 \leq \tilde{s},$$

*where $\kappa := \frac{\rho_+}{\rho_- - \xi}$. Then for all $k = 1, \cdots, K$,*

$$\|(\boldsymbol{\beta}^k)_{\overline{S}^*}\|_0 \leq \tilde{s},$$

*and*

$$\phi_{\boldsymbol{\lambda}}(\boldsymbol{\beta}^k) - \phi_{\boldsymbol{\lambda}}(\boldsymbol{\beta}^*) \leq \frac{21/2}{\rho_- - \xi}\|\boldsymbol{\lambda}_{S^*}\|_2^2.$$

*Proof.* Recall that $\boldsymbol{\beta}^k = \mathcal{T}_{\alpha\boldsymbol{\lambda}}\left(\bar{\boldsymbol{\beta}}^k\right)$ where

$$\bar{\boldsymbol{\beta}}^k = \boldsymbol{\beta}^{k-1} - \alpha\nabla L_{\boldsymbol{\lambda}}(\boldsymbol{\beta}^{k-1})$$

$$= \boldsymbol{\beta}^{k-1} - \alpha\nabla L_{\boldsymbol{\lambda}}(\boldsymbol{\beta}^*) + \alpha\left(\nabla L_{\boldsymbol{\lambda}}(\boldsymbol{\beta}^*) - \nabla L_{\boldsymbol{\lambda}}(\boldsymbol{\beta}^{k-1})\right).$$

To show $\left\|(\boldsymbol{\beta}^k)_{\overline{S^*}}\right\|_0 \leq \tilde{s}$, we need to prove that, for $j \in \overline{S^*}$, the number of $j$'s such that $|\bar{\beta}_j^k| > \alpha \lambda_j$ is no more than $\tilde{s}$. Note that

$$
|\bar{\beta}_j^k| \leq |\beta_j^{k-1}| + \alpha \left| \left(\nabla L_{\boldsymbol{\lambda}}(\boldsymbol{\beta}^*) - \nabla L_{\boldsymbol{\lambda}}(\boldsymbol{\beta}^{k-1})\right)_j \right| + \alpha \left| [\nabla L_n(\boldsymbol{\beta}^*)]_j \right|
$$

$$
\leq |\beta_j^{k-1}| + \alpha \left| \left(\nabla L_{\boldsymbol{\lambda}}(\boldsymbol{\beta}^*) - \nabla L_{\boldsymbol{\lambda}}(\boldsymbol{\beta}^{k-1})\right)_j \right| + \frac{1}{8} \cdot \alpha \lambda_j.
$$

Therefore, $\forall c \in [0, 1]$,

$$
\left\|(\boldsymbol{\beta}^k)_{\overline{S^*}}\right\|_0 \leq \mathrm{Card}\left(\{j \in \overline{S^*} : |\beta_j^{k-1}| > c\frac{7}{8}\alpha \lambda_j\}\right) \tag{59}
$$

$$
+ \mathrm{Card}\left(\{j \in \overline{S^*} : \left|\left(\nabla L_{\boldsymbol{\lambda}}(\boldsymbol{\beta}^*) - \nabla L_{\boldsymbol{\lambda}}(\boldsymbol{\beta}^{k-1})\right)_j\right| > (1-c) \cdot \frac{7}{8}\lambda_j\}\right), \tag{60}
$$

where $\mathrm{Card}\,()$ represents the size of the set. For the term in Eq. 59,

$$
\mathrm{Card}\left(\{j \in \overline{S^*} : |\beta_j^{k-1}| > c\frac{7}{8}\alpha \lambda_j\}\right) \leq \sum_{j \in \overline{S^*}} \frac{|\beta_j^{k-1}|}{c\frac{7}{8}\alpha \lambda_j} = \frac{8}{c \cdot 7\alpha} \sum_{j \in \overline{S^*}} \frac{|\beta_j^{k-1}|}{\lambda_j}
$$

$$
\leq \frac{8}{c \cdot 7\alpha \min(\boldsymbol{\lambda}_{\overline{S^*}})^2} \|(\boldsymbol{\beta}^{k-1})_{\overline{S^*}}\|_{\boldsymbol{\lambda}_{\overline{S^*}},1}
$$

$$
= \frac{8}{c \cdot 7\alpha \min(\boldsymbol{\lambda}_{\overline{S^*}})^2} \|(\boldsymbol{\beta}^{k-1} - \boldsymbol{\beta}^*)_{\overline{S^*}}\|_{\boldsymbol{\lambda}_{\overline{S^*}},1}
$$

**Bounding** $\|(\boldsymbol{\beta}^{k-1} - \boldsymbol{\beta}^*)_{\overline{S^*}}\|_1$. Following the derivation of Eq. 58, we can obtain

$$
(\rho_- - \xi)\|\boldsymbol{\beta}^{k-1} - \boldsymbol{\beta}^*\|_2^2
$$
$$
\leq \frac{21}{(\rho_- - \xi)}\|\boldsymbol{\lambda}_{S^*}\|_2^2 + \frac{17}{4}\|(\boldsymbol{\beta}^{k-1} - \boldsymbol{\beta}^*)_{S^*}\|_{\boldsymbol{\lambda}_{S^*},1} - \frac{7}{4}\|(\boldsymbol{\beta}^{k-1} - \boldsymbol{\beta}^*)_{\overline{S^*}}\|_{\boldsymbol{\lambda}_{\overline{S^*}},1},
$$

which implies

$$
\|(\boldsymbol{\beta}^{k-1} - \boldsymbol{\beta}^*)_{\overline{S^*}}\|_{\boldsymbol{\lambda}_{\overline{S^*}},1} \leq \frac{12}{\rho_- - \xi}\|\boldsymbol{\lambda}_{S^*}\|_2^2 + \frac{17}{7}\|(\boldsymbol{\beta}^{k-1} - \boldsymbol{\beta}^*)_{S^*}\|_{\boldsymbol{\lambda}_{S^*},1}.
$$

Let $a \geq 0$ be a constant. If $\|(\boldsymbol{\beta}^{k-1} - \boldsymbol{\beta}^*)_{S^*}\|_{\boldsymbol{\lambda}_{S^*},1} \leq \frac{a}{\rho_- - \xi}\|\boldsymbol{\lambda}_{S^*}\|_2^2$, then

$$
\|(\boldsymbol{\beta}^{k-1} - \boldsymbol{\beta}^*)_{\overline{S^*}}\|_{\boldsymbol{\lambda}_{\overline{S^*}},1} \leq \left(12 + \frac{17a}{7}\right)\frac{\|\boldsymbol{\lambda}_{S^*}\|_2^2}{\rho_- - \xi}.
$$

If $\|(\boldsymbol{\beta}^{k-1} - \boldsymbol{\beta}^*)_{S^*}\|_{\boldsymbol{\lambda}_{S^*},1} > \frac{a}{\rho_- - \xi}\|\boldsymbol{\lambda}_{S^*}\|_2^2$, then

$$
\|(\boldsymbol{\beta}^{k-1} - \boldsymbol{\beta}^*)_{\overline{S^*}}\|_{\boldsymbol{\lambda}_{\overline{S^*}},1} \leq \left(\frac{12}{a} + \frac{17}{7}\right)\|(\boldsymbol{\beta}^{k-1} - \boldsymbol{\beta}^*)_{S^*}\|_{\boldsymbol{\lambda}_{S^*},1}
$$

$$
\leq \left(\frac{12}{a} + \frac{17}{7}\right)\|\boldsymbol{\lambda}_{S^*}\|_2\|(\boldsymbol{\beta}^{k-1} - \boldsymbol{\beta}^*)\|_2
$$

$$
(\text{Lemma G.3}) \leq \frac{15}{2}\left(\frac{12}{a} + \frac{17}{7}\right)\frac{\|\boldsymbol{\lambda}_{S^*}\|_2^2}{\rho_- - \xi}.
$$

Combining the above two cases yields

$$
\|(\boldsymbol{\beta}^{k-1} - \boldsymbol{\beta}^*)_{\overline{S^*}}\|_{\boldsymbol{\lambda}_{\overline{S^*}},1} \leq \min_{a \geq 0} \max\left\{12 + \frac{17a}{7}, \frac{15}{2}\left(\frac{12}{a} + \frac{17}{7}\right)\right\}\frac{\|\boldsymbol{\lambda}_{S^*}\|_2^2}{\rho_- - \xi}
$$

$$
\leq \left(\frac{423}{14}\right)\frac{\|\boldsymbol{\lambda}_{S^*}\|_2^2}{\rho_- - \xi},
$$

where the last inequality is obtained by taking $a = \frac{255}{34}$. Therefore, the term in Eq. 59 can be bounded by

$$
\begin{aligned}
\mathrm{Card}\left(\{j \in \overline{S^*} : |\beta_j^{k-1}| > c\frac{7}{8}\alpha\lambda_j\}\right) &\leq \frac{8\|(\beta^{k-1} - \beta^*)_{\overline{S^*}}\|_{\lambda_{\overline{S^*}},1}}{c \cdot 7\alpha \min(\lambda_{\overline{S^*}})^2} \\
&\leq \left(\frac{1692}{49c\alpha \min(\lambda_{\overline{S^*}})^2}\right)\frac{\|\lambda_{S^*}\|_2^2}{\rho_- - \xi} \\
&\leq \frac{1692}{49c\alpha(\rho_- - \xi)}\left(\frac{\|\lambda_{S^*}\|_2)}{\min(\lambda_{\overline{S^*}})}\right)^2.
\end{aligned}
$$

Now we bound the term in Eq. 60 as follows. Denote the set as $S' := \{j \in \overline{S^*} : \left|(\nabla L_\lambda(\beta^*) - \nabla L_\lambda(\beta^{k-1}))_j\right| > (1-c) \cdot \frac{7}{8}\lambda_j\}$ and its size as $s' := |S'|$. Then

$$
\begin{aligned}
s' &= \mathrm{Card}\left(\left\{j \in \overline{S^*} : \left|(\nabla L_\lambda(\beta^*) - \nabla L_\lambda(\beta^{k-1}))_j\right| > (1-c) \cdot \frac{7}{8}\lambda_j\right\}\right) \\
&\leq \frac{1}{(1-c) \cdot \frac{7}{8}}\sum_{j \in S'}\frac{\left|(\nabla L_\lambda(\beta^*) - \nabla L_\lambda(\beta^{k-1}))_j\right|}{\lambda_j} \\
&\leq \frac{1}{(1-c) \cdot \frac{7}{8}}\sum_{j \in S'}\frac{\left|(\nabla L_\lambda(\beta^*) - \nabla L_\lambda(\beta^{k-1}))_j\right|}{\min(\lambda_{\overline{S^*}})} \\
&\leq \frac{1}{(1-c) \cdot \frac{7}{8}}\frac{\sqrt{s'}}{\min(\lambda_{\overline{S^*}})}\left\|\nabla L_\lambda(\beta^*) - \nabla L_\lambda(\beta^{k-1})\right\|_2 \\
&\leq \frac{1}{(1-c) \cdot \frac{7}{8}}\frac{\sqrt{s'}}{\min(\lambda_{\overline{S^*}})}(\rho_+)\|\beta^* - \beta^{k-1}\|_2.
\end{aligned}
$$

The last inequality holds because $L_\lambda$ is $(\rho_+)$-smooth in the restricted subspace by Lemma B.1. Applying Lemma G.3, we have

$$
\begin{aligned}
s' &\leq \frac{64(\rho_+)^2}{49(1-c)^2 \min(\lambda_{\overline{S^*}})^2}\|\beta^* - \beta^{k-1}\|_2^2 \\
&\leq \frac{64(\rho_+)^2}{49(1-c)^2 \min(\lambda_{\overline{S^*}})^2}\left(\frac{15/2\|\lambda_{S^*}\|_2}{\rho_- - \xi}\right)^2 \leq \frac{3600}{49(1-c)^2}\left(\frac{\rho_+}{\rho_- - \xi}\right)^2\left(\frac{\|\lambda_{S^*}\|_2}{\min(\lambda_{\overline{S^*}})}\right)^2.
\end{aligned}
$$

Combining the above bounds for the terms in Eq. 59 and Eq. 60, we have

$$
\begin{aligned}
\left\|(\beta^k)_{\overline{S^*}}\right\|_0 &\leq \min_{c \in (0,1)}\left(\left(\frac{1692}{49c\alpha}\right)\frac{1}{\rho_- - \xi} + \frac{3600}{49(1-c)^2}\left(\frac{\rho_+}{\rho_- - \xi}\right)^2\right)\left(\frac{\|\lambda_{S^*}\|_2}{\min(\lambda_{\overline{S^*}})}\right)^2 \\
&\leq \left(\frac{121}{\alpha_{\min}(\rho_- - \xi)} + 144\left(\frac{\rho_+}{\rho_- - \xi}\right)^2\right)\left(\frac{\|\lambda_{S^*}\|_2}{\min(\lambda_{\overline{S^*}})}\right)^2 \\
&\leq \tilde{s}
\end{aligned}
$$

The second last inequality is obtained by taking $c = \frac{5}{7}$.

Finally, if we can show $\phi_\lambda(\beta^k) - \phi_\lambda(\beta^*) \leq \frac{21/2}{\rho_- - \xi}\|\lambda_{S^*}\|_2^2$, then we can show by math induction that $\left\|(\beta^k)_{\overline{S^*}}\right\|_0 \leq \tilde{s}$ for all $k$.

Taking $z = \beta^{k-1}$ in Eq. 53, we have

$$
\phi_\lambda(\beta^k) \leq \phi_\lambda(\beta^{k-1}) - \frac{2\frac{1}{s} - \rho_+}{2}\left\|\beta^k - \beta^{k-1}\right\|_2^2 \leq \phi_\lambda(\beta^{k-1}).
$$

Therefore,

$$
\phi_\lambda(\beta^k) - \phi_\lambda(\beta^*) \leq \phi_\lambda(\beta^{k-1}) - \phi_\lambda(\beta^*) \leq \frac{21/2}{\rho_- - \xi}\|\lambda_{S^*}\|_2^2.
$$

$\square$

**Lemma G.5** (Statistical error). *Assume Assumption G.1. If*

$$\omega_{\boldsymbol{\lambda}}(\boldsymbol{\beta}) \leq \epsilon, \quad \text{with } \epsilon \leq 1/2,$$
$$\|\boldsymbol{\beta} - \boldsymbol{\beta}^*\|_0 \leq s^* + 2\tilde{s},$$
$$\lambda_j \geq 8 \left| [L_n(\boldsymbol{\beta}^*)]_j \right|,$$

*then the following relations hold true.*

$$\|\boldsymbol{\beta} - \boldsymbol{\beta}^*\|_2 \leq \frac{\left(\epsilon + \frac{17}{8}\right) \|\boldsymbol{\lambda}_{S^*}\|_2}{\rho_- - \xi} \leq \frac{21/8}{\rho_- - \xi} \|\boldsymbol{\lambda}_{S^*}\|_2,$$

$$\phi_{\boldsymbol{\lambda}}(\boldsymbol{\beta}) - \phi_{\boldsymbol{\lambda}}(\boldsymbol{\beta}^*) \leq \frac{3\epsilon(\epsilon + 17/8)}{7/8 - \epsilon} \frac{\|\boldsymbol{\lambda}_{S^*}\|_2^2}{\rho_- - \xi} \leq \frac{21/2}{\rho_- - \xi} \|\boldsymbol{\lambda}_{S^*}\|_2^2.$$

$$\phi_{\boldsymbol{\lambda}}(\boldsymbol{\beta}) - \phi_{\boldsymbol{\lambda}}(\widehat{\boldsymbol{\beta}}_{\boldsymbol{\lambda}}) \leq \epsilon \left( \frac{3(\epsilon + 17/8)}{7/8 - \epsilon} + 51/7 \right) \frac{\|\boldsymbol{\lambda}_{S^*}\|_2^2}{\rho_- - \xi} \leq \frac{29/2}{\rho_- - \xi} \|\boldsymbol{\lambda}_{S^*}\|_2^2.$$

*Proof.* Since $\|\boldsymbol{\beta} - \boldsymbol{\beta}^*\|_0 \leq s^* + 2\tilde{s}$, then by the RSC and RSS in Lemma B.1,

$$(\rho_- - \xi)\|\boldsymbol{\beta}^* - \boldsymbol{\beta}\|_2^2 \leq (\boldsymbol{\beta} - \boldsymbol{\beta}^*)^\top \nabla L_{\boldsymbol{\lambda}}(\boldsymbol{\beta}) - (\boldsymbol{\beta} - \boldsymbol{\beta}^*)^\top \nabla L_{\boldsymbol{\lambda}}(\boldsymbol{\beta}^*). \tag{61}$$

Let $\boldsymbol{\xi} \in \partial\|\boldsymbol{\beta}\|_1$ be the subgradient that attains the minimum in $\omega_{\boldsymbol{\lambda}}(\boldsymbol{\beta})$. Adding and subtracting $(\boldsymbol{\beta} - \boldsymbol{\beta}^*)^\top (\boldsymbol{\lambda} \circ \boldsymbol{\xi})$ to the right-hand side of Eq. 61 yields

$$(\rho_- - \xi)\|\boldsymbol{\beta}^* - \boldsymbol{\beta}\|_2^2 \leq \underbrace{(\boldsymbol{\beta} - \boldsymbol{\beta}^*)^\top (\nabla L_{\boldsymbol{\lambda}}(\boldsymbol{\beta}) + \boldsymbol{\lambda} \circ \boldsymbol{\xi})}_{\text{(i)}} - \underbrace{(\boldsymbol{\beta} - \boldsymbol{\beta}^*)^\top \nabla L_{\boldsymbol{\lambda}}(\boldsymbol{\beta}^*)}_{\text{(ii)}}$$
$$- \underbrace{(\boldsymbol{\beta} - \boldsymbol{\beta}^*)^\top (\boldsymbol{\lambda} \circ \boldsymbol{\xi})}_{\text{(iii)}}.$$

Now we bound the terms (i)-(iii).

**Bound (i) using $\omega_{\boldsymbol{\lambda}}(\boldsymbol{\beta})$.** Since $\boldsymbol{\xi}$ attains the minimum in $\omega_{\boldsymbol{\lambda}}(\boldsymbol{\beta})$, by definition,

$$\omega_{\boldsymbol{\lambda}}(\boldsymbol{\beta}) = \max_{\boldsymbol{\beta}' \in \Omega} \left\{ \frac{(\boldsymbol{\beta} - \boldsymbol{\beta}')^\top}{\|\boldsymbol{\beta} - \boldsymbol{\beta}'\|_{\boldsymbol{\lambda},1}} (\nabla L_{\boldsymbol{\lambda}}(\boldsymbol{\beta}) + \lambda \boldsymbol{\xi}) \right\}.$$

Therefore,

$$\text{(i)} = (\boldsymbol{\beta} - \boldsymbol{\beta}^*)^\top (\nabla L_{\boldsymbol{\lambda}}(\boldsymbol{\beta}) + \lambda \boldsymbol{\xi}) \leq \omega_{\boldsymbol{\lambda}}(\boldsymbol{\beta})\|\boldsymbol{\beta} - \boldsymbol{\beta}^*\|_{\boldsymbol{\lambda},1} \leq \epsilon\|\boldsymbol{\beta} - \boldsymbol{\beta}^*\|_{\boldsymbol{\lambda},1}.$$

**Bound (ii).** Note that $L_{\boldsymbol{\lambda}} = L_n + Q_{\boldsymbol{\lambda}}$. Thus,

$$|\text{(ii)}| = \left| (\boldsymbol{\beta} - \boldsymbol{\beta}^*)^\top \nabla L_n(\boldsymbol{\beta}^*) + (\boldsymbol{\beta} - \boldsymbol{\beta}^*)^\top \nabla Q_{\boldsymbol{\lambda}}(\boldsymbol{\beta}^*) \right|$$
$$\leq \left| (\boldsymbol{\beta} - \boldsymbol{\beta}^*)^\top \nabla L_n(\boldsymbol{\beta}^*) \right| + \left| (\boldsymbol{\beta} - \boldsymbol{\beta}^*)^\top \nabla Q_{\boldsymbol{\lambda}}(\boldsymbol{\beta}^*) \right|$$

Since $\lambda_j \geq 8 \left| [L_n(\boldsymbol{\beta}^*)]_j \right|$, we have

$$|(\boldsymbol{\beta} - \boldsymbol{\beta}^*)^\top \nabla L_n(\boldsymbol{\beta}^*)| \leq \frac{1}{8}\|\boldsymbol{\beta} - \boldsymbol{\beta}^*\|_{\boldsymbol{\lambda},1}.$$

Since $q'_{\lambda_j}(0) = 0$ and $q'_{\lambda_j}(\beta) \leq \lambda_j$ (Assumption B.1), it holds true that

$$|(\boldsymbol{\beta} - \boldsymbol{\beta}^*)^\top \nabla Q_{\boldsymbol{\lambda}}(\boldsymbol{\beta}^*)| = |(\boldsymbol{\beta} - \boldsymbol{\beta}^*)_{S^*}^\top (\nabla Q_{\boldsymbol{\lambda}}(\boldsymbol{\beta}^*))_{S^*}|$$
$$\leq \|(\boldsymbol{\beta} - \boldsymbol{\beta}^*)_{S^*}\|_{\boldsymbol{\lambda}_{S^*},1}.$$

Therefore,

$$|\text{(ii)}| \leq \frac{1}{8}\|\boldsymbol{\beta} - \boldsymbol{\beta}^*\|_{\boldsymbol{\lambda},1} + \|(\boldsymbol{\beta} - \boldsymbol{\beta}^*)_{S^*}\|_{\boldsymbol{\lambda}_{S^*},1}.$$

**Bound (iii) by the definition of subgradient.**

$$\text{(iii)} = (\boldsymbol{\beta} - \boldsymbol{\beta}^*)^\top (\boldsymbol{\lambda} \circ \boldsymbol{\xi})$$
$$= \langle \boldsymbol{\lambda}_{S^*} \circ (\boldsymbol{\beta} - \boldsymbol{\beta}^*)_{S^*}, \boldsymbol{\xi}_{S^*} \rangle + \langle \boldsymbol{\lambda}_{\overline{S^*}} \circ (\boldsymbol{\beta} - \boldsymbol{\beta}^*)_{\overline{S^*}}, \boldsymbol{\xi}_{\overline{S^*}} \rangle.$$

By Holder's inequality and the fact $\|\boldsymbol{\xi}\|_\infty \leq 1$,

$$\langle \boldsymbol{\lambda}_{S^*} \circ (\boldsymbol{\beta} - \boldsymbol{\beta}^*)_{S^*}, \boldsymbol{\xi}_{S^*} \rangle \geq -\|(\boldsymbol{\beta} - \boldsymbol{\beta}^*)_{S^*}\|_{\boldsymbol{\lambda}_{S^*},1}.$$

Since $\boldsymbol{\xi} \in \partial \|\boldsymbol{\beta}\|_1$ and that $\boldsymbol{\beta}_{\overline{S^*}}^* = \mathbf{0}$,

$$\langle \boldsymbol{\lambda}_{\overline{S^*}} \circ (\boldsymbol{\beta} - \boldsymbol{\beta}^*)_{\overline{S^*}}, \boldsymbol{\xi}_{\overline{S^*}} \rangle = \langle \boldsymbol{\lambda}_{\overline{S^*}} \circ \boldsymbol{\beta}_{\overline{S^*}}, \boldsymbol{\xi}_{\overline{S^*}} \rangle = \|\boldsymbol{\beta}_{\overline{S^*}}\|_{\boldsymbol{\lambda}_{\overline{S^*}},1} = \|(\boldsymbol{\beta} - \boldsymbol{\beta}^*)_{\overline{S^*}}\|_{\boldsymbol{\lambda}_{\overline{S^*}},1}. \qquad (62)$$

Combining the above three equations, it holds true that

$$(\text{iii}) \geq -\|(\boldsymbol{\beta} - \boldsymbol{\beta}^*)_{S^*}\|_{\boldsymbol{\lambda}_{S^*},1} + \|(\boldsymbol{\beta} - \boldsymbol{\beta}^*)_{\overline{S^*}}\|_{\boldsymbol{\lambda}_{\overline{S^*}},1}.$$

**Combine (i)-(iii).** Combining the bounds for (i)-(iii), it holds true that

$$(\rho_- - \xi)\|\boldsymbol{\beta}^* - \boldsymbol{\beta}\|_2^2$$
$$\leq \left( \epsilon + \frac{1}{8} + 1 + 1 \right) \|(\boldsymbol{\beta} - \boldsymbol{\beta}^*)_{S^*}\|_{\boldsymbol{\lambda}_{S^*},1} - \left( 1 - \epsilon - \frac{1}{8} \right) \|(\boldsymbol{\beta} - \boldsymbol{\beta}^*)_{\overline{S^*}}\|_{\boldsymbol{\lambda}_{\overline{S^*}},1}$$
$$= \left( \epsilon + \frac{17}{8} \right) \|(\boldsymbol{\beta} - \boldsymbol{\beta}^*)_{S^*}\|_{\boldsymbol{\lambda}_{S^*},1} - \underbrace{\left( \frac{7}{8} - \epsilon \right)}_{\geq 0} \|(\boldsymbol{\beta} - \boldsymbol{\beta}^*)_{\overline{S^*}}\|_{\boldsymbol{\lambda}_{\overline{S^*}},1} \qquad (63)$$
$$\leq \left( \epsilon + \frac{17}{8} \right) \|(\boldsymbol{\beta} - \boldsymbol{\beta}^*)_{S^*}\|_{\boldsymbol{\lambda}_{S^*},1}.$$

Using the fact that $\|(\boldsymbol{\beta} - \boldsymbol{\beta}^*)_{S^*}\|_{\boldsymbol{\lambda}_{S^*},1} \leq \|\boldsymbol{\lambda}_{S^*}\|_2\|(\boldsymbol{\beta} - \boldsymbol{\beta}^*)_{S^*}\|_2$, it implies the first conclusion:

$$\|\boldsymbol{\beta}^* - \boldsymbol{\beta}\|_2 \leq \frac{\left( \epsilon + \frac{17}{8} \right) \|\boldsymbol{\lambda}_{S^*}\|_2}{\rho_- - \xi}. \qquad (64)$$

Now we prove the objective value. Since $L_{\boldsymbol{\lambda}}$ is convex on $\boldsymbol{\beta}, \boldsymbol{\beta}^*$ and $\|\cdot\|_1$ is convex,

$$\phi_{\boldsymbol{\lambda}}(\boldsymbol{\beta}^*) \geq \phi_{\boldsymbol{\lambda}}(\boldsymbol{\beta}) + (\boldsymbol{\beta}^* - \boldsymbol{\beta})^\top (\nabla L_{\boldsymbol{\lambda}}(\boldsymbol{\beta}) + \boldsymbol{\lambda} \circ \boldsymbol{\xi}),$$

which implies

$$\phi_{\boldsymbol{\lambda}}(\boldsymbol{\beta}) - \phi_{\boldsymbol{\lambda}}(\boldsymbol{\beta}^*) \leq (\boldsymbol{\beta} - \boldsymbol{\beta}^*)^\top (\nabla L_{\boldsymbol{\lambda}}(\boldsymbol{\beta}) + \boldsymbol{\lambda} \circ \boldsymbol{\xi})$$
$$\leq \|\boldsymbol{\beta} - \boldsymbol{\beta}^*\|_{\boldsymbol{\lambda},1} \omega_{\boldsymbol{\lambda}}(\boldsymbol{\beta}) \leq \epsilon \|\boldsymbol{\beta} - \boldsymbol{\beta}^*\|_{\boldsymbol{\lambda},1} \qquad (65)$$

Now we bound the norm $\|\boldsymbol{\beta} - \boldsymbol{\beta}^*\|_{\boldsymbol{\lambda},1}$ as follows. By triangle inequality,

$$\|\boldsymbol{\beta} - \boldsymbol{\beta}^*\|_{\boldsymbol{\lambda},1} \leq \|(\boldsymbol{\beta} - \boldsymbol{\beta}^*)_{S^*}\|_{\boldsymbol{\lambda}_{S^*},1} + \|(\boldsymbol{\beta} - \boldsymbol{\beta}^*)_{\overline{S^*}}\|_{\boldsymbol{\lambda}_{\overline{S^*}},1}. \qquad (66)$$

To bound $\|(\boldsymbol{\beta} - \boldsymbol{\beta}^*)_{\overline{S^*}}\|_{\boldsymbol{\lambda}_{\overline{S^*}},1}$, moving the term $\|(\boldsymbol{\beta} - \boldsymbol{\beta}^*)_{\overline{S^*}}\|_{\boldsymbol{\lambda}_{\overline{S^*}},1}$ in Eq. 63 to the left hand side yields

$$(7/8 - \epsilon)\|(\boldsymbol{\beta} - \boldsymbol{\beta}^*)_{\overline{S^*}}\|_{\boldsymbol{\lambda}_{\overline{S^*}},1} \leq (\epsilon + 17/8)\|(\boldsymbol{\beta} - \boldsymbol{\beta}^*)_{S^*}\|_{\boldsymbol{\lambda}_{S^*},1},$$

which implies

$$\|(\boldsymbol{\beta} - \boldsymbol{\beta}^*)_{\overline{S^*}}\|_{\boldsymbol{\lambda}_{\overline{S^*}},1} \leq \frac{\epsilon + 17/8}{7/8 - \epsilon} \|(\boldsymbol{\beta} - \boldsymbol{\beta}^*)_{S^*}\|_{\boldsymbol{\lambda}_{S^*},1}. \qquad (67)$$

Plugging it into Eq. 66, we have

$$\|\boldsymbol{\beta} - \boldsymbol{\beta}^*\|_{\boldsymbol{\lambda},1} \leq \left( 1 + \frac{\epsilon + 17/8}{7/8 - \epsilon} \right) \|(\boldsymbol{\beta} - \boldsymbol{\beta}^*)_{S^*}\|_{\boldsymbol{\lambda}_{S^*},1} \leq \left( \frac{3}{7/8 - \epsilon} \right) \|\boldsymbol{\lambda}_{S^*}\|_2 \|\boldsymbol{\beta} - \boldsymbol{\beta}^*\|_2$$
$$(\text{by Eq. } 64) \leq \left( \frac{3}{7/8 - \epsilon} \right) \|\boldsymbol{\lambda}_{S^*}\|_2 \frac{\left( \epsilon + \frac{17}{8} \right) \|\boldsymbol{\lambda}_{S^*}\|_2}{\rho_- - \xi}$$
$$= \frac{3(\epsilon + 17/8)}{7/8 - \epsilon} \frac{\|\boldsymbol{\lambda}_{S^*}\|_2^2}{\rho_- - \xi}. \qquad (68)$$

Combining it with Eq. 65, we have

$$\phi_{\boldsymbol{\lambda}}(\boldsymbol{\beta}) - \phi_{\boldsymbol{\lambda}}(\boldsymbol{\beta}^*) \leq \frac{3\epsilon(\epsilon + 17/8)}{7/8 - \epsilon} \frac{\|\boldsymbol{\lambda}_{S^*}\|_2^2}{\rho_- - \xi}.$$

Finally, we derive the bound for the term $\phi_{\boldsymbol{\lambda}}(\boldsymbol{\beta}) - \phi_{\boldsymbol{\lambda}}(\widehat{\boldsymbol{\beta}}_{\boldsymbol{\lambda}})$. Since $\|\boldsymbol{\beta} - \widehat{\boldsymbol{\beta}}_{\boldsymbol{\lambda}}\|_0 \leq s^* + 2\tilde{s}$ and $\phi_{\boldsymbol{\lambda}}$ is convex on the restricted space, then

$$
\begin{aligned}
\phi_{\boldsymbol{\lambda}}(\boldsymbol{\beta}) - \phi_{\boldsymbol{\lambda}}(\widehat{\boldsymbol{\beta}}_{\boldsymbol{\lambda}}) &\leq (\nabla L_{\boldsymbol{\lambda}}(\boldsymbol{\beta}) + \boldsymbol{\lambda} \circ \boldsymbol{\xi})^\top (\boldsymbol{\beta} - \widehat{\boldsymbol{\beta}}_{\boldsymbol{\lambda}}) \\
&\leq \omega_{\boldsymbol{\lambda}}(\boldsymbol{\beta}) \|\boldsymbol{\beta} - \widehat{\boldsymbol{\beta}}_{\boldsymbol{\lambda}}\|_{\boldsymbol{\lambda},1} \\
&\leq \epsilon \left( \|\boldsymbol{\beta} - \boldsymbol{\beta}^*\|_{\boldsymbol{\lambda},1} + \|\boldsymbol{\beta}^* - \widehat{\boldsymbol{\beta}}_{\boldsymbol{\lambda}}\|_{\boldsymbol{\lambda},1} \right) \\
\text{(by Eq. 68)} \quad &\leq \epsilon \left( \frac{3(\epsilon + 17/8)}{7/8 - \epsilon} \frac{\|\boldsymbol{\lambda}_{S^*}\|_2^2}{\rho_- - \xi} + \|\boldsymbol{\beta}^* - \widehat{\boldsymbol{\beta}}_{\boldsymbol{\lambda}}\|_{\boldsymbol{\lambda},1} \right).
\end{aligned}
\tag{69}
$$

Now we only need to bound the term $\|\boldsymbol{\beta}^* - \widehat{\boldsymbol{\beta}}_{\boldsymbol{\lambda}}\|_{\boldsymbol{\lambda},1}$. We can derive its bound following the same steps as we derive Eq. 68 and using the fact that $\omega_{\boldsymbol{\lambda}}(\widehat{\boldsymbol{\beta}}_{\boldsymbol{\lambda}}) = 0$, which will imply that

$$
\|\widehat{\boldsymbol{\beta}}_{\boldsymbol{\lambda}} - \boldsymbol{\beta}^*\|_1 \leq \frac{3(0 + 17/8)}{7/8 - 0} \frac{\|\boldsymbol{\lambda}_{S^*}\|_2^2}{\rho_- - \xi} = \frac{51/7 \|\boldsymbol{\lambda}_{S^*}\|_2^2}{\rho_- - \xi}.
$$

Plugging it into Eq. 69, we have

$$
\begin{aligned}
&\phi_{\boldsymbol{\lambda}}(\boldsymbol{\beta}) - \phi_{\boldsymbol{\lambda}}(\widehat{\boldsymbol{\beta}}_{\boldsymbol{\lambda}}) \\
&\leq \epsilon \left( \frac{3(\epsilon + 17/8)}{7/8 - \epsilon} + 51/7 \right) \frac{\|\boldsymbol{\lambda}_{S^*}\|_2^2}{\rho_- - \xi}.
\end{aligned}
$$

Assuming $\epsilon \leq 1/2$, then

$$
\phi_{\boldsymbol{\lambda}}(\boldsymbol{\beta}) - \phi_{\boldsymbol{\lambda}}(\widehat{\boldsymbol{\beta}}_{\boldsymbol{\lambda}}) \leq \frac{29/2 \|\boldsymbol{\lambda}_{S^*}\|_2^2}{\rho_- - \xi}.
$$

$\square$

**Lemma G.6** (Optimality when $\lambda$ decreases). *Assume Assumption G.1. If*

$$
\omega_{\boldsymbol{\lambda}_{t-1}}(\boldsymbol{\beta}) \leq \epsilon_{t-1}, \tag{70}
$$
$$
\boldsymbol{\lambda}_t = \boldsymbol{\eta} \circ \boldsymbol{\lambda}_{t-1} \quad \text{with} \quad \eta_j \in [0.9, 1], \tag{71}
$$
$$
\|\boldsymbol{\beta}\|_0 \leq \tilde{s}, \tag{72}
$$

*then*

$$
\omega_{\boldsymbol{\lambda}_t}(\boldsymbol{\beta}) \leq \frac{\epsilon_{t-1} + 0.2}{0.9}
$$
$$
\phi_{\boldsymbol{\lambda}_t}(\boldsymbol{\beta}) - \phi_{\boldsymbol{\lambda}_t}(\widehat{\boldsymbol{\beta}}_{\boldsymbol{\lambda}_t}) \leq \left( \frac{3(\epsilon_{t-1} + 17/8)}{0.9(7/8 - \epsilon_{t-1})} + 51/7 \right) \frac{\epsilon_{t-1} + 0.2}{0.9} \frac{\|(\boldsymbol{\lambda}_t)_{S^*}\|_2^2}{\rho_- - \xi}.
$$

*If $\epsilon_{t-1} \leq 1/4$, then*

$$
\omega_{\boldsymbol{\lambda}_t}(\boldsymbol{\beta}) \leq \frac{1}{2}
$$
$$
\phi_{\boldsymbol{\lambda}_t}(\boldsymbol{\beta}) - \phi_{\boldsymbol{\lambda}_t}(\widehat{\boldsymbol{\beta}}_{\boldsymbol{\lambda}_t}) \leq \frac{10 \|(\boldsymbol{\lambda}_t)_{S^*}\|_2^2}{\rho_- - \xi}.
$$

*Proof.* Recall the definition of $\omega_{\boldsymbol{\lambda}_{t-1}}(\boldsymbol{\beta})$.

$$
\omega_{\boldsymbol{\lambda}_{t-1}}(\boldsymbol{\beta}) = \min_{\boldsymbol{\xi}' \in \partial\|\boldsymbol{\beta}\|_1} \max_{\boldsymbol{\beta}' \in \Omega} \left\{ \frac{(\boldsymbol{\beta} - \boldsymbol{\beta}')^\top}{\|\boldsymbol{\beta} - \boldsymbol{\beta}'\|_{\boldsymbol{\lambda}_{t-1},1}} (\nabla L_{\boldsymbol{\lambda}_{t-1}}(\boldsymbol{\beta}) + \boldsymbol{\lambda}_{t-1}\boldsymbol{\xi}') \right\}
$$

Let $\boldsymbol{\xi} \in \partial\|\boldsymbol{\beta}\|_1$ be the subgradient that attains the minimum in $\omega_{\boldsymbol{\lambda}_{t-1}}(\boldsymbol{\beta})$. Then

$$
\omega_{\boldsymbol{\lambda}_{t-1}}(\boldsymbol{\beta}) = \max_{\boldsymbol{\beta}' \in \Omega} \left\{ \frac{(\boldsymbol{\beta} - \boldsymbol{\beta}')^\top}{\|\boldsymbol{\beta} - \boldsymbol{\beta}'\|_{\boldsymbol{\lambda}_{t-1},1}} (\nabla L_{\boldsymbol{\lambda}_{t-1}}(\boldsymbol{\beta}) + \boldsymbol{\lambda}_{t-1}\boldsymbol{\xi}) \right\}.
$$

Now we consider $\boldsymbol{\lambda}_t = \boldsymbol{\eta} \circ \boldsymbol{\lambda}_{t-1}$. By definition,

$$
\begin{aligned}
\omega_{\boldsymbol{\lambda}_t}(\boldsymbol{\beta}) &= \min_{\boldsymbol{\xi}' \in \partial \|\boldsymbol{\beta}\|_1} \max_{\boldsymbol{\beta}' \in \Omega} \left\{ \frac{(\boldsymbol{\beta} - \boldsymbol{\beta}')^\top}{\|\boldsymbol{\beta} - \boldsymbol{\beta}'\|_{\boldsymbol{\lambda}_t,1}} (\nabla L_{\boldsymbol{\lambda}_t}(\boldsymbol{\beta}) + \boldsymbol{\lambda}_t \boldsymbol{\xi}') \right\} \\
&\leq \max_{\boldsymbol{\beta}' \in \Omega} \left\{ \frac{(\boldsymbol{\beta} - \boldsymbol{\beta}')^\top}{\|\boldsymbol{\beta} - \boldsymbol{\beta}'\|_{\boldsymbol{\lambda}_t,1}} (\nabla L_{\boldsymbol{\lambda}_t}(\boldsymbol{\beta}) + \boldsymbol{\lambda}_t \boldsymbol{\xi}) \right\}.
\end{aligned}
$$

Note that

$$
\begin{aligned}
\nabla L_{\boldsymbol{\lambda}_t}(\boldsymbol{\beta}) + \boldsymbol{\lambda}_t \boldsymbol{\xi} &= \left( \nabla L_{\boldsymbol{\lambda}_{t-1}}(\boldsymbol{\beta}) + \boldsymbol{\lambda}_{t-1} \boldsymbol{\xi} \right) \\
&\quad + (\boldsymbol{\lambda}_t - \boldsymbol{\lambda}_{t-1}) \circ \boldsymbol{\xi} \\
&\quad + \left( \nabla Q_{\boldsymbol{\lambda}_t}(\boldsymbol{\beta}) - \nabla Q_{\boldsymbol{\lambda}_{t-1}}(\boldsymbol{\beta}) \right),
\end{aligned}
$$

which implies

$$
\begin{aligned}
\omega_{\boldsymbol{\lambda}_t}(\boldsymbol{\beta}) \leq \max_{\boldsymbol{\beta}' \in \Omega} \underbrace{\left\{ \frac{(\boldsymbol{\beta} - \boldsymbol{\beta}')^\top}{\|\boldsymbol{\beta} - \boldsymbol{\beta}'\|_{\boldsymbol{\lambda}_t,1}} \left( \nabla L_{\boldsymbol{\lambda}_{t-1}}(\boldsymbol{\beta}) + \boldsymbol{\lambda}_{t-1} \boldsymbol{\xi} \right) \right\}}_{(\mathrm{i})} \\
+ \max_{\boldsymbol{\beta}' \in \Omega} \underbrace{\left\{ \frac{(\boldsymbol{\beta} - \boldsymbol{\beta}')^\top}{\|\boldsymbol{\beta} - \boldsymbol{\beta}'\|_{\boldsymbol{\lambda}_t,1}} (\boldsymbol{\lambda}_t - \boldsymbol{\lambda}_{t-1}) \circ \boldsymbol{\xi} \right\}}_{(\mathrm{ii})} \\
+ \max_{\boldsymbol{\beta}' \in \Omega} \underbrace{\left\{ \frac{(\boldsymbol{\beta} - \boldsymbol{\beta}')^\top}{\|\boldsymbol{\beta} - \boldsymbol{\beta}'\|_{\boldsymbol{\lambda}_t,1}} \left( \nabla Q_{\boldsymbol{\lambda}_t}(\boldsymbol{\beta}) - \nabla Q_{\boldsymbol{\lambda}_{t-1}}(\boldsymbol{\beta}) \right) \right\}}_{(\mathrm{iii})}.
\end{aligned}
$$

Using the fact that $\|\boldsymbol{\beta} - \boldsymbol{\beta}'\|_{\boldsymbol{\lambda}_t,1} \geq 0.9 \|\boldsymbol{\beta} - \boldsymbol{\beta}'\|_{\boldsymbol{\lambda}_{t-1},1}$, term (i) can be bounded as follows.

$$
(\mathrm{i}) \leq \frac{1}{0.9} \frac{(\boldsymbol{\beta} - \boldsymbol{\beta}')^\top}{\|\boldsymbol{\beta} - \boldsymbol{\beta}'\|_{\boldsymbol{\lambda}_{t-1},1}} \left( \nabla L_{\boldsymbol{\lambda}_{t-1}}(\boldsymbol{\beta}) + \boldsymbol{\lambda}_{t-1} \boldsymbol{\xi} \right) \leq \frac{1}{0.9} \omega_{\boldsymbol{\lambda}_{t-1}}(\boldsymbol{\beta}) \leq \frac{\epsilon_{t-1}}{0.9}.
$$

Using the fact $|\xi_j| \leq 1$ and the fact that $[\boldsymbol{\lambda}_{t-1} - \boldsymbol{\lambda}_t]_j = \left( \frac{1}{\eta_j} - 1 \right) [\boldsymbol{\lambda}_t]_j \leq \left( \frac{1}{0.9} - 1 \right) [\boldsymbol{\lambda}_t]_j$, term (ii) can be bounded as follows.

$$
(\mathrm{ii}) \leq \sum_{j=1}^d \frac{|\boldsymbol{\beta}_j - \boldsymbol{\beta}'_j|}{\|\boldsymbol{\beta} - \boldsymbol{\beta}'\|_{\boldsymbol{\lambda}_t,1}} \left( \frac{1}{0.9} - 1 \right) [\boldsymbol{\lambda}_t]_j = \frac{\|\boldsymbol{\beta} - \boldsymbol{\beta}'\|_{\boldsymbol{\lambda}_t,1}}{\|\boldsymbol{\beta} - \boldsymbol{\beta}'\|_{\boldsymbol{\lambda}_t,1}} \left( \frac{1}{0.9} - 1 \right) = \frac{1}{0.9} - 1.
$$

Term (iii) is bounded similarly as term (ii). Since $\left| q_{\lambda_j}(\beta) - q_{\lambda'_j}(\beta) \right| \leq \left| \lambda_j - \lambda'_j \right|$, we have

$$
(\mathrm{iii}) \leq \sum_{j=1}^d \frac{|\boldsymbol{\beta}_j - \boldsymbol{\beta}'_j|}{\|\boldsymbol{\beta} - \boldsymbol{\beta}'\|_{\boldsymbol{\lambda}_t,1}} [\boldsymbol{\lambda}_{t-1} - \boldsymbol{\lambda}_t]_j \leq \sum_{j=1}^d \frac{|\boldsymbol{\beta}_j - \boldsymbol{\beta}'_j|}{\|\boldsymbol{\beta} - \boldsymbol{\beta}'\|_{\boldsymbol{\lambda}_t,1}} \left( \frac{1}{0.9} - 1 \right) [\boldsymbol{\lambda}_t]_j = \frac{1}{0.9} - 1.
$$

Combining the bounds for (i)-(iii), we have

$$
\omega_{\boldsymbol{\lambda}_t}(\boldsymbol{\beta}) \leq \frac{\epsilon_{t-1} + 2}{0.9} - 2 = \frac{\epsilon_{t-1} + 0.2}{0.9}
$$

The first part of the proof is finished. Now we bound the term $\phi_{\boldsymbol{\lambda}_t}(\boldsymbol{\beta}) - \phi_{\boldsymbol{\lambda}_t}(\widehat{\boldsymbol{\beta}}_{\boldsymbol{\lambda}_t})$. Since $\|\boldsymbol{\beta} - \widehat{\boldsymbol{\beta}}_{\boldsymbol{\lambda}}\|_0 \leq s^* + 2\tilde{s}$ and $\phi_{\boldsymbol{\lambda}}$ is convex on the restricted space, then

$$
\begin{aligned}
\phi_{\boldsymbol{\lambda}_t}(\boldsymbol{\beta}) &- \phi_{\boldsymbol{\lambda}_t}(\widehat{\boldsymbol{\beta}}_{\boldsymbol{\lambda}_t}) \\
&\leq (\nabla L_{\boldsymbol{\lambda}_t}(\boldsymbol{\beta}) + \boldsymbol{\lambda}_t \circ \boldsymbol{\xi})^\top (\boldsymbol{\beta} - \widehat{\boldsymbol{\beta}}_{\boldsymbol{\lambda}_t}) \\
&\leq \omega_{\boldsymbol{\lambda}_t}(\boldsymbol{\beta}) \|\boldsymbol{\beta} - \widehat{\boldsymbol{\beta}}_{\boldsymbol{\lambda}_t}\|_{\boldsymbol{\lambda}_t,1} \\
&\leq \frac{\epsilon_{t-1} + 0.2}{0.9} \left( \|\boldsymbol{\beta} - \boldsymbol{\beta}^*\|_{\boldsymbol{\lambda}_t,1} + \|\boldsymbol{\beta}^* - \widehat{\boldsymbol{\beta}}_{\boldsymbol{\lambda}_t}\|_{\boldsymbol{\lambda}_t,1} \right)
\end{aligned} \tag{73}
$$

Following the same way we derive Eq. 67, we can obtain that

$$\|(\boldsymbol{\beta} - \boldsymbol{\beta}^*)_{\overline{S^*}}\|_{(\boldsymbol{\lambda}_t)_{\overline{S^*}},1} \le \frac{\epsilon_{t-1} + 17/8}{7/8 - \epsilon_{t-1}} \|(\boldsymbol{\beta} - \boldsymbol{\beta}^*)_{S^*}\|_{(\boldsymbol{\lambda}_t)_{S^*},1}.$$

Therefore,

$$\|\boldsymbol{\beta} - \boldsymbol{\beta}^*\|_{\boldsymbol{\lambda}_t,1} = \|(\boldsymbol{\beta} - \boldsymbol{\beta}^*)_{\overline{S^*}}\|_{(\boldsymbol{\lambda}_t)_{\overline{S^*}},1} + \|(\boldsymbol{\beta} - \boldsymbol{\beta}^*)_{S^*}\|_{(\boldsymbol{\lambda}_t)_{S^*},1} \tag{74}$$

$$\le \frac{3}{7/8 - \epsilon_{t-1}} \|(\boldsymbol{\beta} - \boldsymbol{\beta}^*)_{S^*}\|_{(\boldsymbol{\lambda}_t)_{S^*},1}$$

$$\le \frac{3\|(\boldsymbol{\lambda}_t)_{S^*}\|_2}{7/8 - \epsilon_{t-1}} \|\boldsymbol{\beta} - \boldsymbol{\beta}^*\|_2$$

$$(\text{By Lemma G.5}) \quad \le \frac{3\|(\boldsymbol{\lambda}_t)_{S^*}\|_2}{7/8 - \epsilon_{t-1}} \frac{\left(\epsilon_{t-1} + \frac{17}{8}\right) \|(\boldsymbol{\lambda}_{t-1})_{S^*}\|_2}{\rho_- - \xi}$$

$$= \frac{3\left(\epsilon_{t-1} + 17/8\right)}{0.9(7/8 - \epsilon_{t-1})} \frac{\|(\boldsymbol{\lambda}_t)_{S^*}\|_2^2}{\rho_- - \xi}. \tag{75}$$

Using the same arguments for the term $\|\boldsymbol{\beta}^* - \widehat{\boldsymbol{\beta}}_{\boldsymbol{\lambda}_t}\|_{\boldsymbol{\lambda}_t,1}$ and using the fact that $\omega_{\boldsymbol{\lambda}_t}(\widehat{\boldsymbol{\beta}}_{\boldsymbol{\lambda}_t}) \le 0 \cdot \boldsymbol{\lambda}_t$, we obtain that

$$\|\widehat{\boldsymbol{\beta}}_{\boldsymbol{\lambda}_t} - \boldsymbol{\beta}^*\|_1 \le \frac{3\left(0 + 17/8\right)}{7/8 - 0} \frac{\|(\boldsymbol{\lambda}_t)_{S^*}\|_2^2}{\rho_- - \xi} = \frac{51/7\|(\boldsymbol{\lambda}_t)_{S^*}\|_2^2}{\rho_- - \xi}.$$

Combining this inequality with Eq. 73 and Eq. 75, we have

$$\phi_{\boldsymbol{\lambda}_t}(\boldsymbol{\beta}) - \phi_{\boldsymbol{\lambda}_t}(\widehat{\boldsymbol{\beta}}_{\boldsymbol{\lambda}_t})$$
$$\le \left(\frac{3\left(\epsilon_{t-1} + 17/8\right)}{0.9(7/8 - \epsilon_{t-1})} + 51/7\right) \frac{\epsilon_{t-1} + 0.2}{0.9} \frac{\|(\boldsymbol{\lambda}_t)_{S^*}\|_2^2}{\rho_- - \xi}.$$

Assuming $\epsilon_{t-1} \le 1/4$, then

$$\phi_{\boldsymbol{\lambda}_t}(\boldsymbol{\beta}) - \phi_{\boldsymbol{\lambda}_t}(\widehat{\boldsymbol{\beta}}_{\boldsymbol{\lambda}_t}) \le \frac{10\|(\boldsymbol{\lambda}_t)_{S^*}\|_2^2}{\rho_- - \xi}.$$

$\square$

**Lemma G.7** (Coercivity of the gradient). *[Lemma 3.11 in* Bubeck (2014)*] Let $f$ be $a$-strongly convex $b$-smooth and on $\mathcal{B} \subseteq \mathbb{R}^d$. Then for all $\boldsymbol{\beta}, \boldsymbol{\beta}' \in \mathcal{B}$, one has*

$$\langle \nabla f(\boldsymbol{\beta}) - \nabla f(\boldsymbol{\beta}'), \boldsymbol{\beta} - \boldsymbol{\beta} \rangle \ge \frac{ab}{a+b} \|\boldsymbol{\beta} - \boldsymbol{\beta}'\|_2^2 + \frac{1}{a+b} \|\nabla f(\boldsymbol{\beta}) - \nabla f(\boldsymbol{\beta}')\|_2^2.$$

# H   DETAILS OF SECTION 5: EXTENSION TO UNSUPERVISED LEARNING-TO-LEARN SETTING

In addition to the assumptions in the supervised setting, in this unsupervised setting, we assume for each estimation problem, both $L_{n_1}(Z_{1:n_1})$ and $L_{n_2}(Z_{1:n_2})$ satisfy condition (b) and (c) in Assumption C.1. Similar to the supervised setting, the generalization error of the optimizer $\theta_U^*$ can be bounded by

$$\mathcal{L}_{gen}(\mathbb{P}(\mathcal{P}); \theta_U^*) \leq \underbrace{\mathbb{E}_{(Z_{1:n}, \boldsymbol{\beta}^*) \sim \mathbb{P}(\mathcal{P})} \|\boldsymbol{\beta}_T(Z_{1:n_1}; \theta_U^*) - \boldsymbol{\beta}^*\|_2^2 - \frac{1}{m} \sum_{i=1}^m \left\| \boldsymbol{\beta}_T(Z_{1:n_1}^{(i)}; \theta_U^*) - \boldsymbol{\beta}^{*(i)} \right\|_2^2}_{\text{generalization gap: Theorem 4.1}}$$

(76)

$$+ \underbrace{\frac{1}{m} \sum_{i=1}^m \left\| \boldsymbol{\beta}_T(Z_{1:n_1}^{(i)}; \theta_U^*) - \boldsymbol{\beta}^{*(i)} \right\|_2^2}_{\text{supervised training error}}.$$

(77)

However, in the unsupervised setting, the minimizer $\theta_U^*$ only guarantees that unsupervised training loss function $\frac{1}{m} \sum_{i=1}^m L_{n_2}\left(Z_{1:n_2}^{(i)}, \boldsymbol{\beta}_T(Z_{1:n_1}^{(i)}; \theta_U^*)\right)$ is small. It requires further derivations to show that the corresponding supervised training loss in Eq. 77 is small, too.

Therefore, in the following, we bridge the gap between the supervised training loss and the unsupervised training loss. Based on the proof of Theorem D.1, we know that $\|\boldsymbol{\beta}_T(Z_{1:n_1}^{(i)}; \theta_U^*)|_{\overline{S^*}}\|_0 \leq \tilde{s}$. Therefore, we can apply the restricted strong convexity of $L_{n_2}$ to obtain the following inequality for $\boldsymbol{\beta}_T = \boldsymbol{\beta}_T(Z_{1:n_1}^{(i)}; \theta_U^*)$, $\boldsymbol{\beta}^* = \boldsymbol{\beta}^{*(i)}$, and $Z = Z_{1:n_2}^{(i)}$ for each $i = 1, \cdots, m$.

$$\|\boldsymbol{\beta}_T - \boldsymbol{\beta}^*\|_2^2 \leq \frac{2}{\rho_-} \left( L_{n_2}(Z, \boldsymbol{\beta}_T) - L_{n_2}(Z, \boldsymbol{\beta}^*) + \nabla L_{n_2}(Z, \boldsymbol{\beta}^*)^\top (\boldsymbol{\beta}^* - \boldsymbol{\beta}_T) \right)$$

$$\leq \frac{2}{\rho_-} \left( L_{n_2}(Z, \boldsymbol{\beta}_T) - L_{n_2}(Z, \boldsymbol{\beta}^*) + \|\nabla L_{n_2}(Z, \boldsymbol{\beta}^*)\|_2 \|(\boldsymbol{\beta}^* - \boldsymbol{\beta}_T)\|_2 \right)$$

$$\Rightarrow \|\boldsymbol{\beta}_T - \boldsymbol{\beta}^*\|_2^2 \leq \frac{4}{\rho_-} \left( L_{n_2}(Z, \boldsymbol{\beta}_T) - L_{n_2}(Z, \boldsymbol{\beta}^*) \right) + \frac{16}{\rho_-^2} \|\nabla L_{n_2}(Z, \boldsymbol{\beta}^*)\|_2^2$$

Aggregating this inequality for $i = 1, \cdots, m$, we can obtain the following inequality that bounds the supervised training loss using the unsupervised loss.

$$\frac{1}{m} \sum_{i=1}^m \left\| \boldsymbol{\beta}_T(Z_{1:n_1}^{(i)}; \theta_U^*) - \boldsymbol{\beta}^{*(i)} \right\|_2^2 \leq \underbrace{\frac{16}{\rho_-^2 m} \sum_{i=1}^m \|\nabla L_{n_2}(Z_{1:n_2}^{(i)}, \boldsymbol{\beta}^{*(i)})\|_2^2}_{\text{statistical error}}$$

$$+ \underbrace{\frac{4}{\rho_- m} \sum_{i=1}^m \left( L_{n_2}(Z_{1:n_2}^{(i)}, \boldsymbol{\beta}_T(Z_{1:n_1}^{(i)}; \theta_U^*)) - L_{n_2}(Z_{1:n_2}^{(i)}, \boldsymbol{\beta}^{*(i)}) \right)}_{\text{unsupervised training error}}.$$

The right-hand-side of this inequality consists of a statistical error and the unsupervised training error. Finally, to characterize how small the unsupervised training error can be, we can combine the following inequality (by restricted strongly smooth) with the result of Theorem 3.1.

$$L_{n_2}(Z_{1:n_2}, \boldsymbol{\beta}_T(Z_{1:n_1}; \theta_U^*)) - L_{n_2}(Z_{1:n_2}, \boldsymbol{\beta}^*)$$
$$\leq L_{n_2}(Z_{1:n_2}, \boldsymbol{\beta}_T(Z_{1:n_1}; \theta^*)) - L_{n_2}(Z_{1:n_2}, \boldsymbol{\beta}^*)$$
$$\leq \|\nabla_{\boldsymbol{\beta}} L_{n_2}(Z_{1:n_2}, \boldsymbol{\beta}^*)\|_2 \underbrace{\|\boldsymbol{\beta}_T(Z_{1:n_1}; \theta^*) - \boldsymbol{\beta}^*\|_2}_{\text{bounded by Theorem 3.1}} + \frac{\rho_+}{2} \underbrace{\|\boldsymbol{\beta}_T(Z_{1:n_1}; \theta^*) - \boldsymbol{\beta}^*\|_2^2}_{\text{bounded by Theorem 3.1}}.$$

(78)

Overall, the generalization error can be bounded by

$$\mathcal{L}_{gen}(\mathbb{P}(\mathcal{P}); \theta_U^*)$$

$$\leq \underbrace{\mathbb{E}_{(Z_{1:n}, \boldsymbol{\beta}^*) \sim \mathbb{P}(\mathcal{P})} \|\boldsymbol{\beta}_T(Z_{1:n_1}; \theta_U^*) - \boldsymbol{\beta}^*\|_2^2 - \frac{1}{m} \sum_{i=1}^m \left\|\boldsymbol{\beta}_T(Z_{1:n_1}^{(i)}; \theta_U^*) - \boldsymbol{\beta}^{*(i)}\right\|_2^2}_{\text{generalization gap: Theorem 4.1}} *$$

$$+ \frac{4}{\rho_- m} \sum_{i=1}^m \left( \underbrace{\|\nabla_{\boldsymbol{\beta}} L_{n_2}(Z_{1:n_2}^{(i)}, \boldsymbol{\beta}^{*(i)})\|_2}_{\text{statistical error}} \underbrace{\|\boldsymbol{\beta}_T(Z_{1:n_1}^{(i)}; \theta^*) - \boldsymbol{\beta}^{*(i)}\|_2}_{\text{bounded by Theorem 3.1}} + \frac{\rho_+}{2} \underbrace{\|\boldsymbol{\beta}_T(Z_{1:n_1}^{(i)}; \theta^*) - \boldsymbol{\beta}^{*(i)}\|_2^2}_{\text{bounded by Theorem 3.1}} \right)$$

$$+ \underbrace{\frac{16}{\rho_-^2 m} \sum_{i=1}^m \|\nabla L_{n_2}(Z_{1:n_2}^{(i)}, \boldsymbol{\beta}^{*(i)})\|_2^2}_{\text{statistical error}}.$$

# I    DETAILS OF SYNTHETIC EXPERIMENTS

## I.1    A GENTLE REVIEW OF LINEAR REGRESSION AND PRECISION ESTIMATION

### I.1.1    SPARSE LINEAR REGRESSION

A sparse linear regression model reads

$$y = \boldsymbol{x}^\top \boldsymbol{\beta}^* + \epsilon$$

where $\boldsymbol{\beta}^*$ is a sparse vector and $\epsilon$ is Gaussian noise. Given $n$ observations $Z_{1:n} := \{(\boldsymbol{x}_1, y_1), \cdots, (\boldsymbol{x}_n, y_n)\}$, the goal is to estimate the vector $\boldsymbol{\beta}^*$.

In our method, we use the the least-square error to define the empirical loss function.

$$L_n(Z_{1:n}, \boldsymbol{\beta}) = \frac{1}{2n} \sum_{j=1}^{n} \left| \boldsymbol{x}_j^\top \boldsymbol{\beta} - y_j \right|^2$$

### I.1.2    SPARSE PRECISION ESTIMATION

The sparse precision matrix estimation problem in Gaussian graphical models assumes the observation of $n$ samples from a distribution $\mathcal{N}(0, \Sigma)$. Given the $n$ samples, $X_{1:n} := \{\boldsymbol{x}_1, \cdots, \boldsymbol{x}_n\}$ from $\mathcal{N}(0, \Sigma)$, the goal is to estimate the precision matrix $\Theta^* := \Sigma^{-1}$ which is the inverse covariance matrix. This precision matrix represents the conditional independency among random variables.

A commonly used objective for estimating the precision matrix is the $\ell_1$-penalized log determinant divergence:

$$- \log \det(\Theta) + \langle \Theta, \hat{\Sigma}_n \rangle + \lambda \|\Theta\|_1$$

where $\hat{\Sigma}_n$ is the sample covariance matrix. In our method, we use the likelihood term as the empirical loss.

$$L_n(X_{1:n}, \Theta) = - \log \det(\Theta) + \langle \Theta, \hat{\Sigma}_n \rangle$$

## I.2    DATA PREPARATION

For the sparse linear recovery problem, we follow the setting in Wang et al. (2014). We create the synthetic data by sampling a set of estimation problems $\{((X^{(i)}, Y^{(i)}), \boldsymbol{\beta}^{*(i)})\}$. In each problem, the design matrix $X^{(i)} \in \mathbb{R}^{n \times d}$ contains $n = 64$ independent realizations of a random vector $\mathbf{x} \in \mathbb{R}^p$ with $p = 256, 1024$ in easy or difficult setting, respectively. $\mathbf{x}$ follows a zero mean Gaussian distribution with covariance matrix $(\Sigma)_{i,j} = 0.9 \cdot 1_{\{i \neq j\}} + 1 \cdot 1_{\{i=j\}}$. The true parameter vector $\boldsymbol{\beta}^{*(i)}$ has a sparsity $s^* = \|\boldsymbol{\beta}^{*(i)}\|_0 = 16$ and its nonzero entries take values uniformly sampled from $(-2, -\frac{1}{2}) \cup (\frac{1}{2}, 2)$. The support set of each $\boldsymbol{\beta}^{*(i)}$ is independently sampled from a union support set $S$, with $|S| = 128$ to allow some similarity among the problems. The observation $Y^{(i)}$ is sampled such that $Y^{(i)} - X^{(i)} \boldsymbol{\beta}^{*(i)}$ is a $n$-dimensional Gaussian random vector with zero mean and covariance matrix $\mathbf{I}_n$. In all the experiments, 2000 such problems are used for training, 200 such problems are used for validation, and 100 such problems are used for test.

In sparse precision matrix estimation problem, we follow the setting in Guillot et al. (2012). We create the synthetic data by sampling a set of estimation problems $\{(\hat{\Sigma}_n^{(i)}, \Theta^{(i)})\}$. In each problem, the ground truth precision matrix is generated in the following ways: the lower diagonal part (lower triangular parting, exluding diagonal) of true precision matrix $\Theta$ has a sparsity $s^*$, that are uniformly selected from a union support with size $S$. After selecting the nonzero entries, we assign them values uniformly from $(-2, -\frac{1}{2}) \cup (\frac{1}{2}, 2)$ and let the upper diagonal part has the same value as lower diagonal part. Finally, a multiple of the identity was added to the resulting matrix so that the smallest eigenvalue was equal to 1. In this way, $\Theta$ was insured to be sparse, positive definite, and well-conditioned. After the precision matrix is generated, we generate the observational samples for each precision matrix, by generating $n$ independent samples of the random vector $\mathbf{x} \sim \mathcal{N}(0, (\Theta^{(i)})^{-1})$. The samples are denoted by $X^{(i)} \in \mathbb{R}^{n \times d}$. Using the samples, we can compute the sample covariance matrix $\hat{\Sigma}_n^{(i)} = \frac{1}{n} \sum_{i=1}^{n} (X^{(i)})^T X^{(i)}$.

In all the experiments, 2000 such problems are used for training, 200 such problems are used for validation, and 100 such problems are used for test. We use $(n, S, s^*) = (100, 375, 75), (200, 600, 150)$ for $d = 50, 200$, respectively.

### I.3   BASELINE IMPLEMENTATION

**ALISTA**: We follow the implementation in Liu et al. (2019a).

**RNN**: The RNN is designed following Andrychowicz et al. (2016). Initialize the $\beta_0$ as 0. At every step, we compute the gradient $\nabla_\beta L(\beta_t)$ w.r.t. to objective as the input to a LSTM with hidden dimension equals to 128. Then we use the output of the LSTM as the increment:

$$\beta_{t+1} = \beta_t + \text{LSTM}(\nabla_\beta L(\beta_t)) \tag{79}$$

We repeat this iteration 20 steps and use $\beta_{20}$ as our final output.

**RNN-$\ell_1$**: RNN-$\ell_1$ use the same architecture as RNN. The only difference is we add an extra learnable parameter $\lambda_t$ for soft thresholding at each step:

$$\beta_{t+1} = \beta_t + \text{LSTM}(\nabla_\beta L(\beta_t)) \tag{80}$$
$$\beta_{t+1} = \eta_{\lambda_t}(\beta_{t+1}) \tag{81}$$

where the softthresholding $\eta_\lambda$ is an elementwise operator maps $\eta_\lambda(x) = 0$ is $|x| < \lambda$ and $\eta_\lambda(x) = \text{sign}(x)(|x| - \lambda)$.

**GLASSO**: we use the implementation in the sklearn package.

**GGM**: We follow the implementation in Belilovsky et al. (2017).

**APF**: We follow the implementation in Wang et al. (2014). The specific algorithm steps are summarized in Algorithm 3.

---

**Algorithm 3:** The Approximate Path Following Method

**Input:** $\lambda_{\text{tgt}} > 0, \epsilon_{\text{opt}} > 0$ (Here we assume $\epsilon_{\text{opt}} \ll \lambda_{\text{tgt}}/4$), $\eta \in [0.9, 1)$
1  Initialize $\tilde{\beta}_0 \leftarrow \mathbf{0}, L_0 \leftarrow L_{\min}, \lambda_0 = \|\nabla\mathcal{L}(\mathbf{0})\|_\infty, N \leftarrow (\lambda_0/\lambda_{\text{tgt}})/\log(\eta^{-1})$.
2  **for** $t = 1, ..., N - 1$ **do**
3  $\quad$ $\lambda_t \leftarrow \eta^t \lambda_0$
4  $\quad$ $\epsilon_t \leftarrow \lambda_t/4$
5  $\quad$ $\{\tilde{\beta}_t, L_t\} \leftarrow$ Proximal-Gradient$(\lambda_t, \epsilon_t, \tilde{\beta}_{t-1}, L_{t-1}, R)$ as in Algorithm 4.
6  **end**
7  $\lambda_N \leftarrow \lambda_{\text{tgt}}$
8  $\epsilon_N \leftarrow \epsilon_{\text{opt}}$
9  $\{\tilde{\beta}_N, L_N\} \leftarrow$ Proximal-Gradient$(\lambda_N, \epsilon_N, \tilde{\beta}_{N-1}, L_{N-1}, R)$.
10  **return** $\{\tilde{\beta}_t\}_{t=1}^N$.

---

**Algorithm 4:** The Proximal Gradient Method

**Input:** $\lambda_t > 0, \epsilon_t > 0, \beta_t^0 \in \mathbb{R}^p, L_t^0 > 0$
1  Initialize $k \leftarrow 0$.
2  **repeat**
3  $\quad$ $k \leftarrow k + 1$
4  $\quad$ $L_{\text{init}} \leftarrow \max\{L_{\min}, L_t^{k-1}/2\}$
5  $\quad$ $\beta_t^k, L_t^k \leftarrow$ Line-Search$(\lambda_t, \beta_t^{k-1}, L_{\text{init}})$ as in Algorithm 5.
6  **until** $\omega_{\lambda_t}(\beta_t^k) \leq \epsilon_t$ *as defined in Equation 82*;
7  $\tilde{\beta}_t \leftarrow \beta_t^k$
8  $L_t \leftarrow L_t^k$
9  **return** $\{\tilde{\beta}_t, L_t\}$.

---

---

**Algorithm 5:** The Line Search Method

---

**Input:** $\lambda_t > 0, \boldsymbol{\beta}_t^{k-1} \in \mathbb{R}^d, L_{\text{init}} > 0, R > 0$

**1 repeat**

**2** $\quad$ $\boldsymbol{\beta}_t^k \leftarrow \mathcal{T}_{L^k,\lambda_t}\left(\boldsymbol{\beta}_t^{k-1}\right)$ as defined in Equation 83

**3** $\quad$ **if** $\phi_{\lambda_t}\left(\boldsymbol{\beta}_t^k\right) > \psi_{L_t^k,\lambda_t}\left(\boldsymbol{\beta}_t^k;\boldsymbol{\beta}_t^{k-1}\right)$ **then**

**4** $\quad\quad$ $\mid$ $L_t^k \leftarrow L_t^k$.

**5** $\quad$ **end**

**6 until** $\phi_{\lambda_t}(\boldsymbol{\beta}_t^k) \leq \psi_{L_t^k,\lambda_t}(\boldsymbol{\beta}_t^k, \boldsymbol{\beta}_t^{k-1})$ *as defined in Equations 84 and 85*;

**7** $\tilde{\boldsymbol{\beta}}_t \leftarrow \boldsymbol{\beta}_t^k$

**8** $L_t \leftarrow L_t^k$

**9 return** $\{\tilde{\boldsymbol{\beta}}_t, L_t\}$.

---

$$\omega_\lambda(\boldsymbol{\beta}) = \min_{\boldsymbol{\xi}' \in \partial \|\boldsymbol{\beta}\|_1} \max_{\boldsymbol{\beta}' \in \Omega} \left\{ \frac{(\boldsymbol{\beta} - \boldsymbol{\beta}')^T}{\|\boldsymbol{\beta} - \boldsymbol{\beta}'\|_1} \left( \nabla \widetilde{\mathcal{L}}_\lambda(\boldsymbol{\beta}) + \lambda \boldsymbol{\xi}' \right) \right\} \tag{82}$$

$$\left( \mathcal{T}_{L_t^k,\lambda_t}\left(\boldsymbol{\beta}_t^{k-1}\right) \right)_j = \begin{cases} 0 & \text{if } \left|\bar{\beta}_j\right| \leq \lambda_t/L_t^k \\ \text{sign}\left(\bar{\beta}_j\right)\left(\left|\bar{\beta}_j\right| - \lambda_t/L_t^k\right) & \text{if } \left|\bar{\beta}_j\right| > \lambda_t/L_t^k \end{cases} \tag{83}$$

$$\phi_\lambda(\boldsymbol{\beta}) = \widetilde{\mathcal{L}}_\lambda(\boldsymbol{\beta}) + \lambda\|\boldsymbol{\beta}\|_1 \tag{84}$$

$$\psi_{L_t^k,\lambda_t}\left(\boldsymbol{\beta};\boldsymbol{\beta}_t^{k-1}\right) = \mathcal{L}\left(\boldsymbol{\beta}_t^{k-1}\right) + \nabla\mathcal{L}\left(\boldsymbol{\beta}_t^{k-1}\right)^T\left(\boldsymbol{\beta} - \boldsymbol{\beta}_t^{k-1}\right) + \frac{L_t^k}{2}\left\|\boldsymbol{\beta} - \boldsymbol{\beta}_t^{k-1}\right\|_2^2 + \mathcal{P}_{\lambda_t}(\boldsymbol{\beta}) \tag{85}$$

### I.4 TRAINING AND EVALUATION

We trained the learning-based methods to minimize the weighted loss. That's to say, for each sparse linear recovery problem, given the sequence of outputs $(\boldsymbol{\beta}_1, ..., \boldsymbol{\beta}_T)$, the loss is computed as:

$$L = \sum_{i=1}^T \gamma^{T-i}\|\boldsymbol{\beta}_i - \boldsymbol{\beta}^*\|_2^2 \tag{86}$$

where $\gamma$ decreasing from 0.9 to 0.1 during the training process. The procedure is similar in the sparse precision matrix estimation problem. We use optimizer Adam (Kingma & Ba, 2014). For sparse linear recovery problem, we use batch size 10, we train 500 epochs with learning rate 1e-4 and select the model based on the l2 loss on valid data. For sparse precision matrix estimation problem, we use batch size 40, we train 200 epochs with learning rate 1e-3 and select the model based on Frobenius loss on valid data.

For classical algorithms, we select their parameters, $\lambda$ for APF and $\rho$ for GISTA, based on their performance on the validation set. Specifically, we evaluate APF or GISTA with $\rho$ or $\lambda$ from $\{0.01, 0.025, 0.05, 0.1, 0.2\}$ and use one with the best recovery error in test.

Table 3: Training time for SLR (minutes)

| Sizes | $d = 256$ | $d = 1024$ |
|---|---|---|
| PLISA | 393 | 462 |
| ALISTA | 176 | 271 |
| RNN | 96 | 99 |
| RNN$_{\ell_1}$ | 101 | 106 |
| APF | 214 | 426 |

Table 4: Training time for SPE (minutes)

| Sizes | $d = 50$ | $d = 100$ |
|---|---|---|
| PLISA | 35 | 39 |
| GGM | 14 | 43 |
| GISTA | 176 | 116 |
| APF | 316 | 331 |
| GLASSO | 42 | 57 |

We report the total training time for learning based methods as well as the parameter tuning time for classical algorithms. From Table 3 and Table 4, we can see that training a learning-based method is

cheap in our experiments, as 1) A single forward for PLISA or GGM is very fast as stated in Table 1. 2) We can easily parallel the computations to handle a batch of problems during the training time, as the learning-based methods do not require line-search.

The evaluation is performed on a server with CPU: Intel(R) Xeon(R) Silver 4116 CPU @ 2.10GHz, GPU: Nvidia GTX 2080TI, Memory 264G, in single thread.

## I.5 ABLATION STUDY

Table 5: Ablation study of PLISA ($p = 1024$). TPR is the true positive rate of recovering the nonzero entries of $\beta^*$. FPS is the cardinality of false positive entries. Note that the true sparsity level is $s^* = 16$. Standard deviations over 100 test problems are present in the parantheses.

|  | PLISA | PLISA-single | PLISA-$\ell_1$ |
|---|---|---|---|
| $\ell_2$ error | 1.34 (2.28) | 18.25 (6.06) | 2.20 (2.76) |
| TPR | 0.99 (0.01) | 0.62 (0.19) | 0.99 (0.02) |
| FPS | 16.65 (13.60) | 51.07 (6.67) | 25.11 (13.30) |

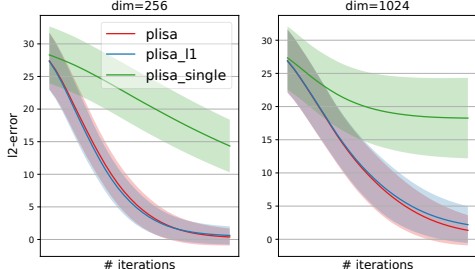

Figure 6: Ablation study.

We consider two variants of PLISA to perform the ablation study. One is PLISA-single which employs a single regularization parameter across different entries, i.e., $\eta_1 = \cdots = \eta_d$ and $\lambda_1^* = \cdots = \lambda_d^*$. The other is PLISA-$\ell_1$ which does not learn the penalty function but uses the $\ell_1$ norm, i.e., $P_{\boldsymbol{w}}(\boldsymbol{\lambda}, \boldsymbol{\beta}) = \|\boldsymbol{\lambda} \circ \boldsymbol{\beta}\|_1$. Fig. 6 and Table 5 show the vanilla PLISA performs better than alternatives. Especially, it has a much better accuracy than PLISA-single. Therefore, this ablation study has validated the effectiveness of using entry-wise regularization parameters and learning the penalty function.

## I.6 REAL-WORLD EXPERIMENTS

We use the following 3 real-world datasets.

(1) **Gene** - a single-cell gene expression dataset that contains expression levels of 45 transcription factors measured at different time-points. We follow Ollier & Viallon (2017) to pick the transcription factor, EGR2, as the response variable and the other 44 factors as the covariates. In this dataset, each time-point is considered as an estimation problem. In each estimation problem, the goal is learning the regression weights $\beta^*$ on the 44 factors to predict the expression level of the target transcription factor, EGR2. Therefore, in this problem, $p = 44$. This dataset contains the gene expression data for 120 single cells at 8 different time points. For each time point, we randomly split the 120 samples into 6 sets, so that each set contains 20 samples. By doing this, we construct 48 estimation problems each of which contains 20 samples. 10 samples are used for recovering the parameters $\beta^*$ and the other 10 samples are used for evaluate it by computing the least-square error. We use 36 problems for training, 6 problems for validation, and 6 problems for testing.

(2) **Parkinsons** - a disease dataset that contains symptom scores of Parkinson for different patients. Each patient is considered as an estimation problem. In each estimation problem, the goal is learning the regression weights $\beta^*$ on 19 bio-medical features to predict the symptom score. This dataset contains 42 patients, so there are 42 estimation problems in total. Each patient is examined at different time-point and each time a sample is generated. For each patient we randomly select 100 samples, so that eventually our dataset contains 42 estimation problems each of which contains 100 samples. 50 are used for recovering the parameter $\beta^*$ and 50 are used for evaluation. We use 28 problems for training, 7 problems for validation, and 7 problems for testing.

(3) **School** - an examination score dataset of students from 139 secondary schools in London. Each school is considered as an estimation problem. In each estimation problem, the goal is learning the regression weights $\beta^*$ on 28-dimensional school and student features to present the exam scores for all students. We use the dataset from Malsar package (Zhou et al., 2011). For each school, we randomly select 40 students as the samples. Since some schools contain less than 40 students, we finally obtain 125 estimation problems (schools) each of wich contains 40 samples. 20 are used for

recovering the parameter $\beta^*$ and 20 are used for evaluation. We use 100 problems for training, 10 problems for validation, and 15 problems for testing.

On each dataset, we train each learning-based algorithm for 200 epochs using Adam with learning rata 1e-3. The batch size is set to be 6.

