# OpenReview forum: "Provable Learning-based Algorithm For Sparse Recovery"
_ICLR.cc/2022/Conference — ICLR 2022 Poster_

### Official Review · Reviewer_ZD71 · 2021-10-29

**Correctness:** 3
**Technical Novelty And Significance:** 2
**Empirical Novelty And Significance:** 2
**Recommendation:** 5
**Confidence:** 4

**Main Review:**

1. Strengths:

+ The idea of unrolling path-following algorithms for learning to optimize hyper-parameters (such as learning rate, regularization modulus and mixture weights of regularization components) is somewhat novel as far as I can see.

+ The analysis is theoretically sound and the writing style is clear and clean.

2. Weaknesses

- My main concern is about the relevance and strength of theory. There are two types of guarantees established in this work on parameter estimation error and generalization gap bounds respectively. As shown in Theorem 3.1, the sparsity recovery analysis basically reveals that there exists a set of hyper-parameters under which the proposed PLISA algorithm is able to efficiently and accurately recover the underlying true model. This result seems not surprising at all as PLISA by definition naturally mimics an existing iterative sparse recovery method with similar guarantees. More importantly, IMO the current convergence/estimation analysis has little to with the optimization of hyper-parameters $\theta$ which is of central interest of deep unrolling but unfortunately totally ignored in the present study. In Theorem 3.1, the existence of the desired set of parameters is lack of transparency in the sense that it relies on another set of conditions as defined in Definition C.1 which are fairly stringent especially the condition (iii).

- Continuing the above comment, the generalization analysis is also of limited interest and novelty. First of all, since the generalization is about the hyper-parameters involved in meta-optimization, most existing uniform convergence analysis techniques in classic learning theory are expected to work for bounding the generalization gap. Is there any particular reason to adopt the framework of Bartlett and Mendelson (2002) for generalization analysis? The technical part is relatively straightforward; it is not clear what’s new in this section of analysis, if any.

- It is unclear how to optimize the meta-parameters contained in Algorithms 1, probably via backpropagation? More implementation detail about meta-parameter update are desirable in addition to the iterative proximal gradient steps for cell objectives.

- Last but not least, I am seriously doubting the practical usage of the proposed approach to real-data sparse learning problems because it is only designed for well-defined sparsity models. An immediate challenge here is that the ground-truth sparsity models are required to supervise the meta-learning, which however are typically unavailable over the training optimization tasks!

3. Minor comments:

- Equation (6): the general case of varying $n_i$ seems not investigated throughout.

- The reason of introducing $\varepsilon>0$ in Theorem 3.1 is not clear.

- Typos: P3, beside Algorithm 1: minimize  minimizes; and others.



**Summary Of The Paper:**

The paper presents a deep unrolling method that unrolls the path-following algorithm of Wang et al. (2014) for learning to regularized sparsity estimation. Theoretical guarantees on sparsity recovery and generalization performances are provided. A simulation study is carried out to verify the theory and effectiveness of the proposed algorithm.

**Summary Of The Review:**

The algorithm is somewhat interesting but the theory is of limited relevance and novelty. The implementation details and practical usage of algorithm are largely unclear in the current stage.

=== Post discussion update ===

After reading other reviews and the author response, I would upgrade my rating to 5: marginally below the acceptance threshold.

---

> ### Author Response · Authors · 2021-11-12
> **Initial response: clarification for Theorem 3.1**
>
> Since the reviewer has strong concerns about both Theorem 3.1 and Theorem 4.1. We respond to these two aspects separately in our initial response.
>
> > As shown in Theorem 3.1, the sparsity recovery analysis basically reveals that there exists a set of hyper-parameters under which the proposed PLISA algorithm is able to efficiently and accurately recover the underlying true model. This result seems not surprising at all as PLISA by definition naturally mimics an existing iterative sparse recovery method with similar guarantees.
>
> 1. Having a similar convergence guarantee as the existing algorithm is one of the most important reasons why people are designing the architecture by unrolling it. By unrolling an algorithm, we benefit from the analysis tool of this algorithm for deriving the guarantees for the deep architecture.
> 2. We highlighted in many places in the paper that our architecture is **different** from the existing algorithm by considering **(1)** element-wise regularization and **(2)** a learnable penalty function, which are infeasible in traditional algorithms. More importantly, we show by Theorem 3.1 that such two designs will **effectively improve the achievable accuracy**, better than the existing algorithm. Therefore, the result in Theorem 3.1 does not simply copy the result for the existing algorithm. It plays an important role in showing the improvement achieved from the new designs.
> 3. Under Theorem 3.1, following the sentence *'In the following, we will elaborate on how the design of entry-wise regularization and learnable penalty function have improved the statistical error.'*, we explained the details in two paragraphs *'(i) Impact of entry-wise regularization'* and *'(ii) Impact of learnable penalty function'*.
>
> > IMO the current convergence/estimation analysis has little to with the optimization of hyper-parameters which is of central interest of deep unrolling but unfortunately totally ignored in the present study.
>
> 1. First of all, Theorem 3.1 shows that the training error can be small. We know in statistical learning that the generalization error is bounded by training error + generalization gap. Essentially Theorem 3.1 and Theorem 4.1 are bounding these two terms (as introduced in Section 2.2).
> 2. Of course, the training error depends on both the (1) model space and the (2) optimization algorithm for searching the model. **We agree that our Theorem 3.1 only handles the first aspect (1)**, by showing the existence of a good model in the model space which can lead to a small training error. However, we argue that it is still important and useful even without analyzing the second aspect (2) for the meta-optimizer (such as SGD or Adam).
> 3. To support this opinion, we would like to cite the following list of recent top conference papers.
> - [NeuRIPs spotlight]: Theorem 2 in Theoretical Linear Convergence of Unfolded ISTA and its Practical Weights and Thresholds
> - [ICLR Spotlight]: Theorem 2-3 in SPARSE CODING WITH GATED LEARNED ISTA
> - [ICLR]: Theorem 1 in ALISTA: Analytic Weights Are As Good As Learned Weights in LISTA
> - [AAAI]: Theorem 1 in Learned Extragradient ISTA with Interpretable Residual Structures for Sparse Coding
> - [ICML]: Theorem 1 in Differentiable Linearized ADMM.
> - The list can go on
>
> All these papers focus on proving the learning-based algorithms can achieve a small error if a proper set of parameters are selected, **without** proving such parameters can be obtained by the meta-optimizer such as SGD, Adam, etc. Furthermore, those listed theorems are **the only** key theorems in these papers, unlike our paper which has further supplied the generalization analysis in Theorem 4.1.
>
> 4. Finally, we agree that proving the effectiveness of the meta-optimizer (SGD, Adam, etc) is important, but we argue that model capacity analysis is important and non-trivial by itself, which constitutes the key contributions in many papers. Studying the meta-optimizer is another topic that can require the scoop of another paper.
>
> > In Theorem 3.1, the existence of the desired set of parameters is lack of transparency in the sense that it relies on another set of conditions as defined in Definition C.1 which are fairly stringent especially the condition (iii)
>
> - We appreciate that the reviewer read the Appendix. Definition C.1 defines some conditions. However, for Theorem 3.1 to hold, we need only **one set of parameters** satisfying the conditions because Theorem 3.1 is a statement of **existence**. In this case, conditions in Definition C.1 are **trivial**. This is why we don't need to mention Definition C.1 in Theorem 3.1. We are not trying to hide anything.
>
> - To be more specific, the condition (iii) mentioned by the reviewer says $w_2' \frac{1}{b}+ w_3'\frac{1}{a-1}\leq\xi_{max}<\rho_-$. It is trivial because $\rho_->0$, and these exists weights $w$ such that $w_1'$ tends to 1 and $w_2',w_3'$ tend to 0.
>
> - In the revised paper, we can state this more explicitly in the proof of Theorem 3.1.

---

> > ### Author Response · Authors · 2021-11-12
> > **Initial response: clarification for Theorem 4.1**
> >
> > > First of all, since the generalization is about the hyper-parameters involved in meta-optimization, most existing uniform convergence analysis techniques in classic learning theory are expected to work for bounding the generalization gap. Is there any particular reason to adopt the framework of Bartlett and Mendelson (2002) for generalization analysis? The technical part is relatively straightforward; it is not clear what’s new in this section of analysis, if any.
> >
> > 1. When we summarize our contributions in the Introduction Section, we stated: 'By **(1)** combining the analysis tools of deep learning theory and optimization algorithms, our result reveals a **(2)** novel connection between the generalization ability of PLISA and its algorithmic properties including the convergence rate and stability. Benefit from this connection, our generalization bound is tight in the sense that **(3)** it matches the interesting behavior of PLISA observed in experiments - the generalization gap could decrease in
> > the number of layers, which is rarely observed in conventional neural networks.'
> >
> > 2. **(1) Novel proof technique.** Technically, our analysis carefully combines the generalization analysis framework and the convergence/stability analysis for optimization algorithms. A novel combination of existing analysis tools is a new contribution, isn't it? More importantly, we are not randomly combining technique A with technique B. This is specific for understanding the algorithm unrolling architecture, which has led to interesting results for this problem.
> >
> > 3. **(2) Novel connection.** Because of the novel combination, our derived bound has linked the generalization ability of PLISA to its convergence and stability properties. Such a linkage will NOT be observed if we do not carefully incorporate the optimization analysis tools in the generalization analysis!
> >
> > 4. **(3) New result and tighter bound.** Apart from the linkage, our result is new because it shows the generalization gap could **decrease** in the number of layers, and this matches the empirical observations!
> >
> > 5. **Not straightforward.** As mentioned in the Related Work Section, 'Joukovsky et al. (2021) do not connect the generalization analysis with algorithmic properties to provide tighter bounds as in our work.' The work of Joukovsky et al. (2021) has adopted a similar generalization analysis framework as ours to analyze unrolled algorithms, **without combining** it with optimization analysis in the proof. As a result, their bounds **cannot reveal** the linkage as ours and **cannot explain** the decrease in the generalization gap when using more layers. We think it is overconfident to say our technical part is straightforward.
> >
> > 6. **Why this framework.** Our key result is Lemma 4.1 which shows the parameter perturbation error. In fact, this can be used for deriving generalization bounds both in the Rademacher complexity framework and the PAC-Bayes framework. Honestly, there is no strong reason for choosing the Rademacher complexity framework. Generalization bounds for various neural architectures such as feed-forward neural networks [1], RNN [2], GNN [3], etc, are derived under this framework. We go for a commonly adopted framework, as developing a new framework for generalization analysis isn't our main purpose.
> >
> > [1] Bartlett et al., Spectrally-normalized margin bounds for neural networks, NeuRIPs 2017
> >
> > [2] Chen et al., On generalization bounds of a family of recurrent neural networks, AISTAT 2020
> >
> > [3] Garg et al., Generalization and representational limits of graph neural networks, ICML 2020

---

> > > ### Author Response · Authors · 2021-11-12
> > > **Initial response: other questions and plan for revised paper**
> > >
> > > > It is unclear how to optimize the meta-parameters contained in Algorithms 1, probably via backpropagation? More implementation detail about meta-parameter update are desirable in addition to the iterative proximal gradient steps for cell objectives.
> > >
> > > - In Section 2.2 LEARNING-TO-LEARN SETTING, we presented the training loss which is used to optimize the parameters $\theta$.
> > > - In Appendix G: EXPERIMENT DETAILS, we presented "we use optimizer Adam (Kingma & Ba, 2014) to train 500 epochs with learning rate 1e-4". We can move this sentence to the main text.
> > >
> > > > Last but not least, I am seriously doubting the practical usage of the proposed approach to real-data sparse learning problems because it is only designed for well-defined sparsity models. An immediate challenge here is that the ground-truth sparsity models are required to supervise the meta-learning, which however are typically unavailable over the training optimization tasks!
> > >
> > > Thanks for the question! We positioned our paper as a theory-oriented work, but we plan to incorporate new results in the revisited paper to address this concern.
> > >
> > > 1. We will extend the paper to incorporate an unsupervised scenario by using the likelihood (or reconstruction error in the case of compressed sensing) as the meta training loss. This will lead to an extra statistical error term in the theorem due to the gap between the training loss and the true error that we aim to minimize.
> > >
> > > 2. We will add a set of real experiments.
> > >
> > > ---
> > >
> > > Please let us know if there are further suggestions for our paper. We are open to discussion.

---

> > > > ### Comment · Reviewer_ZD71 · 2021-11-16
> > > > **Response**
> > > >
> > > > Thanks for the clarifications. In terms of optimization details, Section 2.2 is mainly about the definition of objective functions for meta-parameters estimation, while Appendix G is mainly about the choice of optimizer. The implementation details such as the computation of (stochastic) gradient are still largly missing throughout the paper.

---

> > > > > ### Author Response · Authors · 2021-11-16
> > > > > **Follow-up response to Reviewer ZD71 (part 3)**
> > > > >
> > > > > Dear reviewer, thank you for your active response. We implement the experiments using *PyTorch* and the gradients are computed automatically. We will release the code upon publication.
> > > > >
> > > > > We disagree with the comment that **"The implementation details such as the computation of (stochastic) gradient are still largly missing throughout the paper"**. As a deep learning problem, our implementation is standard and our statement for the training is standard. We have specified most necessary information for training a deep model:
> > > > >
> > > > > - the architecture (the model)
> > > > > - the training loss
> > > > > - the training optimizer (i.e., Adam) and the learning rate
> > > > > - the number of epochs
> > > > > - the data-split
> > > > > - the computing resources for running the experiments
> > > > >
> > > > > We missed the batch size which is 10 used in the experiments. We will add this number to the appendix. Other than that, we believe we haved provided all important details for reproducing the experiments. Please let us know what are the missing details, resulting in the conclusion "details are largely missing", and we are willing to discuss whether they should be included in the paper.

---

> > > ### Comment · Reviewer_ZD71 · 2021-11-16
> > > **Response**
> > >
> > > Thank you for the detailed response which much clarifies my concern on the novelty of generalization analysis. However, I still have some reservations on the claim about “new result and tighter bound” as quoted below
> > >
> > > *Apart from the linkage, our result is new because it shows the generalization gap could decrease in the number of layers, and this matches the empirical observations!*
> > >
> > > In my opinion, for a deep unrolling framework, it is not surprising that the optimization error component should be decreased with respect to the number of layers (rounds of iteration). Also, the exponential terms on layers appear inside the logarithmic factors, which makes such a scaling effect less interesting for improving the overall statistical efficiency.

---

> > > > ### Author Response · Authors · 2021-11-16
> > > > **Follow-up response to Reviewer ZD71 (part 2)**
> > > >
> > > > We are happy that our response has much clarified your concern about the novelty and we are happy to address the remaining concern.
> > > >
> > > > > Q1: *"our result is new because it shows the generalization gap could decrease in the number of layers, and this matches the empirical observations!" In my opinion, for a deep unrolling framework, it is not surprising that the optimization error component should be decreased with respect to the number of layers (rounds of iteration).*
> > > >
> > > > What our claim says is "it shows the **generalization gap** could decrease in the number of layers". We didn't say "we show the **optimization error** could decrease in the number of layers" and claim it as our novelty.
> > > >
> > > > We are not sure whether the reviewer criticizes this claim because - **after** observing the linkage between the generalization gap and convergence, it seems natural that the generalization gap could decrease. However, we would like to emphasize that the linkage is built by us, is our contribution, and is highly nontrivial.
> > > >
> > > > We use the term '**apart from** the linkage, our result is new because...' in our response because we think making the linkage itself is interesting even if it does not lead to the decrease in layers. If the reviewer thinks the word **apart from** is too strong, we can modify this term in our response. But this should not affect the evaluation of the paper.
> > > >
> > > > > *Q2: Also, the exponential terms on layers appear inside the logarithmic factors, which makes such a scaling effect less interesting for improving the overall statistical efficiency.*
> > > >
> > > > 1. log exp (KT) = KT is linear in K and T. Please see the plots in Fig 3 and Fig 6. It is effective. Besides, what will be an **interesting** rate in generalization error, in terms of the number of layers? We hope the reviewer can elaborate more on this.
> > > > 2. More importantly, the purpose of the analysis is not to **improve** statistical efficiency, it is just to reveal the **actual** generalization gap. Let's say if the model has exactly the generalization error in a log exp rate, why should we aim to prove a rate that decreases faster in layers? Our rate is already tighter than existing ones and has matched the empirical observations.

---

> > ### Comment · Reviewer_ZD71 · 2021-11-16
> > **Response**
> >
> > Thank you for the detailed response. There is no denying that the model capacity result established in Theorem 3.1 ensures that any reasonable optimization algorithm should work no worse than the model capacity. However, my real concern here is that such type of model capacity theory might be of limited interest in the considered problem regime because:
> >
> > 1. Arguably for any sparsity recovery model/algorithm, it might be the most basic requirement that there exists a set of *good* hyper-parameters under which the desired estimation performance can be guaranteed; otherwise why should we bother to develop that model or algorithm in the first place? It is good that Theorem 3.1 guarantees the efficiency and accuracy of the considered penalized model (1) under certain choices of hyper-parameters. However, this kind of guarantee still seems loosely related to the considered deep unrolling algorithm itself.
> >
> > 2. While intuitive and somewhat interesting, the optimization-plus-estimation-type error composite bound in Theorem 3.1 is not surprising and the relevant analysis techniques are not particularly novel from the perspective of sparse learning, especially when the paper is positioned as a theoretical contribution. .
> >
> > By the way, it is still unclear to me why to introduce the quantity $\varepsilon>0$ in Theorem 3.1? I do hope the authors can explain this point in case anything missed.

---

> > > ### Author Response · Authors · 2021-11-16
> > > **Follow-up response to Reviewer ZD71 (part 1)**
> > >
> > > The authors highly appreciate the reviewer's response and involvement in this discussion period. We will clarify any remaining concerns and are open to discussion.
> > >
> > > > *Q1: It is good that Theorem 3.1 guarantees the efficiency and accuracy of the considered penalized model (1) under certain choices of hyper-parameters. However, this kind of guarantee still seems loosely related to the considered deep unrolling algorithm itself.*
> > >
> > > We are honestly confused about this concern. Could the reviewer explain more about what are "the considered penalized model (1)" and "the considered deep unrolling algorithm" respectively in this question? To us, these two terms are referring to the same thing, and therefore we don't understand why the guarantee about the former seems loosely related to the latter.
> > >
> > > - The deep unrolling algorithm contains T cells (Figure 2 and Algorithm 1). $\beta_1,...,\beta_T$ are the outputs of these T cells, and $\beta_T$ is the final output. The output depends on the parameters $\theta$ in the deep unrolling algorithm, so we use the notation $\beta_T(Z_{1:n};\theta)$.
> > > - In other words, $\beta_T(Z_{1:n};\theta)$ is the final output of the deep unrolling algorithm. The guarantee in Theorem 3.1 on the accuracy of $\beta_T(Z_{1:n};\theta)$ is exactly the guarantee on the accuracy achievable by the deep unrolling algorithm.
> > >
> > > > *Q2: While intuitive and somewhat interesting, the optimization-plus-estimation-type error composite bound in Theorem 3.1 is not surprising and the relevant analysis techniques are not particularly novel from the perspective of sparse learning, especially when the paper is positioned as a theoretical contribution.*
> > >
> > > 1. We sincerely ask why a theorem should be surprising?
> > > 2. The analysis of sparse learning has been developed for so many years. Our paper is not to propose a ground-breaking new analysis framework for traditional sparse learning. Instead, we utilize the sparse learning analysis framework to develop **new results to support our claim** for the new deep unrolling algorithm under a nonconvex setting that is not well-studied in the area of deep unrolling. Technically, since we need to handle 'entry-wise regularization parameters', the derivation becomes more difficult, which is not tackled in classic analysis.
> > > 3. Our deep unrolling algorithm is different from classic algorithms by designing **(i) learnable entry-wise regularization parameters** and **(ii) Learnable penalty function**. A key purpose of Theorem 3.1 is to show how these two designs can effectively **improve** the achievable accuracy and why these two designs can make the achievable accuracy **better** than classic algorithms. Without these two designs, the original path-following algorithm is NOT guaranteed to achieve the accuracy indicated in Theorem 3.1. Our theorem has served its purpose.
> > >
> > > > *Q3: By the way, it is still unclear to me why to introduce the quantity $\epsilon$ in Theorem 3.1?*
> > >
> > > Thank you for this technical question. In fact, $\epsilon$ is a subtle technical treatment. To explain it, let us first recall some facts in classic analysis. The architecture of our deep unrolling algorithm is designed based on the classic path-following algorithm. The algorithm starts with a large regularization parameter $ \lambda_0=|\nabla L_n(Z_{1:n},0)|_\infty $ and gradually decreases it to a target regularization parameter $\lambda_T$.
> > >
> > > To achieve **optimal** statistical accuracy, theoretically, $\lambda_T$ should be $c|\nabla L_n(Z_{1:n},\beta^*)|_\infty $ in the classic algorithm.
> > >
> > > However, in our deep unrolling algorithm, we need to handle 'entry-wise regularization parameters' as we mentioned above. To achieve optimal statistical accuracy in our case, theoretically, $\lambda_T$ should be $\lambda_T= c|\nabla L_n(Z_{1:n},\beta^*)|$ (This is a vector, without the $\infty$-norm. The notation $|\cdot|$ is element-wise absolute value). We introduce $\epsilon$ because some entries of $\lambda_T=c|\nabla L_n(Z_{1:n},\beta^*)|$ could be 0. To decrease the corresponding entries of $\lambda_0$ to zero, it will take infinitely many decreasing steps no matter what's the decrease rate. Therefore, we take  $\lambda_T= c|\nabla L_n(Z_{1:n},\beta^*)| \vee \epsilon = \max(\nabla L_n(Z_{1:n},\beta^*)|, \epsilon)$ in the analysis to keep the number of steps finite, but $epsilon$ can be arbitrary small.

---

> > > > ### Comment · Reviewer_ZD71 · 2021-11-18
> > > > **Re: Follow-up response to Reviewer ZD71 (part 1)**
> > > >
> > > > Thank you for your prompt feedback.  I still don't think the insight of Theorem 3.1 is particularly new, even though it is developed for understanding a *new* deep unrolling algorithm for non-convex sparsity recovery.  As pointed out in my initial review, the analysis is carried out under a  strong-convexity lower bound condition of  Assumption C.1(iii). Clearly, this condition  essentially ensures the modified loss $L_{\lambda}$ and regularized loss $\phi_{\lambda}$ to be restricted strongly convex, even though the individual componnets in the penalty $Q_w(\lambda, \beta)$ are allowed to be non-convex.  In this sense, the entire story is about the analysis of a restricted strongly convex sparsity recovey model which IMO is relatibely well-understood in the community. The hard core non-convexity of the  deep unrolling procedure as introduced in this work seems not touched in the current theory.

---

> ### Author Response · Authors · 2021-11-23
> **Paper revision and further response**
>
> Dear Reviewer uCY5,
>
> &nbsp;
>
> Thank you for your time and patience in reviewing our paper! We have posted the revised paper, in which we include the following revision to address your concerns on the practical usage of our method.
>
> -- We add a new section “5 Extension to Unsupervised Learning-to-learn Setting” to explain how our work (both methodologically and experimentally) can extend to the setting when the ground-truth parameter is unavailable in the training set.
>
> -- Experimentally, we conduct experiments on 3 public datasets under the unsupervised learning setting and report the results in Section 7.2.
>
>
> &nbsp;
>
> Furthermore, thank you again for the active engagement in the discussion period! We respond to your latest comments below:
>
> > *“I still don't think the insight of Theorem 3.1 is particularly new, even though it is developed for understanding a new deep unrolling algorithm for non-convex sparsity recovery ... the entire story is about the analysis of a restricted strongly convex sparsity recovey model which IMO is relatibely well-understood in the community.“*
>
> &nbsp;
>
> *** Firstly, ***
>
> There is no denying that the analysis techniques of Theorem 3.1 largely follow the existing optimization literature. We follow the model assumptions in two interesting papers on nonconvex learning [1,2], so we totally agree with the reviewer that Theorem 3.1 is an analysis of restricted strongly convex sparsity recovery models.
>
> - [1] Wang, Z., Liu, H., & Zhang, T. (2014). Optimal computational and statistical rates of convergence for sparse nonconvex learning problems. *Annals of statistics*.
> - [2] Loh, P. L., & Wainwright, M. J. (2015). Regularized M-estimators with nonconvexity: Statistical and algorithmic theory for local optima. *The Journal of Machine Learning Research*.
>
> We respect the opinion of Reviewer uCY5, but we want to emphasize that we did not claim in the paper that our analysis techniques in Theorem 3.1 are novel. We would like to kindly refer the reviewer to both the abstract and introduction, in which the novelty of this paper is highlighted from **3 other aspects**.
>
> &nbsp;
>
>
> *** Secondly, ***
>
> However, Theorem 3.1 is **a necessary piece** in this paper, because the theoretical story in this paper is bounding the generalization error of PLISTA$_\theta$ based on:
>
> &nbsp;
>
> &nbsp;
>  &nbsp;&nbsp;&nbsp;
> &nbsp;
> &nbsp;
> &nbsp;
> *generalization error <= training error (Theorem 3.1) + generalization gap (Theorem 4.1)*
>
> &nbsp;
>
> Theorem 3.1 is a necessary piece to complete the story. It implies that
>
> -- the training error **can be small without using too many layers** in PLISTA$_\theta$
>
> -- the use of entry-wise regularization and learnable penalty can both **help to reduce** the achievable training error
>
> Therefore, Theorem 3.1 has served its purpose in our paper, and in both abstract and introduction, we have only emphasized the technical novelty in analysis techniques for Theorem 4.1.
>
> &nbsp;
>
>
> *** To improve clarity ***
>
> In the revised paper, we replace the sentence *“this generalization error depends on both the (i) empirical error (i.e., training loss) and the (ii) generalization gap”* by *Equation 8*. We hope Equation 8 can better indicate the roles of Theorem 3.1 and Theorem 4.1 in the whole story to improve the clarity!
>
> &nbsp;
>
> ===
>
>
> We hope our revision and response help to address your concerns. We sincerely appreciate your responses during the past discussion period. We are more than happy and open to further discussion in the remaining week! Thank you again for your time.

---

### Official Review · Reviewer_AZru · 2021-10-30

**Correctness:** 3
**Technical Novelty And Significance:** 3
**Empirical Novelty And Significance:** 4
**Recommendation:** 6
**Confidence:** 4

**Main Review:**

update
-----
I maintain my rating after reading the author(s)' response.

The approach from this paper is promising and mostly novel. The paper is generally well written. Overall I recommend accept. I have a few concerns regarding the theoretical claim and experiment.

Strong points
-----
1. The paper tackles a classic and important problem (i.e., compressed sensing), and the learning-to-learn approach should be a good step in this direction.
2. The framework from the paper appears flexible enough to handle other sparse estimation problems. So the work may lead to future impact.
3. The author(s) gives some theoretical justification of their procedures.

Concerns
-----
1. In abstract and later in the intro, the author(s) claims that "our analysis makes novel connections between the generalization ability and algorithmic properties such as stability and convergence". The author(s) should be careful about the scope of this claim. The connection between algorithmic stability and generalization is textbook classic. The paper applies this framework to the specific setting of learning-based sparse recovery.

2. Theorem 3.1 appears somewhat weak. The claim is that "there exists a set of parameters θ .... such that the estimation error is small". However, these parameters are also learned. Is there any (theoretical) justification why the algorithm would find such good parameters θ?

3. In Section 4 on generalization, the author(s) should clarify the distributional assumptions here. Are we assuming random iid design? Is $\beta^*$ also distributional? (Note that Assumption 3.1 is a set of deterministic conditions.)

4. The paper does not compare PLISA with other classic algorithms in the sparse recovery literature, for example, convex relaxation or linear sketching methods. It would to nice if the experiments are more comprehensive.

Other minor issues
-----
1. There has been a few works on unrolling or differentiable learning of classic algorithms. For example, https://arxiv.org/abs/1910.13984 and https://arxiv.org/abs/1903.08850. This is very closely related to the nature of this work. The authors should make a reference.
2. I am not familiar Wang, et al (2014). It would be helpful if the author(s) could give a quick rundown of the algorithm therein, perhaps in the appendix. What are the main features of Wang, et al (2014) that PLISA borrows?

**Summary Of The Paper:**

The paper proposes a learning-to-learn algorithm for the classic problem of compressed sensing. The algorithm is based on a deep unrolling of a non-convex optimization procedure from a prior work (Wang et al, 2014) in the literature. The paper contains some theory (for capacity and generalization) and experiments, establishing the efficacy of the algorithm.

**Summary Of The Review:**

The paper aims at an important problem and provides a novel solution. The theory and experiment are generally solid. I believe the work could have future impact.

---

> ### Author Response · Authors · 2021-11-12
> **Initial response and plan for the revised paper**
>
> We thank the reviewer for the overall positive comments and really appreciate the high-quality questions. We present an initial response below. Please let us know if there are more suggestions that we should consider in our revised paper.
>
> > **"our analysis makes novel connections between the generalization ability and algorithmic properties such as stability and convergence". The author(s) should be careful about the scope of this claim. The connection between algorithmic stability and generalization is textbook classic.**
>
> We fully understand the reviewer's point - the connection between generalization bound and algorithmic stability is well-known. However, when people talk about this well-known connection, they are referring to **the stability of the meta-algorithm**  (such as SGD and Adam) which is used to optimize the parameters in the deep model. However, what if the algorithm is used as a part of the deep model? How is the **stability of the algorithm used as layers** in the deep model related to the generalization of the deep model? This is what our claim is talking about. Therefore, we say we make novel connections.
>
> In the well-known connection, the referred algorithm is optimizing the deep model. In our connection, the referred algorithm plays a different role. It is used as the layers in the deep model. We will think about how to make this clear to avoid being like an overclaim.
>
> > **Theorem 3.1 appears somewhat weak. The claim is that "there exists a set of parameters $\theta$ such that the estimation error is small". Is there any (theoretical) justification why the algorithm would find such good parameters $\theta$?**
>
> We know that in statistical learning, the generalization error can be bounded by training error + generalization gap. The purpose of our Theorem 3.1 is to say **the training error can be very small**, by showing the existence of a good set of parameters. We would first clarify that our purpose is not to find **the specific set of parameters** indicated in our theorem but to prove a small upper bound for the minimal training error that can be achieved by our deep model. This is referred to as the capacity (or expressiveness and approximation power) of the model.
>
> We agree that it is also important to show the effectiveness of the meta-algorithm (such as SGD, Adam) for minimizing the training loss, because the actual training error depends on **both** the capacity of the model and the meta-algorithm for optimizing the loss. However, analyzing the meta-algorithm can require the scoop of another paper.
>
> Finally, the analysis of the model capacity by itself is important and valuable. For instance, analyzing the capacity of deep models for algorithm learning is the main and almost only contribution in a sequence of top conference papers cited in the Related Work Section (Chen et al., 2018; Liu et al., 2019a; Wu et al., 2020; Kim & Park, 2020).
>
> > **In Section 4 on generalization, the author(s) should clarify the distributional assumptions here. Are we assuming random iid design? Is**
> $\beta^*$ **also distributional? (Note that Assumption 3.1 is a set of deterministic conditions.)**
>
> Interesting questions!
>
> - **distributional?** Yes. In Theorem 4.1, we simply said we *assume $D_m\sim \mathbb{P}(\mathcal{P})^m$*. This assumption means the problem $(Z_{1:n}, \beta^*)$ is distributional, following some probability distribution $\mathbb{P}(\mathcal{P})$ on the problem space. We will make this more clear.
>
> - **deterministic conditions.** It is really a good reading that the reviewer finds the assumption deterministic. Many statements become deterministic because of our use of the term $\nabla_\beta L(Z_{1:n},\beta^*)$ throughout the paper. This term reveals how well the samples $Z_{1:n}$ can represent the whole population by measuring the loss gradient at the true parameter $\beta^*$. Although we assume $(Z_{1:n}, \beta^*)$ follow some problem distribution $\mathbb{P}(\mathcal{P})$, we do not explicitly assume $Z_{1:n}$ are i.i.d samples of the model parameterized by $\beta^*$. However, the term $\nabla_\beta L(Z_{1:n},\beta^*)$ appears in the statistical error in Theorem 3.1. We know that if we assume i.i.d samples, we can further bound the norm of $\nabla_\beta L(Z_{1:n},\beta^*)$ by a rate in terms of $n, p$, and $s^*$ with high probability, and it will tend to 0 almost surely when $n$ tends to infinity.
>
> > **The paper does not compare PLISA with other classic algorithms in the sparse recovery literature.**
>
> Due to the use of non-convex penalties, when we choose classic algorithms as the baseline, we focus on algorithms designed for non-convex problems. Other than the APF baseline, we tried another non-convex algorithm by (Loh 2015). Unfortunately, it works worse than APF so we did not report it. However, we will add experiments on more challenging settings and real datasets as suggested by other reviewers to make our experiments more promising.

---

> > ### Author Response · Authors · 2021-11-12
> > **Response to minor issues mentioned by the reviewer**
> >
> > > There has been a few works on unrolling or differentiable learning of classic algorithms. For example, https://arxiv.org/abs/1910.13984 and https://arxiv.org/abs/1903.08850. This is very closely related to the nature of this work. The authors should make a reference.
> >
> > Thank you for pointing out the related works. They seem to be very related at our first glimpse. We will read them in more detail to see how to discuss these works in the revised paper.
> >
> > > I am not familiar Wang, et al (2014). It would be helpful if the author(s) could give a quick rundown of the algorithm therein, perhaps in the appendix. What are the main features of Wang, et al (2014) that PLISA borrows?
> >
> > Yes, we can statement the algorithm in Wang, et al (2014) in the appendix to be self-contained.
> >
> > This algorithm is guaranteed to converge to a sparse local minimum for a non-convex learning objective. This is the main feature that we hope PLISTA can have because PLISTA contains non-convex penalties and possibly non-convex likelihood loss.
> >
> > Overall, PLISA has a very similar structure as this algorithm, except that:
> > - The step size in the original algorithm is obtained by line-search and it is a learnable parameter in PLISA.
> > - The number of steps K in each cell in PLISTA is fixed, but it is determined by a stopping criterion in the original algorithm.
> > - Most importantly, PLISTA contains element-wise regularization parameters and a learnable combination of penalty functions. In the original algorithm, a penalty is prefixed and a single regularization is used across all entries.

---

> ### Author Response · Authors · 2021-11-23
> **Paper revision and further response**
>
> Dear Reviewer AZru,
>
> Thank you again for the constructive comments! We have posted our revised paper, in which we include the following revisions to address your concerns.
>
> &nbsp;
>
> *** Our claim ***
>
> &nbsp;
>
> We modify our claim on “connections between the generalization ability and algorithmic properties” to “connections between the generalization ability and algorithmic properties such as stability and convergence **of the unrolled algorithm**” to emphasize that we are talking about the stability of the algorithm used as a part of the architecture, not the algorithm used for training the architecture.
>
> &nbsp;
>
> *** More comprehensive experiments as suggested
>
> &nbsp;
>
> We include two additional sets of experiments:
> - To demonstrate the method in a more sophisticated setting other than linear regression, we consider the sparse precision matrix estimation problem in Gaussian graphical models and report the results in section 7.1.2. In this experiment, we include two more classic algorithms GISTA and GLASSO as the baseline. Regarding learning-based methods, we include GGM (a CNN-based architecture) as the baseline.
> - To demonstrate the method on real-world datasets, we conduct experiments on 3 public datasets under the unsupervised sparse linear regression learning setting, in which the true parameters are unavailable in the dataset. Real-world datasets may not satisfy the theoretical assumptions in this paper. This experiment is conducted only to demonstrate the robustness of the proposed method.
> - Apart from the experiments on real-world datasets under the unsupervised setting, we also explain how our theorems can extend to analyze the unsupervised learning setting in Section 5.
>
> &nbsp;
>
> ***Related works
>
> &nbsp;
>
> We thank the reviewer for referring us to these papers! We’ve read them and they are relevant, so we’ve cited these two papers after our statement “Many works share the idea of unrolling or differentiating through algorithms to design the architecture” in the related work. Furthermore, we stated, “We will refer the audience to Shlezinger et al. (2020); Chen et al. (2021) for a more comprehensive summary of related works.” to direct the audience to a more comprehensive list of related works.
>
> &nbsp;
>
> ***Details of the algorithm in Wang, et al (2014)
>
> &nbsp;
>
> Thank you for the suggestion on improving the clarity of the paper! In the revised version:
> -  We include the Algorithm box of Wang, et al (2014)  (copied from their paper) in Appendix H and mention it in the main paper.
> -  In the initial paper, we present the difference between our architecture and Wang, et al (2014) in multiple paragraphs separately. Now in the revised paper, we summarize them in a single paragraph (near Algorithm 1, highlighted in blue color) so that the audience can better understand their differences.
>
> &nbsp;
>
> ===
>
> We hope our revision helps to address your concerns. We are open to discussion in the remaining week of the discussion period.

---

### Official Review · Reviewer_uCY5 · 2021-11-02

**Correctness:** 3
**Technical Novelty And Significance:** 2
**Empirical Novelty And Significance:** 2
**Recommendation:** 6
**Confidence:** 3

**Main Review:**

My first concern is that there is not enough interest or applications for learning-based compressed sensing. It is not a good assumption that there exists a supervised dataset of sparse recovery instances available for tuning the parameters of the proposed algorithm. The fact that all experiments in this paper are based on synthetic datasets and no real datasets are used in this paper confirms my concern.

In terms of presentation, I think it would help if the authors discussed the path-following algorithm a bit since their result is based on this algorithm.

Moreover, the algorithm is recursive and each recursion of the algorithm involves running an instance of the path-following algorithm. So even after fully training the parameters, just evaluating the algorithm on an input instance is several times slower than the original path-following algorithm. The complexity is even much worse during the training phase because one needs to evaluate the whole algorithm multiple times and compute its derivatives with respect to the learnable parameters. The paper does not analyze the runtime complexity, unfortunately.

On page 5, after assumption 3.1, it is stated that condition b is weaker than RIP. It seems to me that when the loss function is least squares, then condition b is exactly the RIP.

Theorem 3.1 is somehow telling that the optimal parameters give a small error but it does not tell you how easy it is to find the optimal parameters. In particular, the parameters need to be found by solving some optimization problem but it is not even clear if this optimization problem is convex or not. If the problem is non-convex, you would only be able to find a local minimum and Theorem 3.1 would become useless.
Furthermore, \kappa_n in Theorem 3.1 is defined ambiguously. It needs to be defined more specifically.


-------------------------------- Score after revision -----------------------

The revised version looks better now and seems to address my concerns. Therefore, I raise my score to 6.



**Summary Of The Paper:**

The paper proposes a learning-based sparse regression or compressed sensing algorithm. The paper assumes that a supervised dataset of sparse recovery instances is available and uses the dataset to fine-tune the hyperparameters of a sparse recovery algorithm known as the path-following algorithm. The proposed algorithm has a recursive structure where at each recursion it needs to solve an instance of the path-following algorithm with different hyperparameters.

**Summary Of The Review:**

I think there is not enough interest or applications for learning-based compressed sensing. The fact that all experiments in this paper are based on synthetic datasets and no real datasets are used in this paper confirms my concern.


Moreover, the algorithm is recursive and each recursion of the algorithm involves running an instance of the path-following algorithm. So even after fully training the parameters, just evaluating the algorithm on an input instance is several times slower than the original path-following algorithm. The complexity is even much worse during the training phase because one needs to evaluate the whole algorithm multiple times and compute its derivatives with respect to the learnable parameters. The paper does not analyze the runtime complexity, unfortunately.


Theorem 3.1 is somehow telling that the optimal parameters give a small error but it does not tell you how easy it is to find the optimal parameters. In particular, the parameters need to be found by solving some optimization problem but it is not even clear if this optimization problem is convex or not. If the problem is non-convex, you would only be able to find a local minimum and Theorem 3.1 would become useless.

---

> ### Author Response · Authors · 2021-11-12
> **Initial response: clarification for some misunderstandings and plans for the revised paper (Part 1)**
>
> We appreciate the effort and time of the reviewer! It seems some misunderstandings may have affected the reviewer's evaluation. We hope our clarifications can address the reviewer's concerns and the reviewer can re-evaluate the paper. We are open to any more discussions.
>
> > **"Each recursion of the algorithm involves running an instance of the path-following algorithm. So even after fully training the parameters, just evaluating the algorithm on an input instance is several times slower than the original path-following algorithm. The complexity is even much worse during the training phase because one needs to evaluate the whole algorithm multiple times and compute its derivatives with respect to the learnable parameters. The paper does not analyze the runtime complexity, unfortunately."**
>
> 1. We would like to sincerely correct this factual error. Our algorithm PLISA is **NOT** running an instance of the path-following algorithm in each recursion.
>
> - As we stated, “The key idea of path-following algorithms is creating a sequence of surrogate objective functions to gradually approach the target objective function. Following the similar design-logic of the path-following algorithm in Wang et al. (2014), PLISA is designed to sequentially approximate the local minimizers of a sequence of objectives.”
>
> - **Both** the original path-following algorithm and our PLISA consider a sequence of objectives. Each recursion is minimizing one objective by **running a K-step modified proximal gradient algorithm** (Eq. 4 and 5), instead of running an instance of the path-following algorithm.
>
> 2. Fig. 4 has plotted the **Error VS Wall-clock time** curves for all algorithms.
>
> - The caption of Fig. 4 says “Since APF takes a long time to converge, its curves are outside the range of these plots.” Here APF is the path-following algorithm. It is significantly slower than all learning-based algorithms. We also stated at the beginning of Page 9 that “PLISA has been converging much faster than the classic algorithm APF”.
> - APF is slower because the step sizes are determined by line-search, and in each recursion the number of steps K is determined by a convergence criterion, which produces a large K in experiments.
>
> 3. The runtime complexity is essentially revealed in the convergence rate of the algorithm. We have shown in Theorem 3.1 that the optimization error decreases exponentially in KT.
>
> > **The parameters need to be found by solving some optimization problem but it is not even clear if this optimization problem is convex or not. If the problem is non-convex, you would only be able to find a local minimum and Theorem 3.1 would become useless.**
>
> We disagree that analyzing the model capacity (sometimes referred to as expressiveness or approximation power) is useless. Showing the existence of a good model in the parameterized model space is a standard and widely accepted argument in learning theory. If it is useless, the key Theorems in all the following **recent top conference papers** are useless.
> - [NeuRIPs spotlight]: Theorem 2 in Theoretical Linear Convergence of Unfolded ISTA and its Practical Weights and Thresholds
> - [ICLR Spotlight]: Theorem 2-3 in Sparse Coding With Gated Learned ISTA
> - [ICLR]: Theorem 1 in ALISTA: Analytic Weights Are As Good As Learned Weights in LISTA
> - [AAAI]: Theorem 1 in Learned Extragradient ISTA with Interpretable Residual Structures for Sparse Coding
> - [ICML]: Theorem 1 in Differentiable Linearized ADMM.
> - The list can go on
>
> The key and almost only contribution in these papers are showing the learning-based algorithm **can achieve** a small error if a proper set of parameters are selected, **without** proving such parameters can be obtained by the meta-optimizer such as SGD, Adam, etc. We also want to highlight that, more than the capacity analysis conducted in these papers, our paper has further supplied the generalization analysis in Theorem 4.1. Theorem 4.1 is a key contribution that contains novel techniques and reveals new and interesting generalization bounds that match the empirical observations. Theorem 4.1 should not be ignored by the reviewer when scoring the 'technical novelty and significance'.
>
> We totally understand that proving the effectiveness of the meta-optimizer (SGD, Adam, etc) is important in deep learning theory, but we argue that proving the model capacity is important and non-trivial by itself. Studying the meta-optimizer is another topic that can require the scoop of another paper.

---

> > ### Author Response · Authors · 2021-11-12
> > **Initial response: clarification for some misunderstandings and plans for the revised paper (Part 2)**
> >
> > (This is part 2 of the initial response.)
> >
> > > **"It seems to me that when the loss function is least squares, then condition b is exactly the RIP"**
> >
> > For least-squares loss, RIP assumes there exists some constant $\delta\in(0,1)$ such that $\frac{v^\top \nabla^2 L v}{|v|^2}\in[1-\delta,1+\delta]$ whereas our assumption is $\frac{v^\top \nabla^2 L v}{|v|^2}\in[\rho_-,\rho_+]$ for some $\rho_-,\rho_+\geq 0$. The ranges are different although they look similar. In fact, this claim was made in the papers (Zhang, 2010b) and (Wang et al., 2014), which are cited when we make this statement in the paper. On page 2183 of (Wang et al., 2014), justification can be found. We will point out this in the revised paper to make it more clear!
> >
> > > **"I think there is not enough interest or applications for learning-based compressed sensing. The fact that all experiments in this paper are based on synthetic datasets and no real datasets are used in this paper confirms my concern."**
> >
> > We believe when the reviewer said, “there is not enough interest or applications for learning-based compressed sensing”, he/she didn’t mean it. This is a new but rapidly growing area that has attracted a lot of interest. Among the 247 papers cited by the survey, ‘Learning to Optimize: A Primer and A Benchmark’, more than 1/4 are learning-based compressed sensing papers published in recent years, including the ICML 2020 best paper ‘Tuning-free Plug-and-Play Proximal Algorithm for Inverse Imaging Problems’. **Note that this ICML best paper also used supervised datasets in experiments.**
> >
> > Therefore, we believe the real concern of the reviewer is that we didn’t demonstrate the method on a realistic dataset, and there may be concerns about the supervised setting stated in the paper. To address this concern, we have the following plan for the revised paper:
> > 1. We will add a set of real experiments.
> > 2. We will extend the paper to incorporate an unsupervised scenario by using the likelihood (or reconstruction error in the case of compressed sensing) as the meta training loss. This will lead to an extra statistical error term in the theorem due to the gap between the training loss and the true error that we aim to minimize.
> >
> > ---
> >
> > Please let us know if there are further suggestions for our paper. We are open to discussion.

---

> > ### Comment · Reviewer_uCY5 · 2021-11-16
> > **Response to authors**
> >
> > Thanks a lot for the feedback.
> > I am still confused about your algorithm.
> >
> > Looking at Figure 2, it seems to me that your algorithm consists of T cells, where in each cell you run an instance of the APF. In different cells you run instances of APF with different parameters \theta = (\eta, \lambda , w, \alpha). In each cell you are optimizing some objective function that depends on the learnable parameters \theta, and this objective seems to be identical to the APF's objective, isn't this accurate?
> > So the cost of optimizing the objective of each cell is equivalent to the cost of fully running the PFA, no?

---

> > > ### Author Response · Authors · 2021-11-16
> > > **Follow-up Response to Reviewer uCY5**
> > >
> > > We highly appreciate the reviewer's response and involvement in this discussion period. We are happy to clarify more about the confusion and any remaining ones.
> > >
> > > Yes, your summary and understanding of our algorithm are accurate. We think the issue is the misunderstanding of the original APF. In your statement above, if the term **APF** is replaced by **proximal-gradient algorithm**, then it is completely correct.
> > >
> > > Both APF and our algorithm consist of running T many runs of **proximal-gradient algorithm**. To optimize each objective function, both APF and our algorithm run an instance of K-step proximal gradient.
> > >
> > > We are happy to introduce more about the original APF. The idea of path following has been well-studied for sparse recovery problems since 2004. The key idea is constructing a sequence of optimization problems, from easy ones to hard ones, and using the solution of easier problems as the initialization to the harder problems.
> > >
> > > As an example, in our case, APF takes $\lambda_0 = |\nabla L_n(Z_{1:n},0)|$ to construct the 1st objective. This 1st objective is trivial to solve and we know $\beta=0$ is the minimizer. If we gradually decrease the value of $\lambda$ in the objective, the objective will become harder to solve but the statistical recovery accuracy will improve. Statistically, we aim to solve the last objective with small $\lambda$, but computationally, we start from easier objectives, keep the intermediate solutions $\beta_t$ in a fast converging region (around the minimizer of each objective), and gradually approach the solution to the final objective. In such a path-following way, the algorithm is expected to approximate the solution faster than directly optimizing the final objective.
> > >
> > > Therefore, a complete APF consists of solving T many problems by running T instances of the proxima-gradient algorithm to minimize each objective, the same as our algorithm.
> > >
> > > Please let us know if there is any remaining confusion.

---

> > > > ### Comment · Reviewer_uCY5 · 2021-11-17
> > > > **Response to authors**
> > > >
> > > > Ok I see. So both your algorithm and APF have T objective functions to optimize where each objective corresponds to different parameters \theta. The difference between your algorithm and APF is that you learn the parameters \theta while APF assumes that they are hyperparameters that are somehow known. So the "Query Time" of your algorithm and APF are the same and your algorithm requires an additional training time while APF does not. I think there is a crucial need to explain the APF algorithm and mention the differences between this algorithm and yours in your paper.
> > > >
> > > > I'm not sure how expensive is the training of parameters \theta. Do you measure the query time in the experiments section or did you include both training and query time? It is unfair to only compare your query time to APF. I think you need to add new experiments and compare the training time of your algorithm to other learning-based algorithms.

---

> ### Author Response · Authors · 2021-11-23
> **Paper revision and further response**
>
> Dear Reviewer uCY5,
>
> Thank you again for the active engagement in the discussion period! We really appreciate it. We have posted the revised paper, which includes new results to address your initial concerns and the new questions raised during the discussion period, summarized as follows:
>
> &nbsp;
>
> *** Training and testing efficiency ***
>
> &nbsp;
>
> We would like to answer the newly raised question during our discussion first. In our revised paper, we add a new section 7.1.3 to discuss the training and testing time:
>
> Regarding the testing time:
> - Although the “query time” of APF and our method should be similar theoretically, however, since APF performs line-search to select the step sizes, it is much slower in practice, due to the time for line-search and the less optimal step sizes selected.
> - Figure 4 and the newly added Table 1 both show that classic algorithms (APF, GISTA, GLASSO) are in general slower. Moreover, learning-based methods can easily solve **a batch of problems in parallel**, but it is harder for classic algorithms to do that due to the use of line-search and convergence criteria. However, to allow more advantages to classic algorithms, in Figure 4 and Table 1, learning-based methods are **evaluated without using batching**.
>
> Regarding the training time:
> - It is a valid concern whether learning-based methods are expensive to train, so we report the total training time in Table 3 and Table 4 in our revised paper.
> - As mentioned in the paper that we perform grid-search to select the hyperparameters (regularization parameter) in classic algorithms. Therefore, classic algorithms also have a ‘training time’ in our experiments.
> - We can see from Table 3 and Table 4 that training time is not a bottleneck of this problem. Most methods can be trained efficiently. In SPE, classic algorithms even require a longer training time than learning-based methods.
>
> Thanks for the suggestion and hope the additional results can relieve your concerns.
>
> &nbsp;
>
> ***Discussion on the difference between APF and our method
>
> &nbsp;
>
> Thank you for the suggestion on improving the clarity of the paper! In the revised version:
> -  We add the Algorithm box of APF (copied from their paper) in Appendix H and mention it in the main paper.
> - In the initial paper, we present the difference between our method and APF in multiple paragraphs separately. Now we have summarized them in a single paragraph (near Algorithm 1, highlighted in blue color) so that the audience can better understand their difference.
>
>
> We notice the comment that *“The difference between your algorithm and APF is that you learn the parameters $\theta$ while APF assumes that they are hyperparameters”*. To avoid misunderstanding, we would like to further clarify that our $\theta$ is more flexible than hyperparameters in APF. Our $\theta$ represents **entry-wise** regularization parameters to enforce different levels of sparsity to different dimensions, but APF only contains the **uniform** regularization parameter across all dimensions. Theorem 3.1 has illustrated the advantages of using entry-wise regularization.
>
> &nbsp;
>
> ***New experiments
>
> &nbsp;
>
>
> To address the concern raised in the initial review, we include new experiments in the revised paper:
> - To demonstrate the method in a more sophisticated setting other than linear regression, we consider the sparse precision matrix estimation problem in Gaussian graphical models and report the results in section 7.1.2. In this experiment, we include two more classic algorithms GISTA and GLASSO as the baseline. Regarding learning-based methods, we include GGM (a CNN-based architecture) as the baseline.
> - To demonstrate the method on real-world datasets, we conduct experiments on 3 public datasets under the unsupervised sparse linear regression learning setting, in which the true parameters are unavailable in the dataset. Real-world datasets may not satisfy the theoretical assumptions in this paper. This experiment is conducted only to demonstrate the robustness of the proposed method.
> - Apart from the experiments on real-world datasets under the unsupervised setting, we also explain how our theorems can extend to analyze the unsupervised learning setting in Section 5.
>
> &nbsp;
>
> ===
> We hope our revision helps to address your concerns. We thank the reviewer for the previous discussion and we are open to discussion in the remaining week if there is any further concern.

---

### Official Review · Reviewer_7xPS · 2021-11-02

**Correctness:** 4
**Technical Novelty And Significance:** 3
**Empirical Novelty And Significance:** 3
**Recommendation:** 8
**Confidence:** 3

**Main Review:**

The paper is generally well-written. The idea of the algorithm, its precise details, and the assumptions and claims of the theoretical analysis are all presented clearly and in detail. The approach of the algorithm makes sense and seems potentially promising, and targets an important set of problems.

The weakest point of the paper is the experimental section. The experiments are restricted to synthetic data of a limited scale, and to perhaps the most basic sparse recovery setting (linear regression, while the introduction makes the point that the method is more broadly applicable). While I appreciate the theoretical analysis and its validation on synthetic data, the utility of learning-based algorithms hinges to a significant extent on whether they can be substantiated on real data, where the correspondence between the training and test data is not as structured and clear as in the synthetic case. At present, it is hard to say from the paper whether the proposed method is plausibly useful in practice.

There seem to be parts of related literature that are not mentioned. I am not sure if they are directly related, and perhaps the authors could comment on it in the response. These include some additional learning-based algorithms for latent parameter recovery (e.g., [a,b], though they move away from sparsity), and generalization bounds for learning-based algorithms [c,d].

[a] Bora, A., Jalal, A., Price, E. and Dimakis, A.G., Compressed sensing using generative models, ICML 2017
[b] Wu, S., Dimakis, A., Sanghavi, S., Yu, F., Holtmann-Rice, D., Storcheus, D., Rostamizadeh, A. and Kumar, S., Learning a compressed sensing measurement matrix via gradient unrolling, ICML 2019
[c] Gupta, R. and Roughgarden, T., A PAC approach to application-specific algorithm selection, SICOMP 2017
[d] Balcan, M.F., DeBlasio, D., Dick, T., Kingsford, C., Sandholm, T. and Vitercik, E., How much data is sufficient to learn high-performing algorithms? generalization guarantees for data-driven algorithm design, STOC 2021


Post-rebuttal update: The authors' response and revisions of the paper answered my main concerns, primarily the lack of empirical evaluation on real data, which they have now added. As a result, I have revised my numerical assessment of the empirical merit of the paper, and have upgraded my overall score. Pending some concerns raised by other reviews, which would be resolved in the internal discussion phase, from my point of view the paper can be accepted.

**Summary Of The Paper:**

The paper suggests a learning-based algorithm for sparse recovery, where the goal is to recover a latent sparse parameter vector from observations (say, in the basic setting of sparse linear regression, the observations are noisy linear measurements). The algorithm is based on "unrolling" an existing non-learned iterative algorithm, which means fixing some number of iterations and learning some of the algorithm's internal parameters by optimization on an existing training dataset of related problems and their known solutions. The paper theoretically studies the capacity and generalization properties of the proposed algorithm, and presents some experimental results on synthetic data.

**Summary Of The Review:**

This is a well-written paper that presents a plausible approach to a meaningful problem, but falls short in empirical validation.

---

> ### Author Response · Authors · 2021-11-12
> **Initial response and plan for the revised paper**
>
> We thank the reviewer for the overall positive feedback and the meaningful questions! We give an initial response in the following. Please let us know if there are more suggestions that we should consider in our revised paper.
>
> **Experiments**
>
> Thanks for your recognization of our synthetic experiments for verifying the theorems. We agree that to show its practical usage it's better to demonstrate it on realistic datasets. Therefore, in the revised paper, we plan to (1) add a new set of experiments of real datasets, and (2) add a new set of experiments on a more difficult setting - the precision matrix estimation problem in graphical models.
>
> **Related Literature**
>
> Thank you for pointing out the related works. They seem to be very related at our first glimpse. We will read them in more detail to see whether and how to discuss these works in the revised paper.

---

> ### Author Response · Authors · 2021-11-23
> **Paper revision and further response**
>
> Dear Reviewer 7xPS,
>
> Thank you again for the constructive comments! We have posted our revised paper, in which we include the following revisions to address your concerns.
>
> &nbsp;
>
> *** More comprehensive experiments as suggested ***
> - To demonstrate the method in a more sophisticated setting other than linear regression, we consider the sparse precision matrix estimation problem in Gaussian graphical models and report the results in section 7.1.2. In this experiment, we include two more classic algorithms GISTA and GLASSO as the baseline. Regarding learning-based methods, we include GGM (a CNN-based architecture) as the baseline.
> - To demonstrate the method on real-world datasets, we conduct experiments on 3 public datasets under the unsupervised sparse linear regression learning setting, in which the true parameters are unavailable in the dataset. Real-world datasets may not satisfy the theoretical assumptions in this paper. This experiment is conducted only to demonstrate the robustness of the proposed method.
> - Apart from the experiments on real-world datasets under the unsupervised setting, we also explain how our theorems can extend to analyze the unsupervised learning setting in Section 5.
>
> &nbsp;
>
> *** Missing Related Works ***
>
> Thank you very much for referring us to these works. We’ve read them. They are all related and now cited.
>
> Especially, the works of [c,d] on generalization bounds for learning-based algorithms are very related and they are very nice works! However, they have a totally different focus. Unlike our work that analyzes the Lipschitz continuity (Lemma 4.1) of the deep unrolling architecture, their works focus on scenarios **when the Lipschitz continuity is unavailable**. It can be seen from their papers that they focus more on discrete algorithms such as dynamic programming, integer programming, etc, and mainly learn the optimization objectives. Their works are **less relevant to algorithm unrolling** which is a focus in our work. As a consequence, they did not have the motivation to bridge the connection between the generalization and algorithmic properties as we did.
>
> In the revised paper, we have briefly discussed these differences.
>
> &nbsp;
>
> ===
>
> We hope our revision helps to address your concerns. We are open to discussion in the remaining week of the discussion period if there is any further concern.

---

### Author Response · Authors · 2021-11-23
**Summary of major revisions**

We would like to thank all the reviewers for their careful reading, detailed comments, and active engagement in the discussion period. We have incorporated your constructive suggestions into the revised version. The major revisions are highlighted in blue color in the revised paper. We summarized them below:

&nbsp;

*** Extension to unsupervised setting ***

We add a new section “5 Extension to Unsupervised Learning-to-learn Setting” to explain how our work can extend to the setting when the ground-truth parameter is unavailable in the training set.
1. Methodologically, an unsupervised training loss is defined by evaluating the empirical loss (e.g., likelihood) on samples.
2. Theoretically, the generalization error contains an additional statistical error term compared to the supervised setting, which appears because of the gap between the unsupervised loss and the true error that we aim to optimize.

&nbsp;

*** Additional experiments ***

As suggested by the reviewer, we include two additional sets of experiments:
1. In section 7.1.2, we consider the sparse precision matrix estimation problem in Gaussian graphical models to demonstrate the method in a more sophisticated setting.
2. In section 7.2, we conduct experiments on 3 public datasets under the unsupervised setting. Real-world datasets may not satisfy the theoretical assumptions in this paper. This experiment is conducted only to demonstrate the robustness of the proposed method.

Because of the page limit, after adding these new results, we move the ‘ablation study’ in the initial version to Appendix.

&nbsp;

*** More details ***

We’ve improved the clarity of the paper by adding
1. More related works
2. Details of the classic path-following algorithm and its difference from our method
3. The training time required for all methods and the discussion (section 7.2)


Thank all the reviewers for the constructive suggestions on paper refinement!

---

### Decision · Program_Chairs · 2022-01-20

**Decision:**

Accept (Poster)

**Comment:**

Dear Authors,

The paper was received nicely and discussed during the rebuttal period.
There is consensus among the reviewers that the paper should be accepted:

-  This paper does contribute solidly to a timely topic of theoretical understanding of sparisty recovery with deep unroling.
-  The original version had very limited experiments and only synthetic ones, which raised concerns about whether the setting is motivated and whether the algorithm works on actual real data. The revision fixed that to an extent with some experiments on real data.

Yet, there are still some concerns that we suggest to be tackled for the final version:
- The capacity analysis is carried out inside a strongly convex regime while the algorithm is advocated for nonconvex sparsity recovery (see, e.g., the Discussion at the end of Section 2.1 );
- The analysis is relatively loosely connected to the adopted fist-order optimization procedure;
- While the depth of network plays a role in the upper bound of Equation (15), its real impact on generalization gap looks quite limited.

The above are just suggestions to be looked more carefully, but there are not necessary.

The current consensus is that the paper deserves publication.

Best
AC